Letter

# Pyroptotic cell corpses are crowned with F-actin-rich filopodia that engage CLEC9A signaling in incoming dendritic cells

Caroline L. Holley [1,2], Mercedes Monteleone [1] ✉, Daniel Fisch [3], Alexandre E. S. Libert [1,4], Robert J. Ju [1,4], Joon H. Choi [3], Nicholas D. Condon[1], Stefan Emming [1], Joanna Crawford[1], Grace M. E. P. Lawrence[1], Jared R. Coombs [1], James G. Lefevre [1,5], Rinie Bajracharya[1], Mireille H. Lahoud [6], Alpha S. Yap[1], Nicholas Hamilton[1], Samantha J. Stehbens[1,4], Jonathan C. Kagan [3], Nicholas Ariotti [1], Sabrina S. Burgener [1,7] & Kate Schroder [1,7] ✉

While apoptosis dismantles the cell to enforce immunological silence, pyroptotic cell death provokes inflammation. Little is known of the structural architecture of cells undergoing pyroptosis, and whether pyroptotic corpses are immunogenic. Here we report that inflammasomes trigger the Gasdermin-D- and calcium-dependent eruption of filopodia from the plasma membrane minutes before pyroptotic cell rupture, to crown the resultant corpse with filopodia. As a rich store of F-actin, pyroptotic filopodia are recognized by dendritic cells through the F-actin receptor, CLEC9A (DNGR1). We propose that cells assemble filopodia before cell rupture to serve as a posthumous mark for a cell that has died by gasdermin-induced pyroptosis, or MLKL-induced necroptosis, for recognition by dendritic cells. This study reveals the spectacular morphology of pyroptosis and identifies a mechanism by which inflammasomes induce pyroptotic cells to construct a de novo alarmin that activates dendritic cells via CLEC9A, which coordinates the transition from innate to adaptive immunity[1,2].

The immune system first senses microbes using an integrated suite of innate immune receptors that assess the level of organismal threat and launch a proportionately scaled response[3]. Current concepts suggest that when smaller-scale responses fail to eliminate pathogens, the innate immune response escalates, risking collateral damage for antimicrobial defense. As a last resort, inflammasomes are launched to drive cell death, potent inflammatory programs and tissue damage[3]; if this fails to clear an infection, antigen-specific adaptive responses become critical for pathogen elimination. Genetic deficiency of inflammasome proteins causes maladaptive antigen-specific immune responses during infection, vaccination, cancer and autoimmunity[4,5].

Thus inflammasomes, as the final weapon in the innate immune arsenal, instruct the transition from innate to adaptive immunity.

How inflammasomes instruct dendritic cells to initiate adaptive immunity is poorly understood[4]. Inflammasomes are assembled by sensor proteins such as NLRP3 and caspase-4 (CASP4; CASP11 in mice). NLRP3 senses cellular perturbations such as ionic flux induced by bacterial toxins (for example, nigericin) and responds by recruiting an adaptor protein, ASC, which in turn recruits CASP1 and activates its latent protease function. CASP4 and CASP11 (CASP4/11) are related proteases that are activated by cytosolic lipopolysaccharide (LPS) from gram-negative bacteria. Active CASP1 and CASP4/11 both

**Fig. 1 | NLRP3 and CASP11 inflammasome activators trigger the extrusion of F-actin-rich filopodia. a,b,** Live fast Airyscan confocal imaging of BMDMs expressing the PLCδ-PH-GFP PIP2 probe and labeled with 1 μM SiR-actin. Macrophages were primed for 4 h with LPS and stimulated with nigericin (NIG) (**a**) or primed with PAM3CSK4 (PAM, 16 h) and transfected with LPS using FuGENE HD (FuG) (**b**). Images are MIP of Z-stack acquisitions, and representative of n = 3 independent biological experiments. **c,** Quantification of filopodia per cell in each condition of **a** and **b** using manual counting (n = 25 per condition), analyzed using the Mann–Whitney two-sided test. **d,** Airyscan confocal imaging of pyroptotic projections (phalloidin, α-MYO10) for nigericin-stimulated macrophages. Plots of mean fluorescence intensity (MFI) in boxed regions are displayed below. Images are single Z-planes and representative of n = 3 independent biological experiments. **e,** BMDM ectopically expressing PLCδ-PH-GFP (cell membrane) were grown in suspension. 3D cell surface rendering of BMDM and F-actin (SiR-actin) remodeling when pyroptosis was activated (LPS + NIG). **f–h,** Quantification of filopodia per cell from HBEC (**f**), BMDMs (**g**)

or BMDCs (**h**) treated as indicated to induce pyroptosis (LPS + NIG (nigericin), PAM + LPS transfection) or sublytic inflammasome signaling (LPS + PGPC). Protrusions from 25–130 randomly chosen cells with filopodia were counted per condition, and analyzed by a Mann–Whitney two-sided test with Bonferroni correction. **i–j,** WT and *Gsdmd*⁻/⁻ mice (9–10 female mice per condition) were intraperitoneally (i.p.) challenged with LPS (or two mice per genotype for PBS) for 4 h, 3 days after 2% thioglycollate exposure. The peritoneal lavage cells were labeled with phalloidin and DAPI. **i,** The percentage of cells with filopodia per field of view (FOV) (×40 magnification) was enumerated, with at least 100–300 cells recorded from each mouse (3–5 randomly acquired fields of view), analyzed by Mann–Whitney two-sided test. **j,** Images are Z-stack MIPs and representative of n = 9–10 mice from two independent experiments. Scale bars, 10 μm (**a,b,j**), 2 μm (**d**), 5 μm (**e**). Violin plots show mean (solid line) and first and third quartiles (dotted lines). Statistical significance: *P ≤ 0.05; **P ≤ 0.01; ***P ≤ 0.001; ****P ≤ 0.0001.

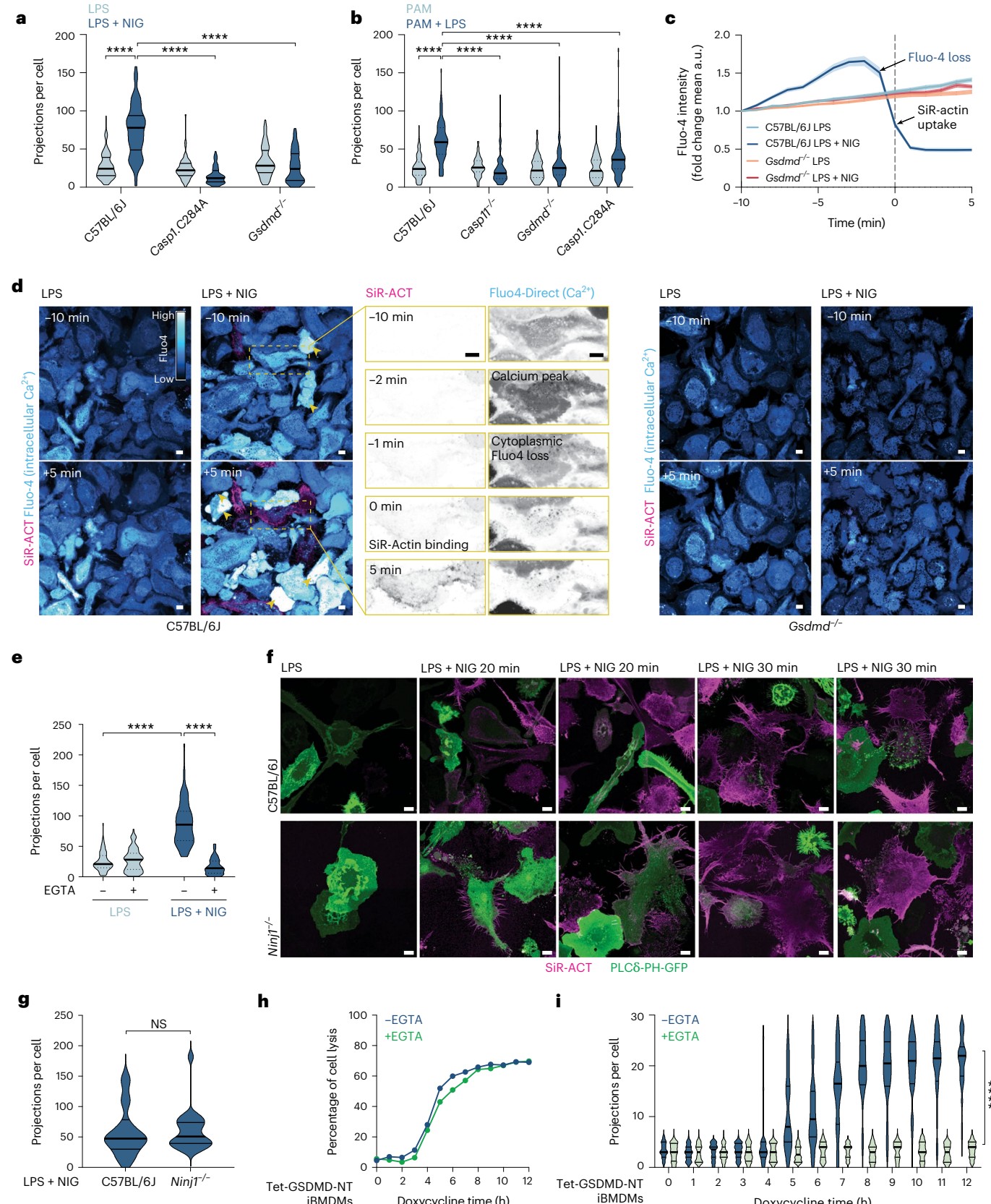

cleave Gasdermin-D (GSDMD) to liberate a pore-forming fragment (GSDMD-NT)[6,7] that permeabilizes the plasma membrane. This can ultimately lead to plasma membrane rupture (PMR)[8] by NINJ1 in pyroptosis, a multi-step pathway of lytic cell death[9]. Pyroptosis is strongly immunostimulatory, in part because interleukin (IL)-1β and -18 are secreted through GSDMD pores before PMR[10–12]. These cytokines upregulate dendritic cell costimulatory molecules, and influence T cell differentiation and effector function[4,13]. Studies with mice unable

**Fig. 2 | GSDMD-NT is sufficient for calcium-induced pyroptotic filopodia, which sprout independently of cell rupture. a,b**, Quantification of filopodia per cell from macrophages in which pyroptosis was stimulated (LPS + NIG, **a**; PAM + LPS, **b**). Data were analyzed using a Kruskal–Wallis test with Dunn's multiple testing correction ($n = 75$ cells per condition from three independent biological experiments). **c**, Quantification of Fluo-4 ($Ca^{2+}$ reporter) intensity extracted from a 45 min acquisition following nigericin-induced pyroptosis in BMDM, relative to the first frame in which the cells become positive for SiR-actin (time 0 min). A total of 72 cells per condition were analyzed pooled from three independent biological experiments. Data are mean intensity (in arbitrary units, a.u.) ± s.e.m. **d**, Spinning disk confocal imaging of macrophages incubated with Fluo-4 and SiR-actin. WT or *Gsdmd*$^{-/-}$ LPS-primed BMDM were left untreated or exposed to nigericin. Yellow arrows indicate cells with a spike in calcium that precedes cell rupture. Images are representative of $n = 3$ independent biological experiments. **e**, Quantification of filopodia from macrophages pretreated with 2 mM EGTA 1 h before nigericin in Extended Data Fig. 5c. Data were analyzed using a Kruskal–Wallis test with Dunn's multiple testing correction ($n = 75$ cells per condition from three independent biological experiments). **f**, Live fast Airyscan confocal imaging of WT versus *Ninj1*$^{-/-}$ LPS-primed BMDM expressing the PLCδ-PH-GFP probe, labeled with SiR-actin and stimulated with nigericin for up to 30 min. Images are Z-stack MIPs and representative of $n = 3$ independent biological experiments, analyzed by an unpaired two-sided Mann–Whitney test. **g**, Quantification of filopodia from nigericin-treated BMDM. Tile scan images were collected between 20–30 min post-NLRP3 activation from $n = 3$ independent biological experiments. Filopodia were counted in SiR-actin-positive cells. **h,i**, LPS-primed iBMDM stably expressing Tet-GSDMD-NT were exposed to doxycycline ± EGTA-AM, and analyzed for cell lysis, presented as mean of technical replicates (**h**) or filopodia (**i**). Filopodia were quantified from 40 randomly chosen cells per condition and time point, corresponding to Extended Data Fig. 5d, and analyzed by a Mann–Whitney two-sided test with a false discovery rate multiple testing correction. Scale bars, 10 μm. Violin plots show mean (solid line) and first and third quartiles (dotted lines). Statistical significance ****$P ≤ 0.0001$.

to signal via IL-1β/18 suggest the existence of additional mechanisms of inflammasome-induced immune activation[14], which include soluble factors released during pyroptotic PMR[15] (for example, IL-1α, ATP, HMGB1, eicosanoids[16]).

While apoptosis requires cell dismantling to enforce immunological silence, little is known of the structural architecture of cells undergoing pyroptosis[17–19], and whether the pyroptotic corpse is immunogenic. Given that neutrophils undergoing CASP4/11-induced pyroptosis repurpose their chromatin as an antimicrobial weapon[20], pyroptotic corpses may have important defense functions beyond the cell's lifespan. Actin is the most abundant protein in the cell[21] and, presumably, the cell corpse. Seeking to characterize actin dynamics during pyroptosis, we performed live imaging of macrophages exposed to NLRP3- and CASP11-activating stimuli. Inflammasomes were activated by a two-step method, in which cells were primed (extracellular LPS, PAM3CSK4), then exposed to an activation stimulus (nigericin to activate NLRP3; cytosolic LPS to activate CASP11) and immediately imaged. Lifeact-eGFP-expressing macrophages[22] suddenly erupt projections rich in filamentous actin (F-actin) at the plasma membrane edge shortly after SYTOX uptake, but before PMR that causes loss of green fluorescent protein (GFP) fluorescence (Extended Data Fig. 1a,b and Supplementary Videos 1 and 2). To better capture events before and after PMR, we expressed the PLCδ-PH-GFP probe that associates with phosphatidylinositol-4,5 bisphosphate[10] to label the plasma membrane of primary macrophages, which were incubated with inflammasome agonists plus SiR-actin to label F-actin during cell permeabilization[23]. In the first minutes of imaging, macrophages assemble dynamic plasma membrane ruffles, after which cells cease ruffling and peripheral projections nucleate outwards and grow rapidly, extending until the cell undergoes PMR, indicated by GFP fluorescence loss (Fig. 1a,b and Supplementary Videos 3 and 4). The resultant corpses remain crowned with SiR-actin-labeled projections. Of note, these projections are low-contrast structures that required labeling for visualization, unlike the high-contrast

pyroptotic blisters that burst during PMR that are resolved using label-free approaches. We further investigated actin dynamics using the well-established F-actin stain, phalloidin. Inflammasome-signaling macrophages, identified by their ASC speck, underwent extensive F-actin remodeling in response to NLRP3 (nigericin, ATP), CASP11 (cytosolic LPS) or NAIP-NLRC4 (flagellin) agonists, particularly at the peripheral membrane region where projections assembled (Fig. 1c and Extended Data Fig. 1c–f). The specific NLRP3 inhibitor MCC950 (refs. 24,25) blocked ATP- and nigericin-induced projections, while flagellin triggered projections in unprimed macrophages (Extended Data Fig. 1e,f), indicating that inflammasome activation, but not priming, was required. Some ASC speck-containing cells appeared to be intact, suggesting they were imaged before PMR (Extended Data Fig. 1c,d(i)), while cells that ruptured had lost their cortical actin and nuclei (Extended Data Fig. 1c,d(ii)). Inflammasome-signaling cells possessed a halo of F-actin-rich projections regardless of whether they had undergone PMR, consistent with projections sprouting before PMR.

Pyroptosis-associated projections were morphologically distinct from previously described pyroptotic bodies[26] or actin-independent apoptopodia[27]. Myosin-X (MYO10) localization to the projection tips (Fig. 1d and Extended Data Fig. 1g), established that these projections were filopodia[28]. Human macrophages exposed to nigericin or the NAIP-NLRC4 agonist PrgI also displayed Myosin-X-positive filopodia (Extended Data Fig. 1h). Filopodial assembly did not require cell adherence during in vitro tissue culture, as nigericin-stimulated macrophages cultured in suspension assembled filopodia (Fig. 1e, Extended Data Fig. 2 and Supplementary Video 5). In human bronchial epithelial cells (HBECs), LPS transfection triggers CASP4 inflammasome signaling and pyroptosis[29], and this was accompanied by the assembly of Myosin-X-positive filopodia (Fig. 1f and Extended Data Fig. 3a). Thus, filopodial assembly is a general feature of human and murine myeloid and nonmyeloid cells initiating pyroptosis, with filopodia persisting within the corpse after cell rupture.

**Fig. 3 | Diverse effectors of lytic cell death induce filopodial assembly. a**, Schematic of experimental protocol for cell death induction by death effector-encoding mRNAs. **b–l**, WT BMDMs were transfected with 5-methoxyuridine mRNA encoding GSDMD-NT (**b–d**), GSDMA3-NT (**e–g**), GSDME-NT (**e–g**), MLKL-4HB (**h–j**) or tBID (**k,l**). Cells were analyzed 4 h after mRNA transfection. For GSDMD, cells were treated with or without LPS for 4 h before analysis. Data are from $n = 3$ independent biological experiments. **b,e,h,k**, LDH release assay. Data were verified for normality, and analyzed by one-way analysis of variance (ANOVA). **c,f,i**, Quantification of filopodial projections. For each condition, random fields of view were selected and lysed cells were identified and assessed for the presence of filopodia ($n = 9–13$). Data were analyzed with a Kruskal–Wallis test with Dunn's multiple testing correction or a Mann–Whitney two-sided test.

**d,g,j,l**, Fixed Airyscan confocal imaging of cells labeled with F-actin phalloidin (green) and DAPI (gray). Images are Z-stack MIPs and are representative of $n = 3$ independent biological experiments. **m**, Confocal imaging of fixed WT and *Gsdmd*$^{-/-}$ macrophages labeled with F-actin phalloidin (green) and α-cleaved caspase-3 antibody (magenta). Magnified insets of inverted phalloidin (green) and cleaved caspase-3 (magenta) are shown in gray. Images are Z-stack MIPs, and are representative of $n = 3$ independent biological experiments. **n**, SEM of macrophage corpses produced by intrinsic apoptosis (0.5 μM ABT-737 + S63845 for 2 h; representative of $n = 2$ independent experiments. Scale bars, 10 μm, except where indicated otherwise. Violin plots show the mean (solid line) and first and third quartiles (dotted lines). Statistical significance: *$P ≤ 0.05$; **$P ≤ 0.01$; ***$P ≤ 0.001$; ****$P ≤ 0.0001$.

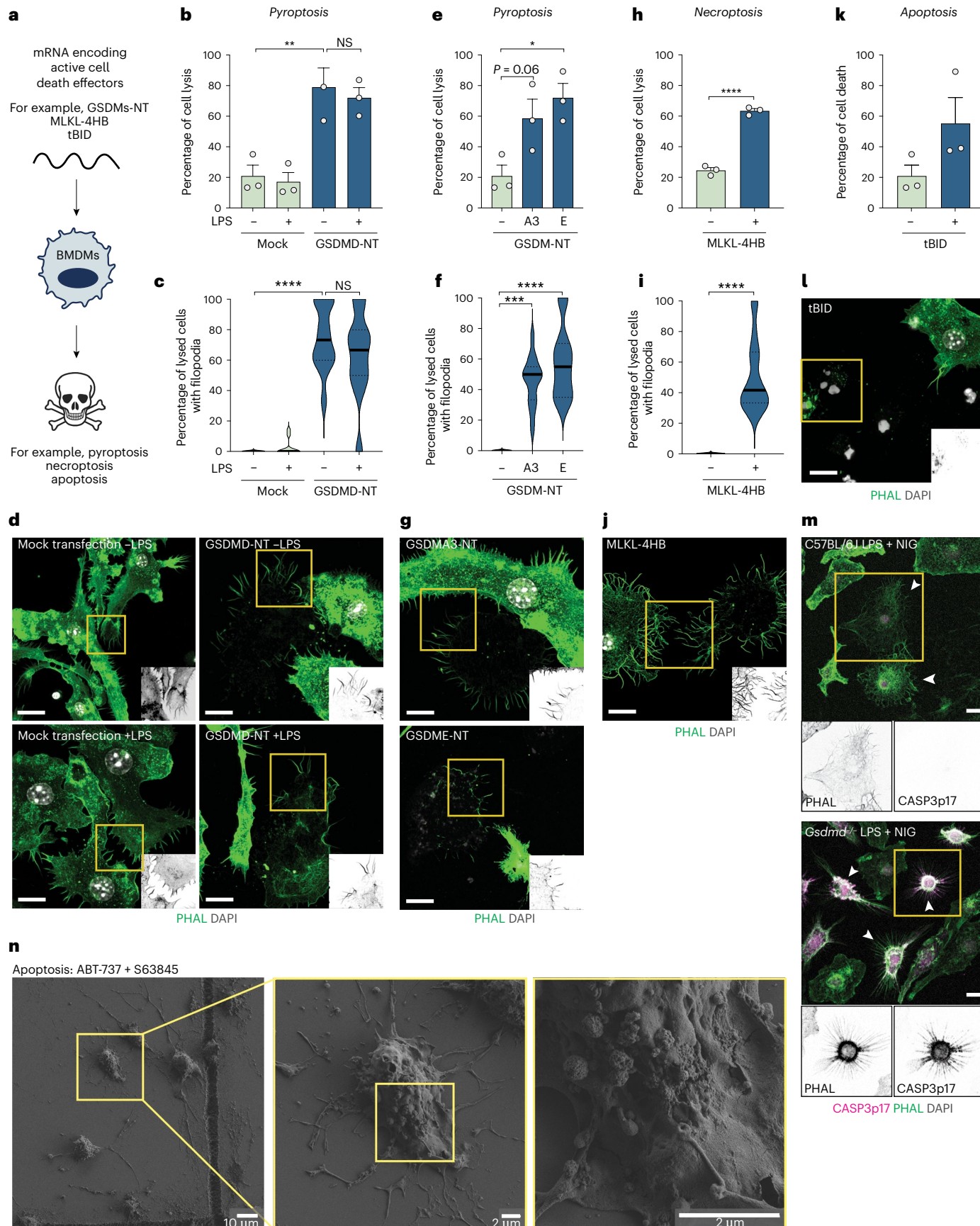

Inflammasome signaling does not always induce PMR and cell lysis[30,31]. Macrophages and dendritic cells exposed to the sublytic inflammasome activator PGPC (1-palmitoyl-2-glutaroyl-*sn*-glycero-3-phosphocholine)[13,31] exhibited a spectacular number of Myosin-X-capped filopodia, which were highly branched (Fig. 1g,h and Extended Data Fig. 3b–g). Thus, PMR is dispensable for inflammasome-induced filopodial assembly.

We next challenged mice with an intraperitoneal injection of LPS. The cells of the peritoneal exudate showed a substantial increase in filopodia as compared to sham controls (Fig. 1i,j), indicating that LPS induces filopodial assembly in vivo.

To determine the mechanisms inducing filopodial assembly, we examined primary macrophages from gene-targeted mice, including *Gsdmd*-deficient mice and mice with a point mutation in the CASP1 catalytic cysteine to render it enzyme-dead (*Casp1^C284A*). Intact *Casp1^C284A* and *Gsdmd^-/-* macrophages with a nigericin-induced ASC speck did not possess pyroptosis-associated filopodia (Fig. 2a and Extended Data Fig. 4a,b). We next examined CASP11-activated macrophages. Given that cytosolic LPS activates the CASP11-GSDMD-NLRP3 signaling axis to assemble ASC specks, we were unable to focus our analyses on ASC speck-containing *Casp11^-/-* and *Gsdmd^-/-* cells. For all genotypes, intact cells were thus randomly selected and quantified for projections. *Casp11* and *Gsdmd* deficiency suppressed LPS-induced filopodial assembly and cell death (Fig. 2b and Extended Data Fig. 4c,d). LPS-stimulated *Casp1^C284A* macrophages containing an ASC speck showed an intermediate phenotype (Extended Data Fig. 4c), consistent with CASP1 contributing to GSDMD cleavage and pyroptosis[32]. GSDMD was also required for filopodial assembly in vivo (Fig. 1i,j). Thus, CASP1 and CASP11, and their substrate GSDMD, are required for inflammasomes to induce filopodial construction. This suggests that CASP1/11-induced GSDMD pores trigger filopodial assembly immediately before pyroptotic PMR.

Calcium is a major regulator of the actin polymerization machinery[33,34] and GSDMD pores are permeable to calcium ions[26,35]. To confirm that GSDMD pores allow $Ca^{2+}$ entry into the cell[26,35], we performed Fluo-4 calcium live cell imaging. Nigericin-stimulated macrophages exhibited marked calcium uptake, which peaked 1–2 minutes before PMR, at which time cells lost Fluo-4 signal and gained SiR-actin (Fig. 2c,d). In individual cells, calcium uptake oscillated before PMR (Extended Data Fig. 5a) perhaps reflecting GSDMD pore dynamics, and calcium uptake preceded filopodial assembly (Supplementary Video 6). *Gsdmd* deficiency blocked nigericin-induced calcium uptake (Fig. 2c,d and Supplementary Video 7). We next chelated extracellular $Ca^{2+}$ using EGTA; this ablated the assembly of filopodia without affecting cell lysis (Fig. 2e and Extended Data Fig. 5b,c). GSDMD pores ultimately induce the membrane perforating activity of NINJ1, which executes PMR. NINJ1 was dispensable for pyroptotic filopodial assembly (Fig. 2f,g). We next inducibly expressed the pore-forming fragment of GSDMD (GSDMD-NT)[36], which triggered macrophage lysis and the assembly of filopodia (Fig. 2h,i and Extended Data Fig. 5d); here, EGTA blocked filopodial assembly without affecting cell lysis. These data

collectively suggest that extracellular $Ca^{2+}$ enters cells through GSDMD pores independently of NINJ1-driven PMR, and this is both necessary and sufficient to activate the actin polymerization machinery, including the Myosin-X molecular motor protein that drives extension of the growing filopodial tip (Fig. 1d and Extended Data Fig. 1g).

Pyroptosis is one of many types of programmed lytic cell death, and is executed by members of the gasdermin family. To assess whether pore-induced filopodial assembly may be a general feature of gasdermin signaling, we expressed the pore-forming region of several gasdermins in primary macrophages (Fig. 3a). After verifying that cell priming did not affect GSDMD-NT-induced filopodia (Fig. 3b–d), we expressed GSDMA3-NT and GSDME-NT in unprimed macrophages; both induced filopodial assembly and cell lysis (Fig. 3e–g). To examine nonpyroptotic programmed cell lysis, we expressed the active form of the necroptosis effector, MLKL (MLKL 4-helix bundle, 4HB). MLKL-4HB induced substantial lysis, with necroptotic corpses crowned with filopodia (Fig. 3h–j). By contrast, tBID-induced apoptotic corpses (Fig. 3k,l) were morphologically distinct and did not possess filopodia. When pyroptosis is blocked by *Gsdmd* deficiency, CASP1 inflammasomes engage apoptosis[6,37,38], allowing us to directly compare pyroptosis versus apoptosis. Wild-type (WT) nigericin-stimulated macrophages were negative for cleaved CASP3, and decorated with pyroptosis-associated filopodia, whereas *Gsdmd^-/-* cells containing ASC specks were apoptotic, with shrunken cell bodies positive for cleaved caspase-3 and often showing retraction fibers (Fig. 3m and Extended Data Fig. 4a). While retraction fibers are morphologically distinct from filopodia, they were counted together with filopodia to minimize observer bias, with the conclusion that nigericin-induced apoptosis did not induce cell projections in *Gsdmd*-deficient macrophages (Fig. 2a). Similarly, apoptosis induced by inhibiting prosurvival and anti-apoptotic effectors did not trigger filopodial assembly (Fig. 3n). Hence, diverse effectors of programmed cell lysis generate cell corpses decorated with filopodia, whose presence distinguishes such corpses from apoptotic corpses.

To capture a live three-dimensional view of pyroptotic blistering and cell rupture during pyroptosis, we performed high-resolution lattice light-sheet imaging. This revealed that PMR occurs asymmetrically in adherent macrophages (Fig. 4a,b and Supplementary Videos 8 and 9), similar to suspended macrophages (Fig. 1e and Extended Data Fig. 2). Pyroptotic blisters (white arrowheads) emerged from one side of the cell and expanded. Other blisters ceased expanding once the dominant blisters reached critical mass. To resolve the corpse ultrastructure, we used scanning electron microscopy (SEM). Pyroptotic corpses often exhibited a cell body with regions partially lost during cell rupture, while the cell remnant was crowned by clearly defined projections that extend the corpse footprint (Fig. 4c,d and Extended Data Figs. 6 and 7). 'Budding' structures suggestive of extracellular vesicles were present along some filaments (Extended Data Figs. 6f and 7f). Phalloidin imaging confirmed that pyroptotic projections are rich in F-actin (Extended Data Figs. 6, 7 and 8a–d) and extremely stable, persisting for up to 24 h

**Fig. 4 | Asymmetric rupture preserves filopodia and corpse remnants.**
**a**, Lattice light-sheet imaging of macrophage expressing PLCδ-PH-GFP (green) and incubated with 1 μM SiR-actin (magenta). Primed macrophages were stimulated with nigericin for the time indicated in each panel. Cells in examples (i) and (ii) are displayed on a shifted axis (indicated in top right of example (i)) to highlight the 3D morphology of macrophages undergoing pyroptosis. Images were processed in Arivis 4D and represented as volume renderings. Each channel was nonlinearly adjusted (gamma, 2) to represent cell features across a wide dynamic range. **b**, (i) LLAMA segmentation[46] of the SiR-actin signal of lattice light-sheet imaging, analyzing nigericin-stimulated cells. Cell body (green), filopodia (magenta) and pyroptotic blisters (blue) were segmented by machine learning and are displayed as volumetric projections. **b**, (ii) Line scans (indicated in yellow on the rotated MIP panel) relative to (i) are displayed as orthogonal slice views to represent the spatial distribution of pyroptotic cell features. Orthogonal slices

were processed using Fiji. **c**,**d**, SEM of macrophage corpses produced by NLRP3- or CASP11-activating stimuli (nigericin, **c**; cytosolic LPS, **d**). Scale as indicated and magnified insets are to the right of each panel. **e**, Quantification of the cell and corpse area of WT LPS-primed BMDM stimulated with nigericin with or without EGTA. Phalloidin-labeled footprints of intact cells containing ASC specks (left), and cells that had undergone PMR and lost the bulk of cell mass (right) were hand-segmented in Fiji. Examples are included in Extended Data Fig. 5c. The area of thresholded phalloidin staining was quantified per cell, and analyzed by an unpaired two-sided Mann–Whitney test. **f**, Schematic view of F-actin remodeling to generate filopodia-rich corpses via GSDMD-dependent $Ca^{2+}$ influx. Scale bars, 10 μm except where indicated otherwise, and data are representative of three independent biological replicates. Violin plots show mean (solid line) and first and third quartiles (dotted lines). Statistical significance ****$P \leq 0.0001$.

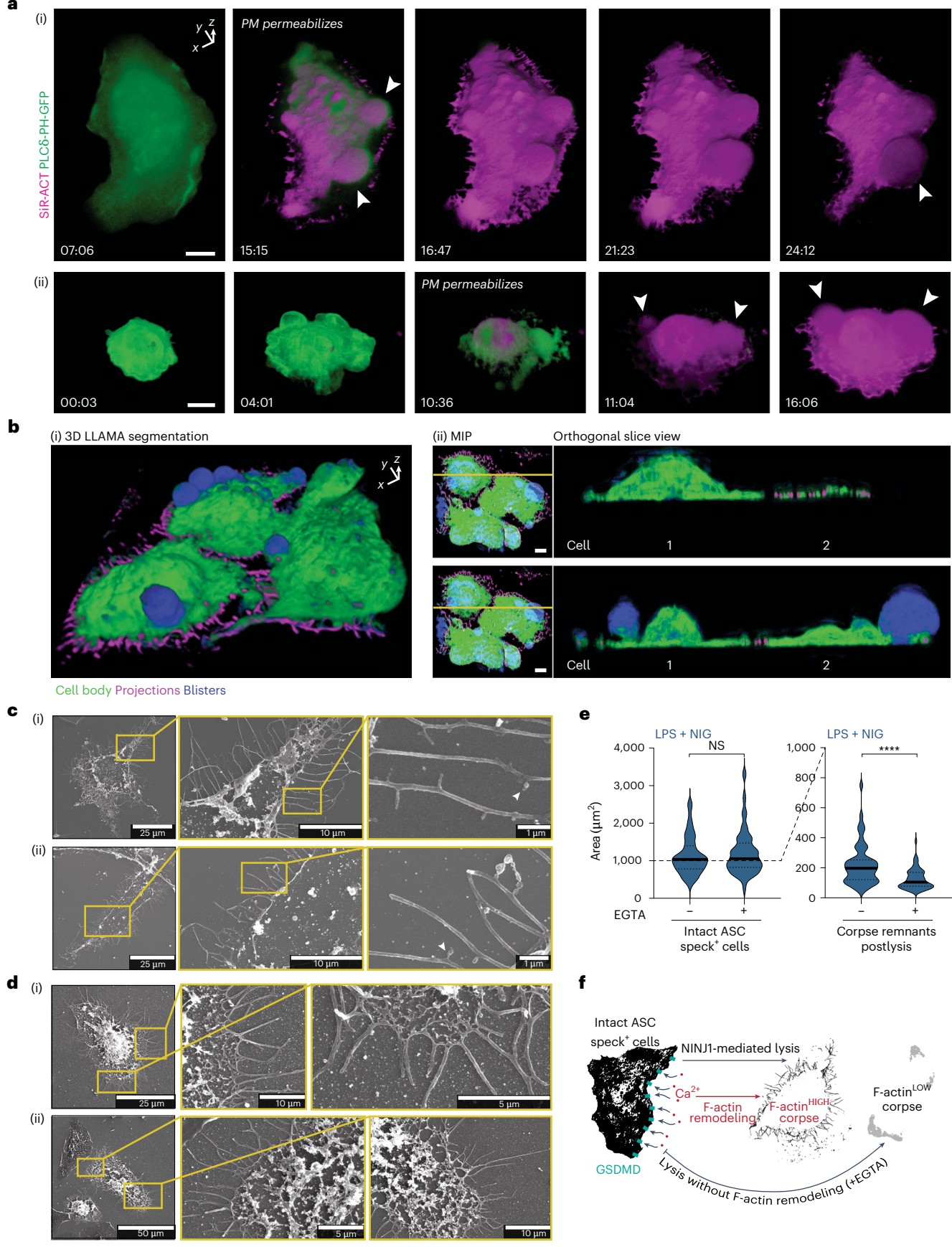

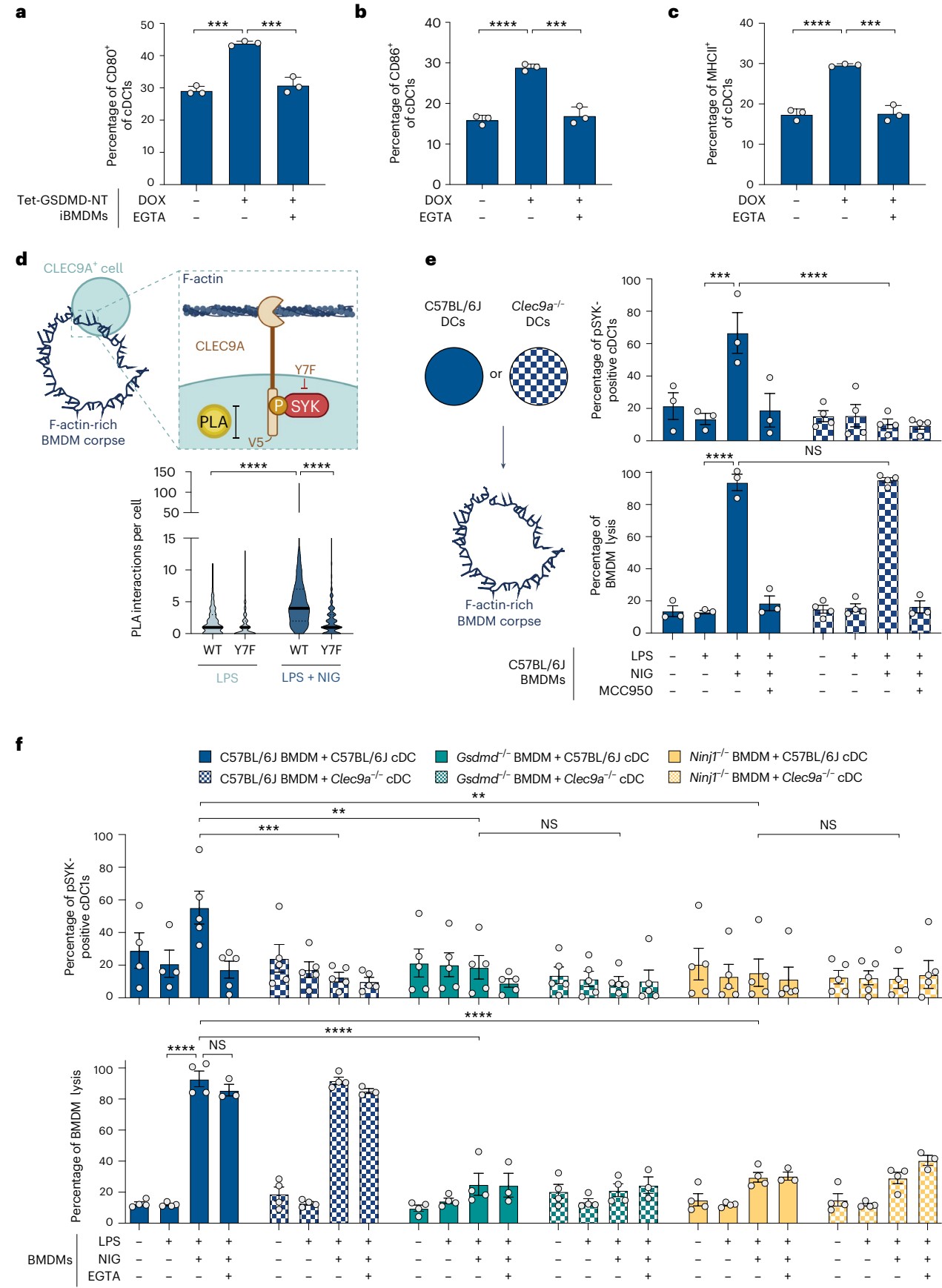

**Fig. 5 | F-actin-rich filopodia from pyroptotic corpses are sensed by CLEC9A, triggering signaling to SYK. a–c,** iBMDM stably expressing Tet-GSDMD-NT were exposed to doxycycline in the presence or absence of EGTA-AM, and incubated with murine dendritic cells. Cell surface expression of DC activation markers CD80 (**a**), CD86 (**b**) and MHCII (**c**) in the cDC1 population (CD11c⁺/CD24⁺/SIRPA⁻) was assessed by flow cytometry (*n* = 3 independent biological experiments). Data are mean ± s.e.m., were verified for normality and analyzed by unpaired two-sided *t*-test. **d,** Experimental method for quantifying phospho-SYK interactions with V5-C-terminally tagged CLEC9A by (PLA in RAW264.7 cells stably expressing murine CLEC9A (WT or Y7F signal-deficient mutant). LPS-primed macrophages, and nigericin-induced macrophage corpses, were incubated with CLEC9A-expressing RAW264.7 cells in the presence of 50 µM pervanadate. Quantification of PLA signal maxima for manually segmented RAW264.7 cells from Airyscan confocal images of macrophages (representative images shown in Extended Data Fig. 9b). Data were analyzed using a Kruskal–Wallis test with Dunn's multiple testing correction (*n* > 260 cells per condition pooled from three independent biological experiments). **e,** Unprimed or LPS-primed WT macrophages were left untreated, or exposed to nigericin with or without MCC950 (10 µM) added 1 h before nigericin. BMDM lysis was measured by LDH release (lower panel). WT or *Clec9a⁻/⁻* dendritic cells were then added to macrophage cultures, and the percentage of the cDC1 population (XCR1⁺/CD24⁺) that was positive for p-SYK was quantified by flow cytometry (upper panel). Data are mean ± s.e.m. and were analyzed by two-way ANOVA with Bonferroni correction (data points graphed are independent biological experiments). **f,** Unprimed or LPS-primed WT, *Gsdmd⁻/⁻* or *Ninj1⁻/⁻* macrophages were left untreated or exposed to nigericin with or without EGTA-AM (10 µM) added 1 h before nigericin. BMDM lysis was measured by LDH release (lower panel). WT or *Clec9a⁻/⁻* dendritic cells were then added to macrophage cultures, and the percentage of the cDC1 population (XCR1⁺/CD24⁺) that was positive for p-SYK was quantified by flow cytometry (upper panel). Data are mean ± s.e.m. and were analyzed by two-way ANOVA with Bonferroni correction (data points graphed are independent biological experiments). Statistical significance: **$P \leq 0.01$; ***$P \leq 0.001$; ****$P \leq 0.0001$.

(Extended Data Fig. 8e). Thus, pyroptotic PMR shatters part of the cell body, leaving behind a remnant corpse crowned by stable filopodia.

Given that filopodia can reinforce cell attachment to a substrate under conditions of cell adherence, we hypothesized pyroptotic filopodia may allow the cell to better withstand the elastic recoil force exerted during asymmetric PMR to preserve the cell corpse. We blocked filopodial assembly with EGTA, which markedly decreased the area of pyroptotic corpse remnants without affecting the area of intact cells yet to undergo PMR (Fig. 4e). Thus, pyroptosis proceeds strategically to induce and preserve filopodia, enabling cells undergoing pyroptotic rupture to generate stable caches of F-actin-rich deposits within the partially preserved corpse (Fig. 4f).

We next investigated the immunogenicity of pyroptotic corpses. Pyroptotic corpses triggered the activation of the cDC1 subset of dendritic cells (DCs), by upregulating CD80, CD86 and MHC class II expression (Fig. 5a–c and Extended Data Fig. 9a). By contrast, pyroptotic corpses generated with EGTA to block filopodial assembly (Fig. 2h,i) did not activate cDC1 cells (Fig. 5a–c). cDC1 cells selectively express the C-type lectin domain family 9 member A (CLEC9A). CLEC9A senses plasma membrane-associated F-actin[39], which it can only sample near membrane tears that expose the intracellular face of the plasma membrane to the extracellular space. CLEC9A is activated by cells that died of mechanical stress (for example, freeze–thaw[1]) or secondary necrosis[40] but its capacity to sense filopodia on corpses generated by programmed cell lysis was untested. CLEC9A ligation and coupled spleen tyrosine kinase (SYK) signaling[1,41] enables cDC1 cells to selectively process dead cell antigens for cross-presentation to T cells to engage the adaptive immune system[1].

To determine whether F-actin from pyroptotic corpses can ligate CLEC9A, we first established a system for quantifying phospho-SYK (p-SYK) interactions with CLEC9A. We stably expressed in RAW264.7 cells, WT murine CLEC9A tagged with V5 on the intracellular tail and also a CLEC9A Y7F point mutant that ablates CLEC9A:SYK interactions (Extended Data Fig. 9b–d). CLEC9A-V5-expressing cells were incubated with primed macrophages versus macrophage pyroptotic corpses, and proximity ligation assay (PLA) quantified interactions between CLEC9A-V5 and p-SYK. WT CLEC9A-V5-expressing cells showed substantially higher V5:p-SYK interactions when exposed to pyroptotic corpses as compared to intact living macrophages, and CLEC9A Y7F mutation ablated V5:p-SYK interaction (Fig. 5d and Extended Data Fig. 9b). Pyroptotic corpses thus preserve caches of F-actin to serve as a ligand for CLEC9A.

To confirm that filopodia on pyroptotic corpses trigger signaling by endogenous CLEC9A in incoming cDC1 cells, we generated macrophage pyroptotic corpses and incubated these with primary murine dendritic cells. cDC1s incubated with macrophage pyroptotic corpses showed elevated phosphorylated SYK compared to cDC1s incubated with primed macrophages, or nigericin-treated macrophages in which pyroptosis is blocked by MCC950 (Fig. 5e and Extended Data Fig. 9e). cDC1 SYK phosphorylation was ablated in *Clec9a*-deficient dendritic cells, verifying that corpses induced p-SYK downstream of CLEC9A in cDC1s (Fig. 5e). We then used this culture system to test the hypothesis that CLEC9A may only access the intracellular F-actin of GSDMD-induced filopodia after NINJ1-driven PMR and externalization of intracellular contents. Indeed, while NINJ1 was dispensable for filopodial assembly (Fig. 2f,g), NINJ1 was required for pyroptotic corpses to induce CLEC9A-driven p-SYK in cDC1s (Fig. 5f). Calcium chelation by EGTA-AM during pyroptosis blocked GSDMD-induced filopodial assembly (Fig. 2) as well as the capacity of cell corpses to trigger CLEC9A-driven p-SYK in cDC1s (Fig. 5f). Nigericin induces *Gsdmd⁻/⁻* macrophages to undergo inflammasome-driven apoptosis[6,37,38] (Fig. 3m), and these cells were unable to provoke CLEC9A-driven p-SYK in cDC1s (Fig. 5f). Thus, inflammasome-driven pyroptotic corpses, but not apoptotic cells, are immunogenic to cDC1s. Thus, the filopodial crowns of pyroptotic corpses engage incoming cDC1 cells via CLEC9A, and activate cDC1 antigen presentation pathways.

Microbes are potent stimulators of pyroptosis[20,30] and pyroptotic corpses capture microbial antigens[42]. CLEC9A binds to exposed F-actin on damaged cells, triggering their engulfment and the cross-presentation of corpse antigens to CD8⁺ T cells to induce antigen-specific immune responses[1,2,40]. CLEC9A also facilitates the conventional presentation of corpse antigens (for example, for T follicular helper cell-mediated humoral immunity)[2] and silences cDC1 microbe-induced inflammatory responses[2], thereby orchestrating the switch from innate to adaptive immunity[43,44]. Thus, filopodial crowns may serve to announce a corpse generated under suspicious circumstances that requires 'autopsy', such that corpse microbial antigens can stimulate adaptive immunity; this may provide a mechanism by which pyroptosis stimulates cDC1 cells to prime durable T cell-mediated antitumor immunity[45].

In conclusion, this study reports that inflammasome signaling induces the coordinated assembly of F-actin-rich filopodia. These projections erupt following Ca²⁺ influx through the GSDMD pore, independently of cell rupture. Filopodia persist after cell rupture, where they flag the pyroptotic corpse for recognition and likely antigen sampling by CLEC9A-bearing cDC1 cells. In orchestrating CLEC9A activation by pyroptotic corpses, inflammasomes may thus coordinate the transition from innate to antigen-specific immune responses during infection, vaccination, cancer and autoimmunity.

## Online content

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

[1]Institute for Molecular Bioscience, University of Queensland, Brisbane, Queensland, Australia. [2]Max Planck Institute for Infection Biology, Berlin, Germany. [3]Division of Gastroenterology, Boston Children's Hospital and Harvard Medical School, Boston, MA, USA. [4]Australian Institute for Bioengineering and Nanotechnology, University of Queensland, Brisbane, Queensland, Australia. [5]Faculty of Science, University of Queensland, Brisbane, Queensland, Australia. [6]Immunity Program, Monash Biomedicine Discovery Institute and Department of Biochemistry and Molecular Biology, Monash University, Clayton, Victoria, Australia. [7]These authors jointly supervised this work: Sabrina S. Burgener, Kate Schroder. ✉e-mail: m.monteleone@imb.uq.edu.au; K.Schroder@imb.uq.edu.au

## Methods

### Mice and tissue harvest

All experimental protocols involving mice were approved by the University of Queensland Animal Ethics Committee. WT (C57BL/6J, or littermate controls as appropriate), $Casp1^{C284A/C284A}$ (B6J-Casp1C284Aem1Ksc CR; generated in-house[32]), $Casp11^{-/-}$ (B6.Casp4tm; backcrossed to C57BL/6J)[47], $Gsdmd^{-/-}$ (C57BL/6N-Gasdmdem1Vmd; backcrossed to C57BL/6J)[6], $Ninj1^{-/-}$ (C57BL/6N-Ninj1-ENU-KO)[8] and Clec9a-GFP knock-in (Jackson, B6(Cg)-$Clec9a^{tm1.1Crs}$/J)[1] mice were housed in specific pathogen-free conditions at the University of Queensland. Six to 20 week-old male and female animals were sex- and aged-matched for each experiment.

### Molecular cloning, plasmid constructs and mRNAs

The pMSCV-PLCδ-PH-GFP construct[10] was expressed in primary bone marrow cells by retroviral transduction[23]. The coding sequences of pEF6-mCLEC9A(WT)-V5 and pEF6-mCLEC9A(Y7F)-V5 were synthesized by Integrated DNA Technologies and inserted into a modified pEF6 backbone. All plasmid open reading frame sequences are supplied in the Supplementary Data.

Cell death effector-encoding messenger RNAs (mRNAs), corresponding to the constitutively active portion of murine GSDMD (residues 1–276), GSDME (residues 1–270), GSDMA3 (residues 1–262), tBID (residues 60–195) and MLKL (residues 1–125, 4HB domain) were synthesized and in vitro transcribed with 5-methoxyuridine modification by the University of Queensland BASE Facility. All mRNAs were validated as endotoxin free. All mRNA open reading frame sequences are supplied in the Supplementary Data.

### Cell culture

All cell culture work was conducted under sterile conditions in class II biosafety cabinets and cells were incubated at 37 °C with 5% $CO_2$. Cell lines were routinely tested for mycoplasma contamination and maintained mycoplasma-free. The murine macrophage cell line RAW264.7 (American Type Culture Collection TIB-71) was maintained in full RAW264.7 media (Roswell Park Memorial Institute, RPMI, 1640 medium (cat. no. 21870092, Gibco), 5% endotoxin-free fetal calf serum (FCS; Gibco), 10 U ml$^{-1}$ penicillin–streptomycin (cat. no. 15070063, Gibco), 2 mM GlutaMAX (cat. no. 35050061, Gibco)). The retroviral packaging Platinum-E (Plat-E) cell line[48] was maintained in full Plat-E media (Dulbecco's Modified Eagle Medium (DMEM; cat. no. 11995073, Gibco), 10% FCS, 10 U ml$^{-1}$ penicillin–streptomycin, 10 µg ml$^{-1}$ blasticidin and 1 µg ml$^{-1}$ puromycin).

Primary murine bone marrow-derived macrophages (BMDMs) were differentiated from bone marrow progenitors[49] and used for experiments on day 7 or 8 of culture. Primary macrophages were maintained in full BMDM media (RPMI, 10% FCS, 10 U ml$^{-1}$ penicillin–streptomycin, 2 mM GlutaMAX, 150 ng ml$^{-1}$ recombinant human colony-stimulating factor-1 (CSF-1; endotoxin free, expressed and purified by the University of Queensland Protein Expression Facility)). cDC1s were produced from fresh bone marrow progenitors differentiated in full cDC1 media (Iscove's Modified Dulbecco's Medium (IMDM) medium (cat. no. 12440053, Gibco), 10% FCS, 10 U ml$^{-1}$ penicillin–streptomycin, 2 mM GlutaMAX, 10 mM HEPES (cat. no. 15630080, Gibco), 1 mM sodium pyruvate (cat. no. 11360-070, Gibco), adjusted to 308 mOsm and supplemented with 200 ng ml$^{-1}$ recombinant mouse Flt3-ligand (Flt3L; 250-31L, Peprotech, or 130-097-372, Miltenyi Biotech)) and used for experiments on day 8 of culture.

Human monocyte-derived macrophages (HMDM) were produced from buffy coats from blood donations to the Australian Red Cross Blood Service from anonymous, informed and consenting adults. Human peripheral blood mononuclear cells were isolated from screened buffy coats by density centrifugation with Ficoll-Plaque Plus (GE Healthcare), and monocytes were isolated by magnetic-assisted cell sorting[49]. Monocyte-derived macrophages were generated by differentiation over 7 days with 150 ng ml$^{-1}$ recombinant human CSF-1 at 37 °C with 5% $CO_2$. HMDMs were used for experimentation on day 7 of their differentiation.

Immortalized HBECs were maintained in complete keratinocyte media (Gibco, cat. no. 17005042, supplemented with the provided bovine pituitary enzyme and epidermal growth factor) and passaged at 70–80% confluency. Then $5 \times 10^4$ cells were seeded onto glass coverslips in complete keratinocyte media 24 h before CASP4 activation assays.

### Inflammasome and apoptosis stimulation

Murine or human macrophages were plated at $0.1 \times 10^6$ cells per 0.5 ml in full media in 24-well cell culture-treated plates containing a 2.5 cm (1 inch) coverslip (for microscopy) or $0.1 \times 10^6$ cells per 0.1 ml full media in 96-well cell culture-treated plates (for lactate dehydrogenase (LDH) assay). To activate the CASP11 inflammasome, cells were primed with 1 µg ml$^{-1}$ of PAM3CSK4 (tlrl-pms, InvivoGen) for 16 h. Media was then removed and replaced with Opti-MEM (cat. no. 31985062, Life Technologies) supplemented with 150 ng ml$^{-1}$ CSF-1. Next, 2 µg ml$^{-1}$ of ultrapure *Escherichia coli* K12 LPS (tlrl-peklps, InvivoGen) was prepared for cytosolic delivery with FuGENE HD (cat. no. E2311, Promega) and the FuGENE HD/LPS complex was delivered to the cells by centrifugation at 600$g$ for 10 min at room temperature. Cells were then incubated for a further 2 h or as otherwise stated. To activate NLRP3, full media was removed and macrophages were primed with 100 ng ml$^{-1}$ of ultrapure *E. coli* K12 LPS in Opti-MEM supplemented with 150 ng ml$^{-1}$ CSF-1 for 4 h, before exposure to 5 µM (BMDM) or 10 µM (HMDM, bone marrow-derived dendritic cells (BMDCs)) nigericin (N7143, Sigma Aldrich) for a further 30–45 min (or as indicated). The NAIP-NLRC4 inflammasome was activated using the protective antigen-Tox system that delivers proteins to the cytosol by N-terminal fusion to the *Bacillus anthracis* lethal factor (LFn), and transmembrane transport through the anthrax protective antigen channel[50,51]. For murine cells, we used a recombinant fusion protein of LFn with *Legionella pneumophila* flagellin (Fla1). For activation of human NAIP-NLRC4, we used a fusion of LFn and *Salmonella typhimurium* PrgI. Cells were stimulated with 250 ng ml$^{-1}$ recombinant PA plus 125 ng ml$^{-1}$ Fla1-LFn or 20 ng ml$^{-1}$ PrgI-LFn in Opti-MEM supplemented with 150 ng ml$^{-1}$ CSF-1 for 1 h. For sublytic inflammasome signaling, BMDM or BMDC were primed with LPS as before, and stimulated with 100 µg ml$^{-1}$ PGPC (10044, Cayman; freshly reconstituted[13]). For induction of apoptosis, BMDMs were left unprimed and intrinsic apoptosis was induced with ABT-737 and S63845 (cat. nos. 11501 and 21131, Cayman Chemical, both used at 500 nM) for 2 h.

To activate the CASP4 inflammasome in HBECs, cells were primed with 1 µg ml$^{-1}$ of PAM3CSK4 for 16 h and transfected with LPS as above, using the transfection reagent Lipofectamine LTX with PLUS reagent (cat. no. 15338100, Invitrogen). Cells were fixed 5.5 h after transfection for confocal imaging.

### SEM

SEM was performed on cells plated on 35-mm dishes with inbuilt high grid-500 1.5' coverslips (cat. no. 81168, Ibidi). Following inflammasome stimulation, cells were fixed in 4% paraformaldehyde (PFA) (C004, ProSciTech) and 0.25% EM-grade glutaraldehyde (C002, ProSciTech) for 20 min. Cells were stained with Alexa Fluor 488 Phalloidin (6.6 µM, cat. no. A12379, Invitrogen) for 15 min and cells of interest were mapped in PBS using a Zeiss AxioObserver widefield microscope and ×40 (numerical aperture (NA) 1.3) or ×63 (NA 1.4) oil objectives using Zen Blue software. Cells were washed in PBS and refixed in 2.5% glutaraldehyde (C002, ProSciTech) in PBS in a BioWave microwave (Ted Pella Inc.) at 80 W for 3 min under vacuum. Dishes were subsequently washed in PBS and water then postfixed in 1% aqueous Osmium tetroxide (OsO4) at 80 W cycling for 6 min under vacuum. Fresh OsO4 solution was added, and cells were postfixed again in the BioWave. Dishes were washed twice in ddH$_2$O and then serially dehydrated in increasing percentages of ethanol (30, 50, 70, 90, 100, 100%). Coverslips were separated

from the dishes while submerged in 100% ethanol and transferred to a Tousimis Autosamdri-815 for critical point drying. Dry coverslips were mounted onto aluminum SEM stubs and carbon coated under vacuum in a Safematic CCU-010 Carbon coater. Stubs were loaded into a field emission FEI Scios Focus Ion Beam Scanning Electron Microscope, and regions of interest were correlated and imaged with secondary electron detection.

## Fluorescence microscopy

Confocal imaging was performed on cells fixed in 4% methanol-free formaldehyde (cat. no. 28906; Pierce, or 50-980-487; EMS) for 15–20 min at room temperature and stored at 4 °C in PBS overnight to quench remaining PFA until required, or quenched with 50 mM NH4Cl if processed immediately. All steps were carried out at room temperature, in the dark. Cells were labeled with 4,6-diamidino-2-phenylindole (DAPI) (0.1 µg ml$^{-1}$, D9542, Sigma Aldrich) and Phalloidin-iFluor 405 (1,000×, cat. no. AB176752, Abcam), Alexa Fluor (AF) 488 Phalloidin (6.6 µM, cat. no. A12379, Invitrogen) or AF647 Phalloidin (13.2 µM, cat. no. A22287, Invitrogen). Antibodies used for immunolabelling were anti-ASC (1:200, N15; Santa-Cruz; 1:800, cat. no. 67824S, Cell Signaling Technology), anti-MYO10 (1:100, cat. no. AB224120, Abcam), antiphospo-SYK (1:100, cat. no. 2711S, Cell Signaling Technology), anticleaved CASP3 (1:100, cat. no. 9664S, Cell Signaling Technology) and anti-V5 (1:200, cat. no. AB27671, Abcam). Secondary antibodies used were goat anti-rabbit AF594 (cat. no. A32740, Invitrogen), donkey anti-mouse AF488 (cat. no. A21202, Invitrogen) and goat anti-rabbit AF647 (cat. no. A32733, Invitrogen). Cells were mounted onto slides using ProLong Gold antifade mounting media (cat. no. P36934, Invitrogen) and sealed with clear nail polish. Images were acquired on a Zeiss Axiovert 200 microscope stand fitted with a LSM 880 confocal scanner running Zeiss Zen Black software. The microscope was equipped with 405, argon ion, 561 and 633 nm lasers. Plan Apochromat ×40 (NA 1.3) or ×63 (NA 1.4) oil immersion objectives were used and where indicated, the 4Y-Fast Airyscan mode was used.

For analyses of filopodia formation in PGPC-treated cells, all steps were carried out at room temperature, in the dark and in PermQuench buffer (0.2% w/v bovine serum albumin and 0.02% w/v saponin in PBS). First, the specimens were permeabilized for 30 min and then stained with rabbit anti-ASC (1:100, cat. no. AG-25B-0006-C100, Adipogen) or anti-MYO10 (1:100, cat. no. AB224120, Abcam) for 1 h. Secondary anti-rabbit-AF488 (cat. no. R37116, Thermo Fisher Scientific) was used, and 1 µg ml$^{-1}$ DAPI (cat. no. D1306, Thermo Fisher Scientific) and 1:400 AF647 Phalloidin (cat. no. A22287, Thermo Fisher Scientific). Specimens were washed and mounted on slides using ProLong Glass Antifade Mountant (cat. no. P36982, Thermo Fisher Scientific) and left to cure overnight. Imaging was performed using a LSM 880 scanning confocal microscope with fast Airyscan (4Y) controlled with Zeiss Zen Black software (Zeiss Instruments) and equipped with incident light fluorescence and laser illumination (405, 458, 488, 514, 561 and 633 nm), two GaAsP, two standard PMT NDD detectors and an Airyscan detector (32 GaAsP array). Imaging used a Plan Apochromat ×40/1.3 or a Plan Apochromat ×63/1.4 oil immersion objective (Zeiss Instruments). In a typical experiment, 9–25 fields of view were imaged using multi-position acquisition. Representative images were acquired using the Airyscan detector. Acquired microscopy data were processed using ImageJ/Fiji and filopodia were manually counted in a minimum of 40 cells per condition from randomly chosen fields of view.

Live fast Airyscan confocal microscopy imaged PLCδ-PH-GFP-expressing macrophages stained with SiR-actin[23] incubated at 37 °C with 5% CO2.

For analysis of calcium oscillations, WT and *Gsdmd*$^{-/-}$ BMDMs were seeded in four-chamber 35-mm glass-bottom dishes (cat. no. D35C4-20-1.5-N, Cellvis) in CSF-1-supplemented phenol red-free Opti-MEM (cat. no. 11058021, Gibco), and primed for 3 h with LPS. To activate NLRP3, 5 µM nigericin was added together with the dyes SiR-Actin and

the calcium biosensor Fluo-4 (cat. no. F10489, Invitrogen). Imaging commenced immediately.

For both fixed and live confocal microscopy, raw images were deconvolved over ten rounds of iterative deconvolution using Microvolution[52] running in Fiji (Microvolution LLC). Images were further subjected to background subtraction and additionally, for quantification, a 2-pixel median filter. Images are represented as maximum intensity projections (MIPs) of Z-stacks, unless indicated otherwise in the figure legend. The number of projections per cell was quantified by manual counting by a blinded operator. Quantification of cell size in EGTA-treated BMDM was performed in Fiji by using the AutoThreshold plugin with the default method to create a mask of each cell. The surface area of each masked cell, or masked cell corpse fragment larger than 10 pixels (approximately 2 µm$^2$), was then measured with the Analyse Particles function.

Lattice light-sheet imaged SiR-actin-stained primary macrophages expressing PLCδ-PH-GFP[23]. Coverslips (1.5′, 5-mm, cat. no. G400-05, ProSciTech) were placed in the 3i LLSM (v.2) 37 °C sample bath containing phenol red-free Opti-MEM and 5 µM nigericin. A selected field of view containing several PLCδ-PH-GFP-positive cells was imaged. A multi-Bessel beam optical lattice pattern was generated through a spatial light modulator and higher orders filtered through an annular mask to generate a noncontinuous light sheet that was dithered across the x axis. The sample was imaged by moving the sample stage in 0.495 µM increments across a region of ~75 × 50 × 100 µm using a 30-ms exposure time for each channel. The spatial light modulator was set to a sheet length of ~20 µM, 44 and 36 Bessel beams were used for the 488 and 642-nm channels, respectively. The inner and outer numerical aperture was 0.493 and 0.550 of the annular masks. The lattice light sheets were generated using the following settings: the spacing factor between each beam was 0.932 µm (488 nm) and 1.233 µm (642 nm), with a cropping factor of 0.175 (488 nm) and 0.190 (642 nm). Then, 161 Z slices were acquired at an angle of 32.8° relative to the sample and recorded using dual Hamamatsu Orca Flash 4.0 cameras. The composite three-dimensional (3D) images each took 12 s to acquire, inclusive of the set delay between sweeps and time taken to alternate between the 488 and 642-nm laser at each slice. Images were deconvolved and deskewed using Microvolution software before being exported to Arivis Vision4D (v.4.1.1) and Fiji/ImageJ (v.1.52k-1.54 f) for image processing. All acquisitions in Fig. 4a,b were subject to nonlinear adjustment (gamma v.2.0).

## Imaging macrophages in suspension

BMDMs expressing PLCδ-PH-GFP were imaged in suspension by plating cells onto passivated glass-bottom dishes to prevent the cell from forming stable adhesions to glass. To PEG-passivated glass surfaces, four-chamber glass-bottom dishes (D35C4-20-1.5-N, Cellvis) were treated with poly-L-lysine (PLL, 20 kDa)-grafted with PEG (2 kDa) (PLL(20)-g[3.5]-PEG(2)) (SuSoS Surface Technology) resuspended in PBS at a final concentration of 1 mg ml$^{-1}$. Glass-bottom dishes were passivated by overnight incubation at 37 °C and 5% CO2. The following day, the PLL-g-PEG solution was aspirated and 450 µl of suspension BMDM PLCδ-PH-GFP cells were seeded into each well with media supplemented with LPS and CSF-1 to prime for 4 h. Before imaging, the cell suspension in each well was pipetted up and down several times to encourage single cell suspension. To trigger pyroptosis, 50 µl of media containing LPS, CSF-1, SiR-actin and nigericin (5 µM) or vehicle was added to each well and imaging was immediately commenced. Imaging was performed on an inverted Andor Dragonfly 500 Spinning Disc Confocal microscope using a ×40 LWD Apochromat 1.15 NA water dipping objective and GenTeal Gel (Alcon) as an immersion medium, with air surrounding the objective sealed off to prevent the immersion medium from evaporating. To capture cells in suspension, temporal Z-stacks were acquired with an Andor Zyla 4.2 sCMOS camera over 45 min at intervals of 1–2 min within the Andor Fusion software. 3D cell surface renders were generated using Chimera X[53].

## Cell death assays

Cell death was assayed by measuring LDH release, using the Cytotox96 nonradioactive cytotoxicity assay (Promega) or CyQUANT LDH Cytotoxicity Assays (cat. no. C20301, Thermo Fisher Scientific) and graphed as a percentage of total cell lysis (control cells treated with 0.1% Triton X-100 (cat. no. T9284, Sigma Aldrich) in Opti-MEM for 5 min to induce 100% cell lysis). Absorbance at 490 nm was measured using the Powerwave XS spectrophotometer (BioTek) or the Infinite PRO 200 plate reader (TECAN).

## Immunoblotting

Cell extracts were collected in boiling western lysis buffer (66 mm Tris-HCl pH 7.4, 2% SDS), and analyzed by SDS–PAGE and immunoblot[25]. Primary antibodies used were: α-v5 monoclonal clone Sv5-PK1 at 1:1,000 (cat. no. ab27671, AbD Serotec), α-GSDMD at 1:1,000 (cat. no. ab209845, Abcam) and glyceraldehyde 3-phosphate dehydrogenase polyclonal at 1:5,000 (cat. no. 2275-PC, R&D Systems). Secondary HRP-conjugated antibodies used were anti-rabbit IgG and anti-mouse IgG both at 1:5,000 (cat. no. 7074, 7076, Cell Signaling Technology). Secondary antibodies on membranes were inactivated by incubation with 30% hydrogen peroxide before reprobe. Membranes were visualized using a Fusion imaging system (Vilber).

## Cell death induction by mRNA transfection

mRNA transfections were performed on days 6–7 of BMDM differentiation using the Neon transfection kit (cat. no. MPK1025R, Thermo Fisher). Briefly, cells were rinsed and lifted in PBS and concentrated to a cell density of $1.5 \times 10^6$ cells per 10 µl in Buffer R immediately before electroporation of 1 µg of GSDMD mRNA and molar ratio concentrations of the following death effectors: GSDME, GSDMA3, tBID and MLKL. Electroporator settings were 1,400 V, 20 ms$^{-1}$, two pulses, using a 10-µl tip. After delivering the electric pulse, cells were transferred to 1.5-ml tubes containing prewarmed assay media (Opti-MEM + CSF-1) and plated at the desired cell density for imaging and LDH release quantification. Cells were allowed to recover for 1 h before LPS priming, and samples were collected over a time course of up to 24 h for analyses.

## CLEC9A activation

CLEC9A recognition of F-actin-rich pyroptotic corpses was first assayed by adding CLEC9A-expressing RAW264.7 cells to pyroptotic BMDMs. RAW264.7 cells stably expressing murine CLEC9A-V5 (WT or Y7F signal-deficient mutant) were cultured to 50% confluency the day before experimentation. Primary macrophages were stimulated to induce pyroptosis in 24-well plates with coverslips and washed three times in sterile Dulbecco's PBS (cat. no. 14190144, Gibco) to remove inflammasome agonists, media and soluble cell and corpse components. A 30 mM pervanadate stock solution was prepared by diluting sodium orthovanadate in sterile Dulbecco's PBS, followed by the addition of 30% $H_2O_2$ (cat. no. HA154-500M, Chem Supply Australia) to a final concentration of 0.18% (w/w) and incubated for 15 min at room temperature. $0.2 \times 10^6$ RAW264.7 cells in Opti-MEM containing 50 µM pervanadate were added to each well containing macrophages, centrifuged (500$g$, 5 min), and incubated for 30 min. p-SYK/mCLEC9A-V5 interactions were assayed using PLA (DUO92004, DUO92002 and DUO92008, Duolink In Situ PLA, Sigma Aldrich) according to the kit instructions, omitting the antibody preincubation step to reduce nonspecific background. Samples were imaged using confocal microscopy. RAW264.7 cells were manually segmented and subjected to background subtraction, median filter and MIP. Data are represented as the number of PLA puncta per cell.

CLEC9A recognition of F-actin-rich pyroptotic corpses was also assayed by culturing murine dendritic cells with pyroptotic corpses. WT, $Gsdmd^{-/-}$, $Ninj1^{-/-}$ macrophages were plated at $0.25 \times 10^6$ cells per 0.5 ml in full BMDM media in 24-well cell culture-treated plates, and primed and stimulated as described above, with and without 1 h of

pretreatment with 10 µM EGTA-AM before nigericin treatment for 45 min. Macrophage media was aspirated and assayed for LDH release, while adherent cells and corpse remnants were washed twice with prewarmed PBS in the plate. Next, $0.75 \times 10^6$ Flt3L-derived BMDCs (WT versus $Clec9a^{-/-}$) were added to each well in 250 µl Opti-MEM containing 1 mM pervanadate, and incubated for 30 min before flow cytometric analyses.

## Dendritic cell activation

For analyzing cDC1 dendritic cell activation by pyroptotic corpses, immortalized BMDMs (iBMDMs) stably expressing Tet-GSDMD-NT were plated in 96-well plates at $5 \times 10^4$ cells per 0.1 ml of complete DMEM. The following day, cells were treated with 1 µg ml$^{-1}$ doxycycline and 500 ng ml$^{-1}$ LPS in the presence or absence of 10 µM EGTA-AM. Then 8 h posttreatment, supernatants were removed and $3–5 \times 10^4$ of Flt3L-BMDCs were added to each well in 30 µl complete IMDM. After 2 h of incubation, the BMDCs in suspension were transferred to 96-well plates, incubated for an additional 3 h and then analyzed by flow cytometry.

## Flow cytometry

For analyzing p-SYK signaling in cDC1 dendritic cells, WT or $Clec9a^{-/-}$ Flt3L-derived BMDCs were washed with PBS and immediately fixed with 1% PFA for 20 min at room temperature. Cells were centrifuged (500$g$ for 5 min) and resuspended in prechilled 300 µl of True-Phos Perm Buffer (cat. no. 425401, Biolegend) and incubated at −20 °C for 45 min. Cells were washed with PBS + 1% FBS and pelleted (2,000$g$ for 5 min). Flt3L-BMDCs were stained for 45 min at 4 °C in PBS with 1% FBS containing the following fluorescently conjugated antibodies and dyes: PE anti-mouse p-SYK (clone l120-722, BD Biosciences, cat. no. 558529), BV711 anti-mouse CD24 (clone M1/69, BD Biosciences, cat. no. 563450), APC anti-mouse XCR1 (clone ZET, Biolegend, cat. no. 148206) and BUV395 anti-mouse CD80 (clone 16-10A1, BD Biosciences, cat. no. 740246). Data were acquired on a BD LSRFortessa X-20 (BD Biosciences) and analyzed using FlowJo v.10 software.

For analyzing cDC1 dendritic cell activation by pyroptotic corpses, Flt3L-BMDCs were washed and stained for 20 min at 4 °C in MACS buffer (PBS with 1% FBS, 2 mM EDTA) containing the following fluorescently conjugated antibodies and dyes: FITC anti-mouse CD11c (clone N418, Biolegend, cat. no. 117306), PerCP-eFluor 710 anti-mouse SIRP alpha (clone P84, Invitrogen, cat. no. 46-1721-82), PE/Cyanine7 anti-mouse CD24 (clone M1/69, Biolegend, cat. no. 101822), PE anti-mouse CD86 (clone GL-1, Biolegend, cat. no. 105008), BV711 anti-mouse CD80 (clone 16-10A1, Biolegend, cat. no. 104743), APC anti-mouse I-A/I-E (MHCII) (clone M5/114.15.2, Biolegend, cat. no. 107614), and Live/Dead Fixable Violet Dead (ThermoFisher, cat. no. L34964). Data were acquired on a FACSymphony A5 SE (BD Biosciences) and analyzed using FlowJo.

## In vivo LPS challenge

WT or $Gsdmd^{-/-}$ mice (females at 8–10 weeks old, 4–5 mice per group) were injected with 200 µl of 10% thioglycollate broth (108190, Sigma Aldrich) into the peritoneal cavity. Three days later, mice were intraperitoneal challenged with either PBS or with 200 µl of 1 mg ml$^{-1}$ LPS (L8643, LPS *Pseudomonas aeruginosa*, Sigma Aldrich). After 4 h, mice were humanely euthanized using $CO_2$. Peritoneal lavage was performed using 5 ml of sterile PBS, and pelleted by centrifugation at 500$g$ for 5 min. Cell pellets were resuspended, counted and equal amounts of cells were centrifuged onto poly-L-lysine-coated (P4707, Sigma Aldrich) coverslips and fixed in 4% PFA for imaging.

## Statistical analysis

Statistical analyses were performed in GraphPad Prism, and are detailed in each figure legend. Violin plots show quartiles (dotted lines) and mean (solid line). Not significant (NS), $P > 0.05$; *$P \le 0.05$; **$P \le 0.01$; ***$P \le 0.001$ and ****$P \le 0.0001$.

**Reporting summary**

Further information on research design is available in the Nature Portfolio Reporting Summary linked to this article.

## Data availability

All data reported in this paper will be shared by the corresponding author upon reasonable request. Source data are provided with this paper.

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

## Acknowledgements

This work was supported by the Australian Research Council (grant nos. Discovery Project DP230100300 to C.L.H. and K.S.; DP190102230 and FL230100100 to A.S.Y.; FT190100516 to S.J.S.; Discovery Early Career Researcher Fellowships DE220100823 and DE200101300 to S.E. and M.M.), the University of Queensland (Research Support Fellowship for C.L.H.), the Yulgilbar Alzheimer's Foundation (PhD stipend for C.L.H.), NIH grants (grant nos. AI167993, AI11655 and DK34854 to J.C.K.) and the National Health and Medical Research Council of Australia (grant nos. GNT113659 to A.S.Y., Fellowship 2009075 to K.S.). D.F. was supported by a Human Frontier Science Program (HFSP) long-term postdoctoral fellowship (grant no. LT0006/2022-L), and an EMBO postdoctoral fellowship (grant no. ALTF 491-2022). S.S.B. was supported by Novartis Foundation for Medical-Biological Research Fellowship. N.D.C. is supported as a CZI Imaging Scientist by grant number 2020-225648 from the Chan Zuckerberg Initiative DAF, an advised fund of Silicon Valley Community Foundation. Cell imaging was performed at Institute for Molecular Bioscience in the Cancer Ultrastructure and Function Facility funded by the Australian Cancer Research Foundation (ACRF). The SEM was performed at the Centre for Microscopy and Microanalysis, University of Queensland. We acknowledge the facilities, and the scientific and technical assistance, of Microscopy Australia at the Centre for Microscopy and Microanalysis, and the BASE mRNA Facility, both at the University of Queensland. BASE is supported by Therapeutic Innovation Australia (TIA). TIA is supported by the Australian Government through the National Collaborative Research Infrastructure Strategy (NCRIS) program. We thank Genentech and the Australian National University for sharing their *Gsdmd*[−/−] and *Ninj1*[−/−] mice, J. Murphy for guidance on designing the MLKL-4HB mRNA construct and F. Wylie for critical reading of early manuscript drafts.

## Author contributions

C.L.H. and K.S. conceived the study and acquired project funding. C.L.H. designed and performed experiments, analyzed data, prepared figures for the first submission and wrote the first manuscript draft. C.L.H. and N.D.C. performed and analyzed lattice light-sheet imaging, with input from J.G.L. and N.H. for LLAMA segmentation analysis. M.M. and S.S.B. designed, performed and analyzed experiments for manuscript revisions, and managed collaborations (M.M. managed A.E.S.L., R.J.J., S.J.S. and N.A.; S.S.B. managed D.F., J.H.C. and J.C.K.). D.F. performed and analyzed experiments with sublytic inflammasome stimuli and iBMDM-GSDMD-NT cells, with input from J.H.C., J.C.K., S.S.B. and K.S. A.E.S.L. performed and analyzed Fluo-4 cell imaging with input from R.J.J., S.J.S., M.M. and K.S. R.J.J. performed and analyzed experiments with suspended macrophages, with input from A.E.S.L., S.J.S. and M.M. J.H.C. performed and analyzed experiments assessing dendritic cell activation by corpses, with input from D.F., J.C.K., S.S.B. and K.S. S.E., J.C., G.M.E.P.L., J.R.C. and R.B. performed experiments, analyzed data, designed experimental tools (for example mRNAs) and gave technical support and advice. M.H.L. and A.S.Y. provided reagents and technical advice. N.A. performed and analyzed electron microscopy. S.S.B. managed mouse ethics and colonies. K.S. supervised the project, prepared figures and wrote the final manuscript with input from all authors. All authors discussed experimental results and gave input into the manuscript.

## Competing interests

K.S., M.H.L. and J.C.K. declare the following competing interests: K.S. is a coinventor on patent applications for NLRP3 inhibitors that have been licensed to Inflazome Ltd. K.S. served on the Scientific Advisory Board of Inflazome (2016–2017) and Quench Bio, USA (2018–2021) and serves on a Scientific Advisory Board for Novartis, Switzerland (since 2020). M.H.L. is an inventor on patents relating to CLEC9A antibodies. J.C.K. consults and holds equity in Corner Therapeutics, Larkspur Biosciences, MindImmune Therapeutics and Neumora Therapeutics; none of these relationships affected this study. The other authors declare no competing interests.

## Additional information

**Extended data** is available for this paper at https://doi.org/10.1038/s41590-024-02024-3.

**Correspondence and requests for materials** should be addressed to Mercedes Monteleone or Kate Schroder.

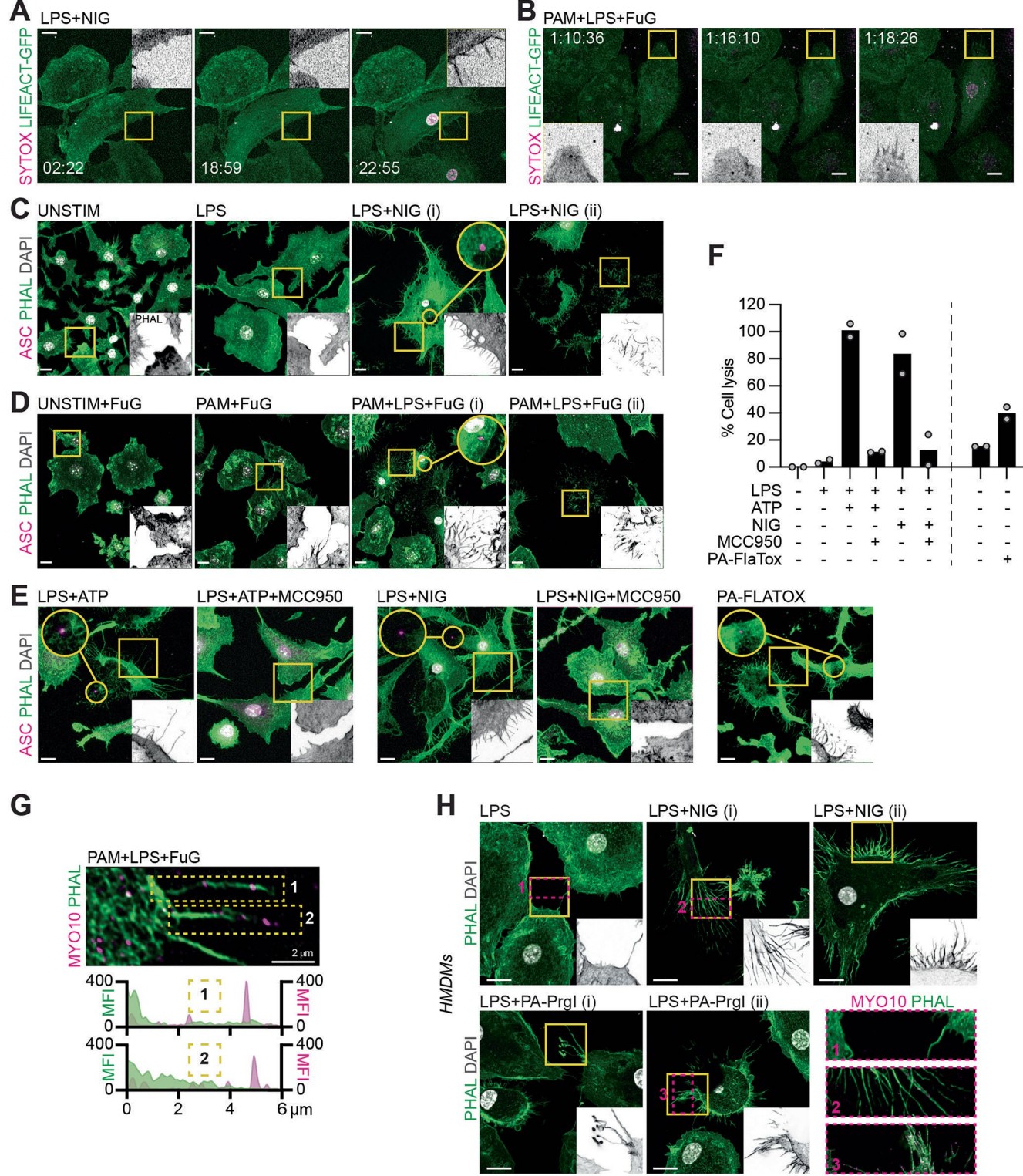

**Extended Data Fig. 1 | See next page for caption.**

**Extended Data Fig. 1 | Inflammasomes trigger the eruption of F-actin projections in macrophages. a-b**, Live fast Airyscan confocal imaging of macrophages from mice expressing the LifeAct- peptide fused to eGFP (green) **a**, primed for 4 h with 100 ng ml⁻¹ ultrapure E. coli LPS and stimulated with 5 μM nigericin, or **b**, primed for 16 h with 1 μg ml⁻¹ PAM3CSK4 and then transfected with 2 μg ml⁻¹ ultrapure E. coli LPS delivered using FuGENE HD, for the time indicated in each panel. Macrophages were incubated with 150 nM Sytox-red (magenta) during imaging. LifeAct-eGFP insets are displayed in grey within each panel. **c-d**, Airyscan confocal imaging of fixed macrophages labelled with DAPI nuclear stain (grey), phalloidin F-actin stain (green, and grey inverted insets in the lower right of panels), and α-ASC antibody (magenta, circular inset). Macrophages were **c**, unstimulated, primed with LPS, or LPS-primed for 4 h and stimulated for 30 min with nigericin, or **d**, mock transfected with FuGENE HD with or without prior priming with PAM3CSK4 for 16 h, or PAM3CSK4-primed and stimulated for 2 h with intracellular LPS packaged with FuGENE HD. **a-d**, Images are maximum intensity projections of Z-stack acquisitions, and representative of n = 3 independent biological experiments. **e**, Confocal imaging of fixed macrophages labelled with phalloidin (green), DAPI (grey), and α-ASC antibody

(magenta). Macrophages were primed with LPS for 4 h and stimulated with 1.25 mM ATP for 1 h, or 5 μM nigericin for 30 min, in the presence or absence of 10 μM MCC950 added 30 min before ATP or nigericin. Unprimed macrophages were stimulated with 250 ng ml⁻¹ PA and 125 ng ml⁻¹ Fla1-LFn (PA-FlaTox) for 1 h to induce NLRC4 signalling. Images are maximum intensity projections of Z-stack acquisitions. **f**, LDH release of **e**, plotted as mean of technical duplicates. **g**, Airyscan confocal imaging of pyroptotic projections labelled with phalloidin (green) and α-MYO10 antibody (magenta). Plots of mean fluorescence intensity (MFI) in regions outlined by dashed yellow rectangles (upper) are displayed in lower panels. Primed macrophages were stimulated with cytosolic LPS, as described earlier. Images are single Z-planes and representative of n = 3 independent biological experiments. **h**, Fixed Airyscan confocal imaging of human monocyte-derived macrophages (HMDM), labelled with F-actin stain phalloidin (green), MYO10 (magenta) and DAPI (grey). HMDM were LPS primed or co-administered LPS plus nigericin (10 μM), or 250 ng ml⁻¹ PA plus 20 ng ml⁻¹ PrgI-LFn (PA-PrgI) to induce NLRC4 signalling. Images are maximum intensity projections of Z-stack acquisitions and representative of n = 3 independent biological experiments. All scale bars = 10 μm except where indicated.

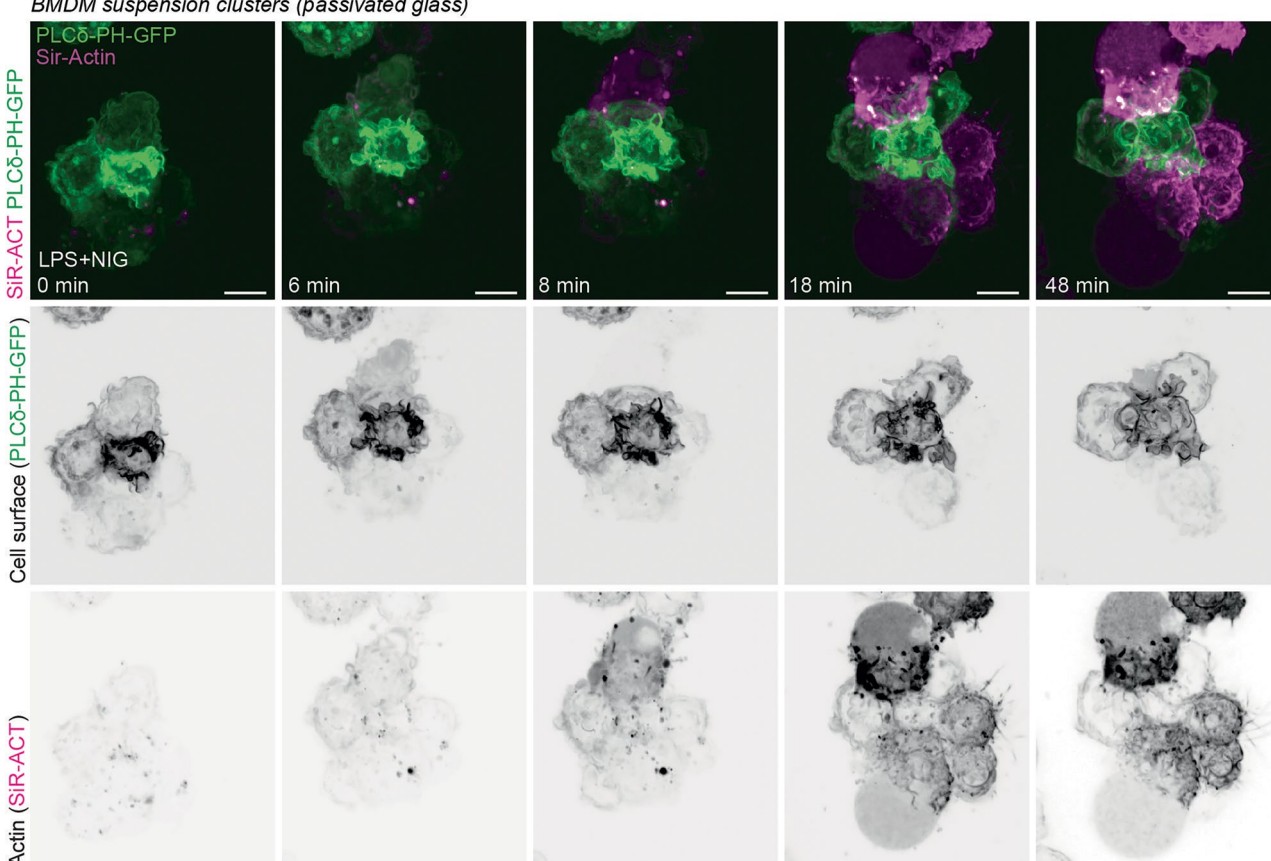

**Extended Data Fig. 2 | Nigericin stimulates the assembly of filopodia in suspended macrophages.** BMDM expressing PLCδ-PH-GFP (cell membrane, green) were grown in suspension on PLL-g-PEG passivated glass and pyroptosis was activated (LPS plus nigericin). Maximum intensity projections time-series of BMDM clusters labelled with cell membrane PLCδ-PH-GFP (green) and F-actin (SiR-actin, magenta). Upon induction of pyroptosis, cell membrane rupture occurs as observed by the loss of green signal (cell membrane PLCδ-PH-GFP, 8 min) and increased positive staining for F-actin (SiR-actin, magenta, 18 min). Data are representative of three independent biological replicates. Scale bars = 10 μm.

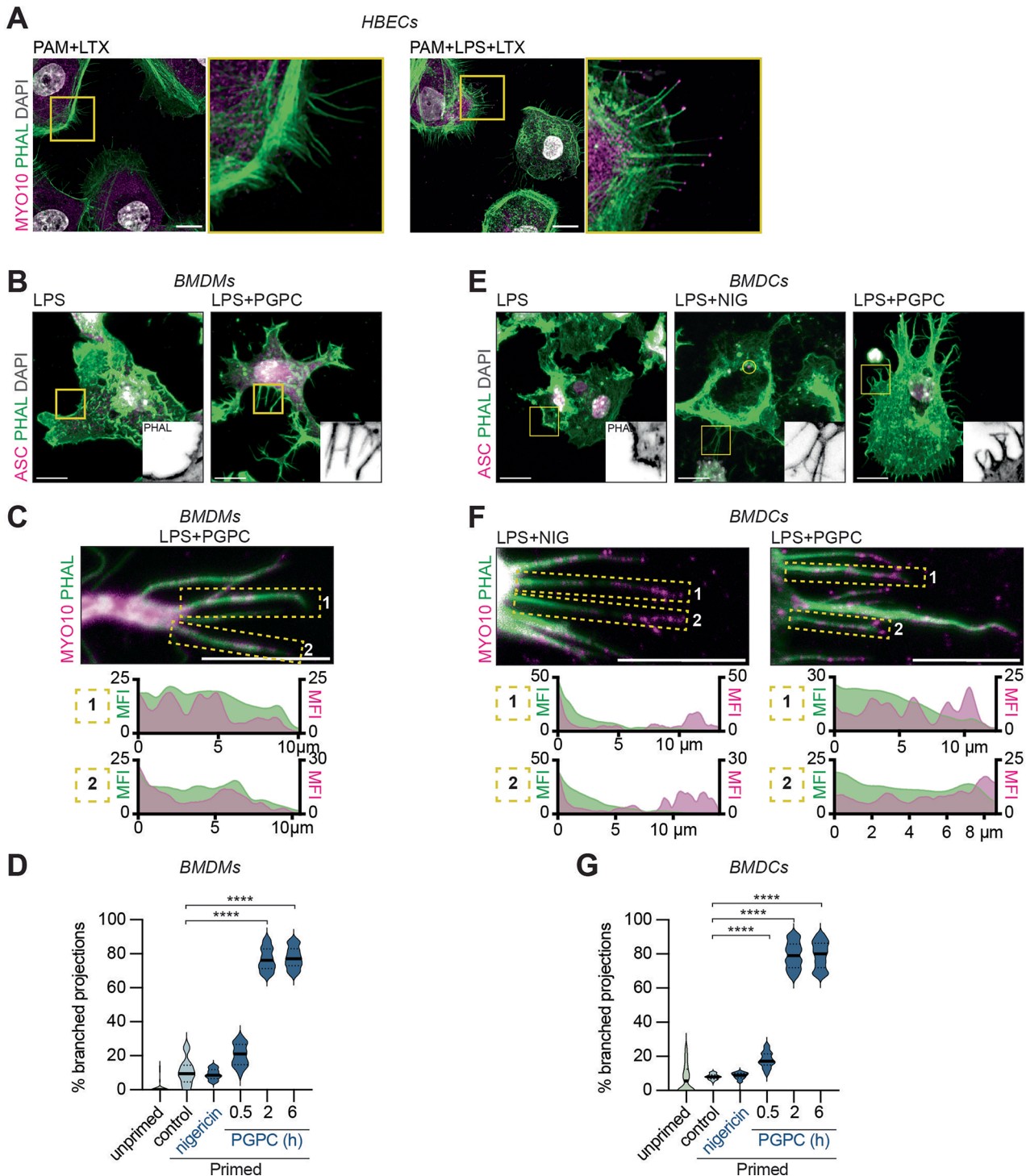

**Extended Data Fig. 3 | Inflammasome activators induce filopodia under non-lytic conditions, and in non-myeloid cells. a**, Fixed Airyscan confocal imaging of human bronchial epithelial cells (HBEC), labelled with F-actin stain phalloidin (green), Myo10 (magenta) and DAPI (grey). Cells were primed for 16 h with 1 μg ml⁻¹ PAM3CSK4 (PAM), and CASP4 was activated with LPS transfection for 5.5 hours. **b-c**, Representative immunofluorescence images of BMDMs or **e-f**, BMDCs primed with LPS and treated with nigericin or PGPC for 0.5 to 6 h to induce pyroptosis or hyperactivation and filopodia formation. **b, e**, Insets depict filopodia. Cells were labelled with F-actin stain phalloidin (green), ASC (magenta) and DAPI (grey). **c, f**, Representative Airyscan immunofluorescence images of filopodia in pyroptotic or hyperactive BMDMs or BMDCs (2 h stimulation). Plots of MFI in regions outlined by dashed yellow rectangles are displayed in lower panels. Cells were labelled with F-actin stain phalloidin (green) and Myo10 (magenta). **d, g**, Quantification of branched projections per cell from BMDMs or BMDCs treated as indicated to induce pyroptosis or sub-lytic inflammasome signalling. Protrusions from 40 randomly chosen cells with filopodia were counted per condition and time point to assess the percentage of projections that were branched. Data were analysed using a Kruskal Wallis test with Dunn's multiple testing correction (n = 25–72 from three independent biological experiments). All images are maximum intensity projections of Z-stack acquisitions and representative of n = 3 independent biological experiments. All scale bars = 10 μm. Violin plots show mean (solid line) and first and third quartiles (dotted lines). Statistical significance: *** p ≤ 0.001; **** p ≤ 0.0001.

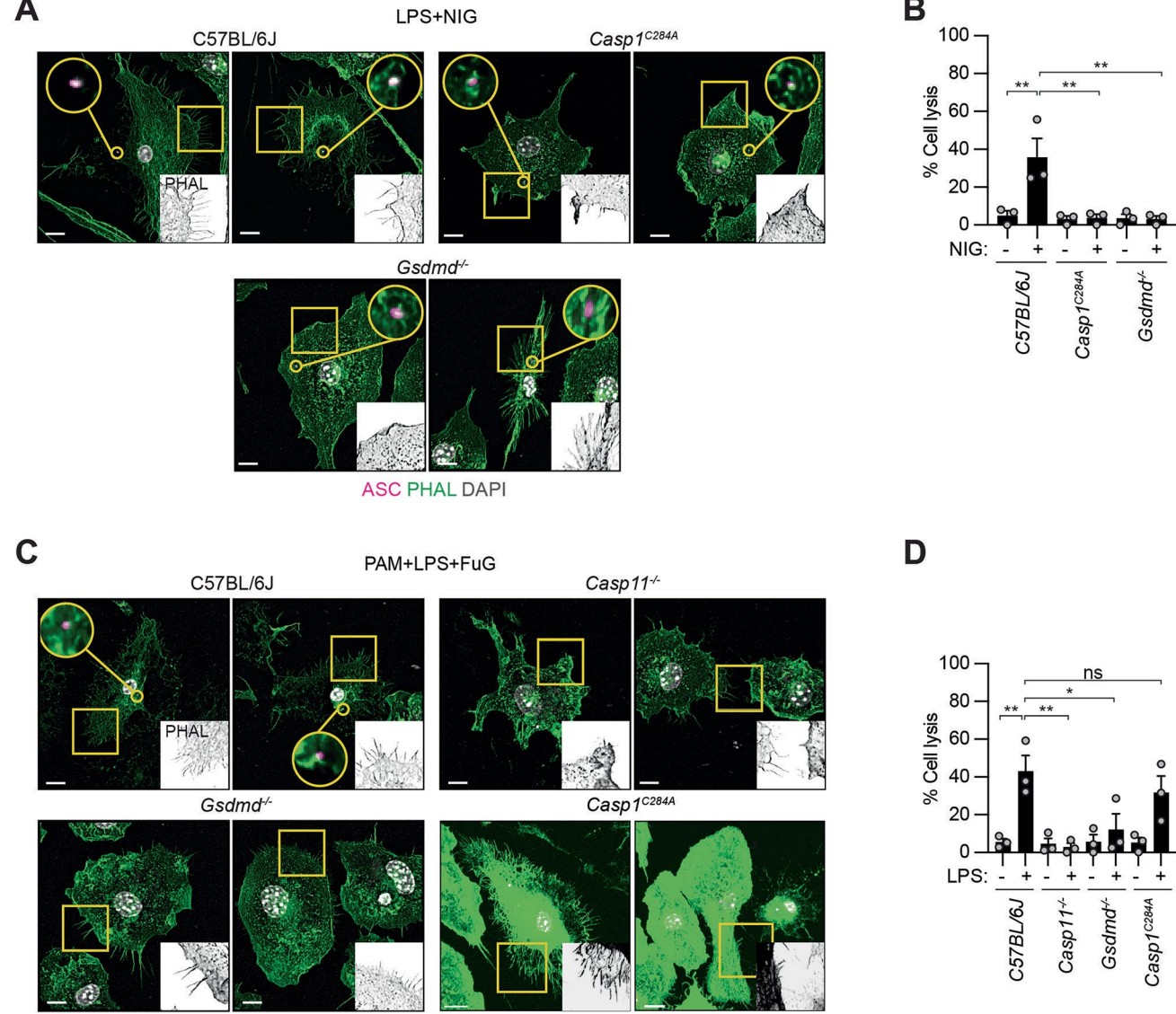

**Extended Data Fig. 4 | GSDMD is required for the assembly of pyroptotic filopodia. a**, Fast Airyscan confocal imaging of WT, Casp1$^{C284A}$ or Gsdmd$^{-/-}$ macrophages labelled with DAPI (grey), F-actin phalloidin (green), and α-ASC antibody (magenta). LPS-primed macrophages stimulated with nigericin. Images are maximum intensity projections of Z-stack acquisitions and representative of n = 3 independent biological experiments. **b**, LDH release data (mean + SEM from n = 3 independent biological experiments; verified for normality and analysed by one-way ANOVA with Dunnett's multiple testing correction) from experiments shown in Extended Data Fig. 4a. **c**, Fast Airyscan confocal imaging of PAM-primed macrophages stimulated with cytosolic LPS, labelled with DAPI (grey), F-actin phalloidin (green), and anti-ASC antibody (magenta). Images are maximum intensity projections of Z-stack acquisitions and are representative of n = 3 independent biological experiments. **d**, LDH release data (mean + SEM from n = 3 independent biological experiments; verified for normality and analysed by one-way ANOVA with Dunnett's multiple testing correction) from experiments shown in Extended Data Fig. 4c. All scale bars = 10 μm.

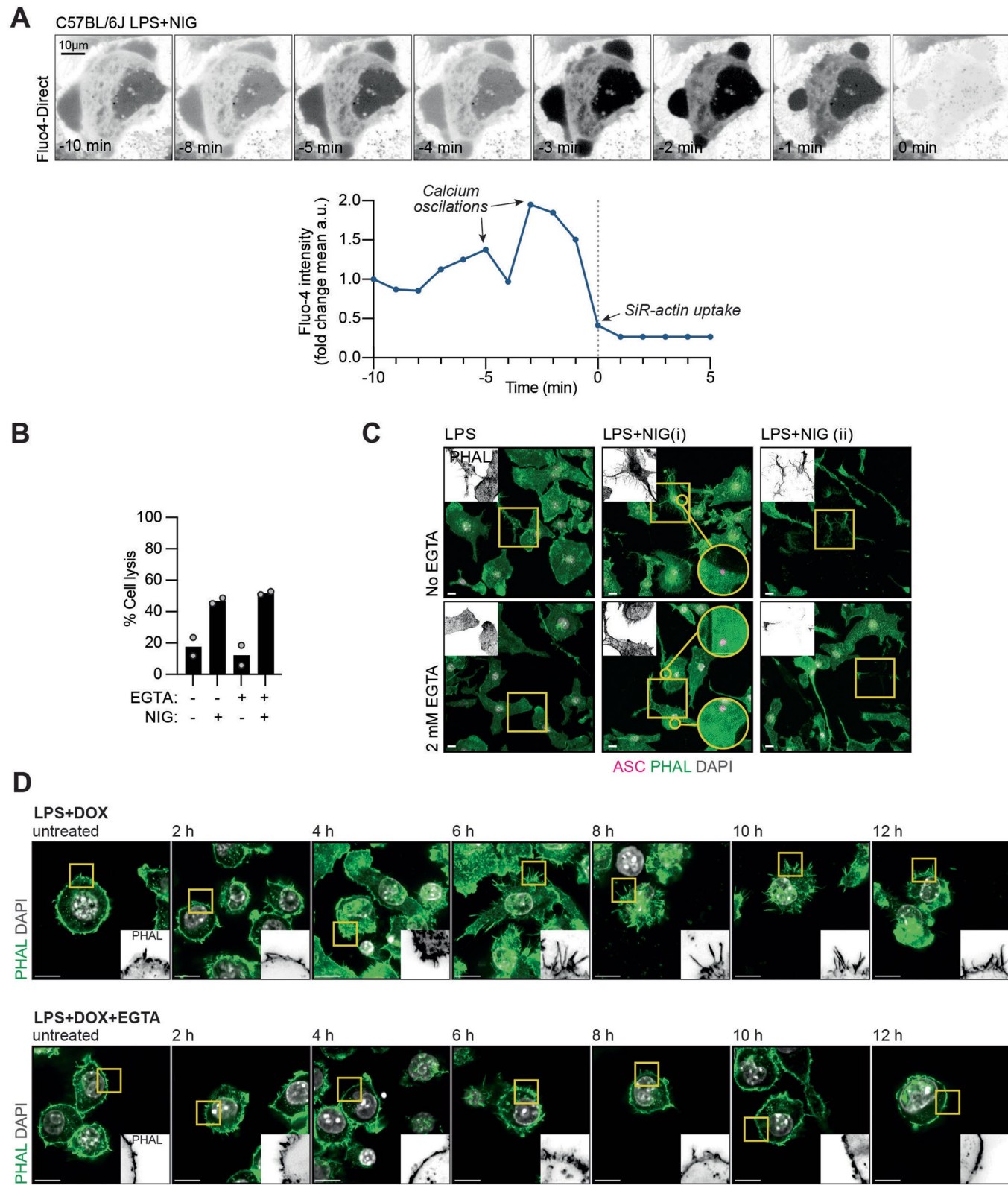

**Extended Data Fig. 5 | GSDMD-dependent Ca2+ influx is sufficient for filopodial assembly. a**, Representation of an individual LPS-primed wild-type BMDM stimulated with nigericin, showcasing the Ca²⁺ elevation oscillations (Fluo-4 intensity) that precede membrane rupture (15 min timeframes extracted from 45 min of imaging). Data are representative of three independent biological replicates. **b**, LDH release data (mean of n = 2 independent experiments) corresponding to Extended Data Fig. 5c data. **c**, Fast Airyscan confocal imaging of macrophages labelled with DAPI (grey), phalloidin (green), and α-ASC antibody (magenta). LPS-primed macrophages were treated with or without nigericin as described earlier, in the presence or absence of 2 mM EGTA added to the cell culture media 30 min before nigericin. Magnified insets of inverted phalloidin are displayed to the right of each panel (grey). Images are maximum intensity projections of Z-stack acquisitions and are representative of n = 3 independent biological experiments. **d**, Representative Airyscan immunofluorescence images of iBMDM+Tet-GsdmD-NT cells treated with doxycycline (Dox) in the presence or absence of EGTA-AM for indicated times. Cells were labelled with DAPI (grey) and phalloidin (green). Data are representative of three independent biological replicates. All scale bars: 10 µm.

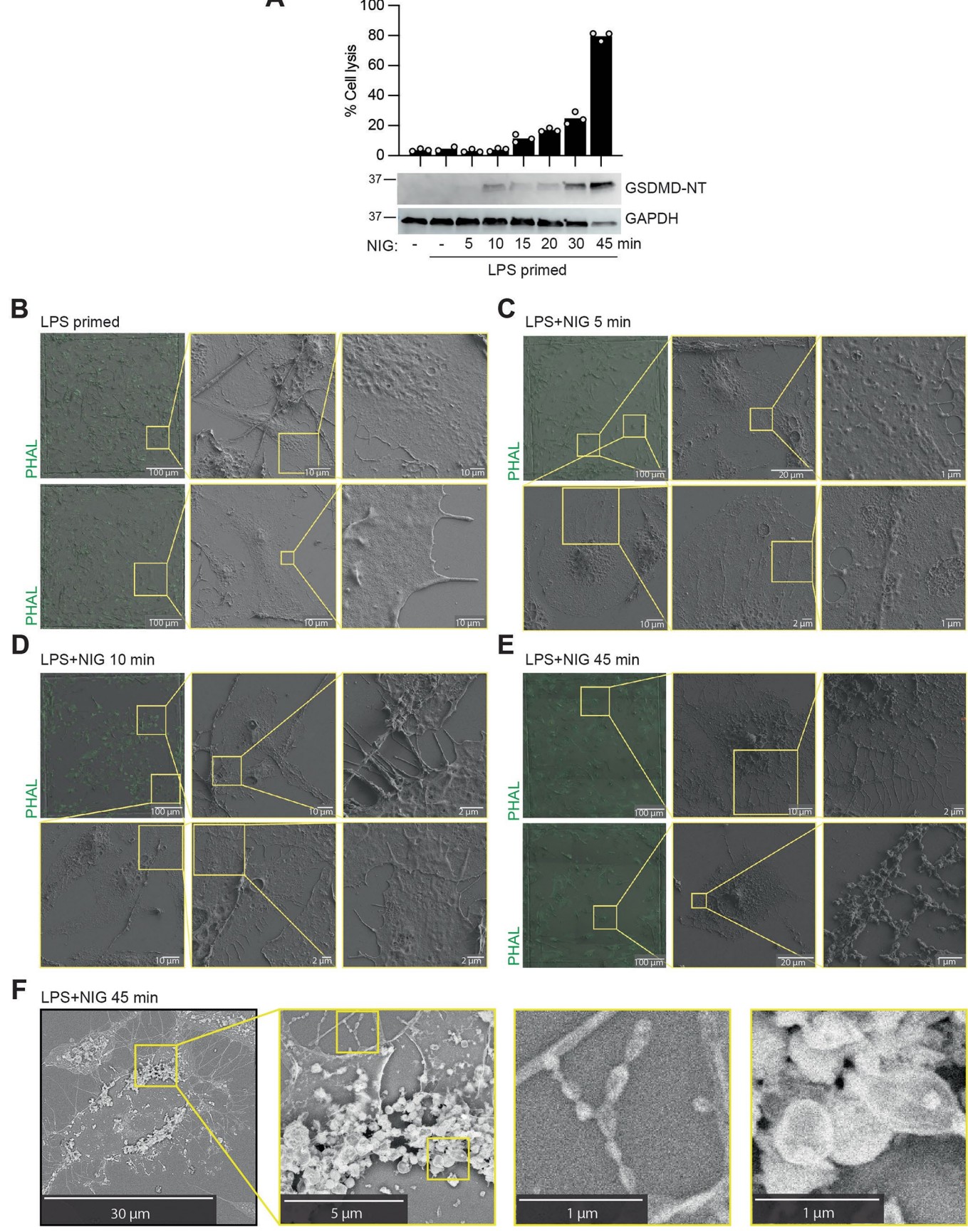

**Extended Data Fig. 6 | See next page for caption.**

**Extended Data Fig. 6 | Correlative light and scanning electron microscopy monitors filopodial assembly over a time course of nigericin stimulation.** **a**, LDH release data (mean of technical replicates from a single experiment, representative of n = 2 independent biological experiments) and NT-GSDMD cleavage immunoblot analysis of cell extracts to accompany panels **b-e** over time.

**b-f**, Scanning EM of LPS primed macrophage stimulated with nigericin to assess filopodia formation at the indicated time points. Scale bars are as indicated, and magnified inserts are shown to the right. All western blots and images are representative of n = 2 independent biological experiments.

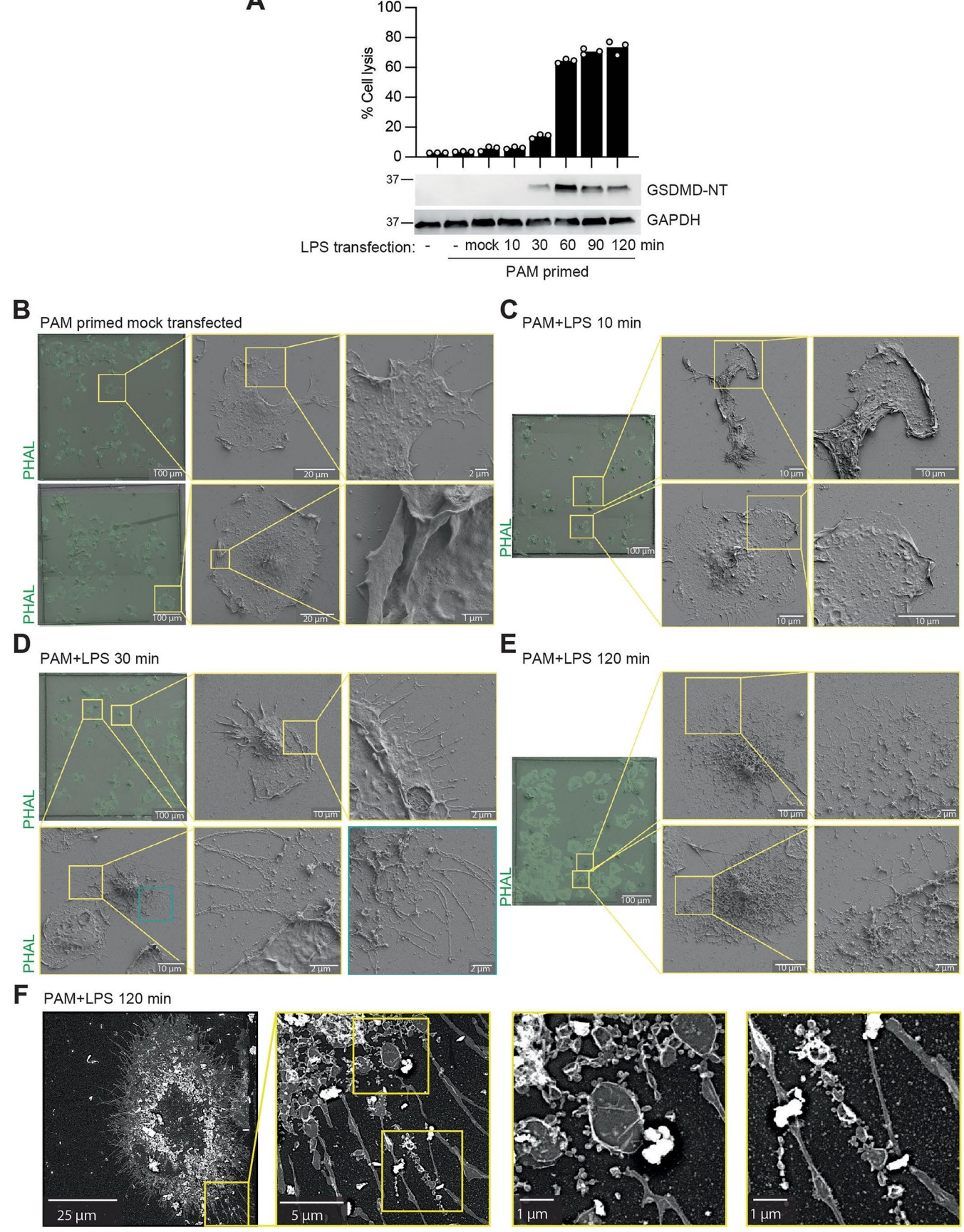

**Extended Data Fig. 7 | See next page for caption.**

**Extended Data Fig. 7 | Correlative light and scanning electron microscopy monitors filopodial assembly over a time course of cytosolic LPS stimulation. a**, LDH release data (mean of technical replicates from a single experiment, representative of n = 2 independent biological experiments) and NT-GSDMD cleavage immunoblot analysis of cell extracts to accompany panels **b-e** over time. **b-f**, Scanning EM of PAM-primed macrophages transfected with LPS for the indicated time points. Scale bars are as indicated, and magnified inserts are shown to the right. All western blots and images are representative of n = 2 independent biological experiments.

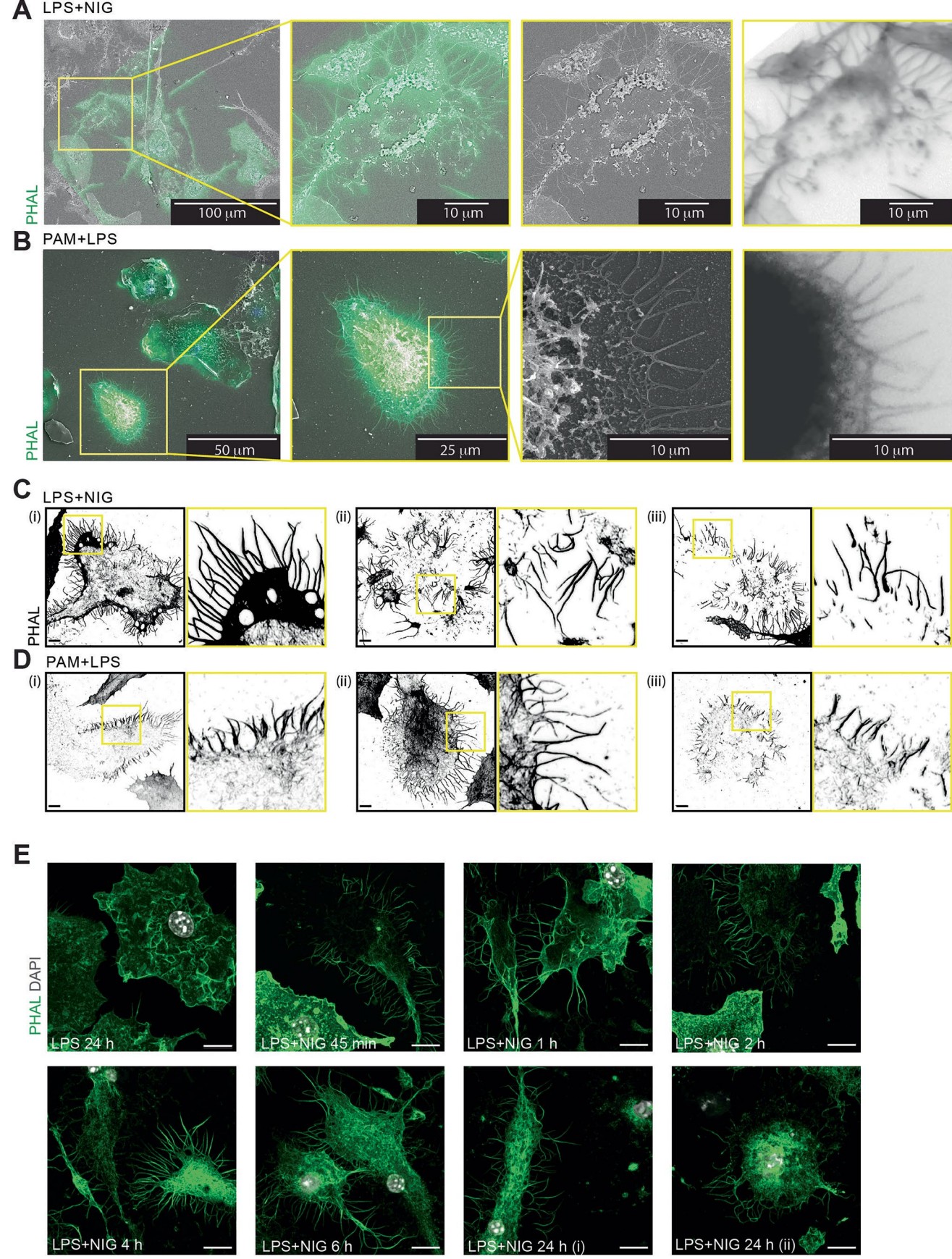

**Extended Data Fig. 8 | See next page for caption.**

**Extended Data Fig. 8 | Inflammasome agonists generate stable caches of F-actin within filopodia. a-e**, Microscopy of macrophage corpses that were generated by exposing primed macrophages to **a,c,e** nigericin (45 mins or as indicated), or **b,d**, cytosolic LPS (2 h). Panels **a-b** show cells imaged with correlative light and electron microscopy to colocalise actin (phalloidin, green). **c-e**, Representative Airyscan immunofluorescence images of cells labelled with DAPI (grey) and phalloidin (green). Images are maximum intensity projections of Z-stack acquisitions. Cells in panel **e** were cultured for up to 24 h post-nigericin to assess filopodia stability over time. All scale bars = 10 μm unless otherwise indicated, and all images are representative images of n = 3 independent biological experiments.

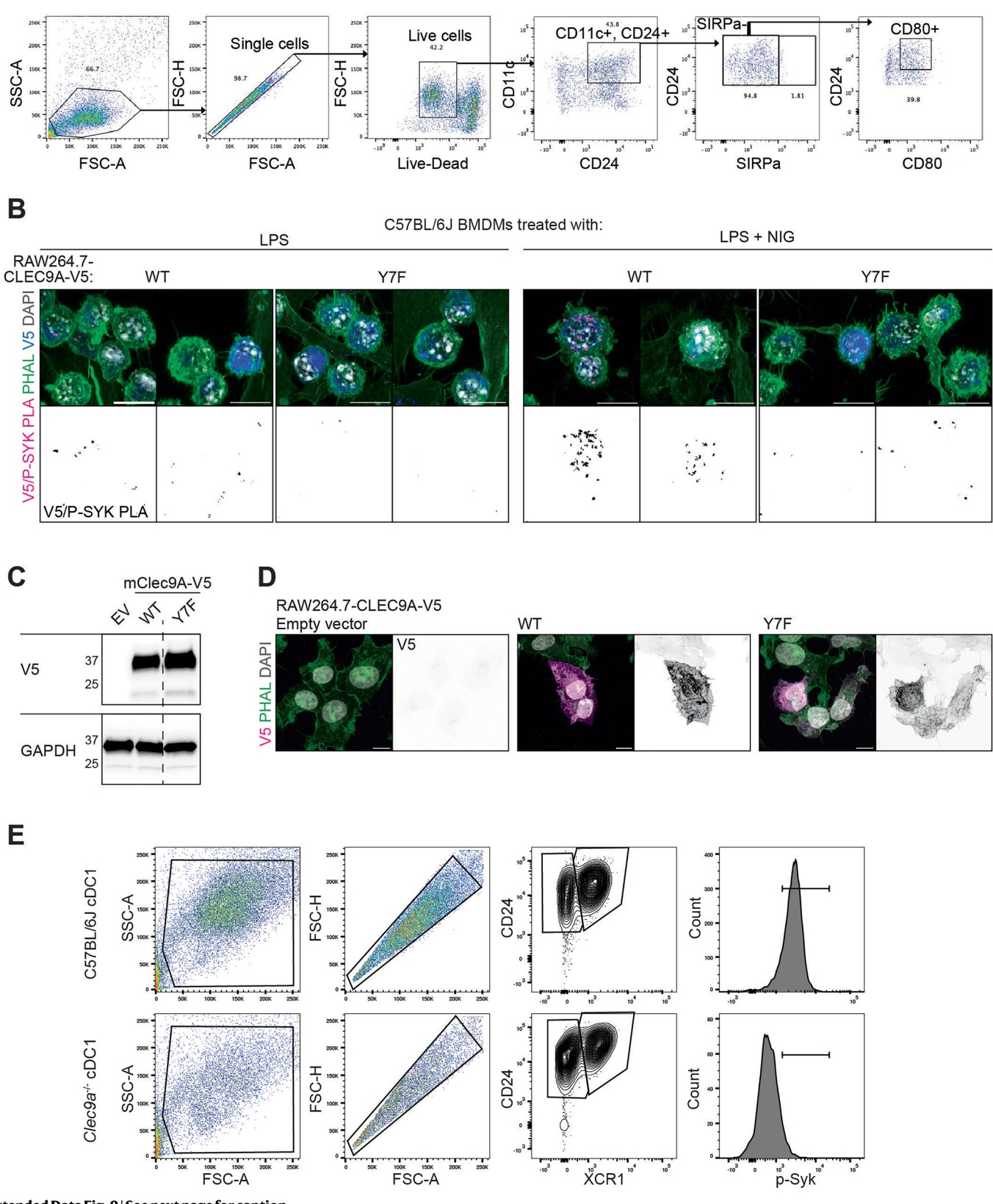

**Extended Data Fig. 9 | See next page for caption.**

**Extended Data Fig. 9 | Filopodia from pyroptotic corpses activate CLEC9A.**
**a**, Gating strategy to identify cDC1s positive for p-SYK upon dendritic cell incubation with macrophages. **b**, Airyscan confocal images of macrophages (primed live cells, versus pyroptotic corpses generated with nigericin) incubated in the presence 50 μM pervanadate with RAW264.7 cells stably expressing murine CLEC9A (wildtype or Y7F signal-deficient mutant) with a V5 tag conjugated to the cytosolic C-terminus. Cells were labelled with phalloidin (green), DAPI (grey), α-V5 antibody (blue), and the proximity ligation assay (PLA) reaction produced puncta when α-V5 and α-p-SYK antibodies co-located (magenta). Data are representative of three independent biological replicates.

**c**, Immunoblot of HEK293T cells ectopically expressing the indicated pEF6-CLEC9A-V5 constructs for 16 h versus pEF6 empty vector (EV), from a single experiment. CLEC9A is 25 kDa, and predicted to be 28-29 kDa with the V5 tag and linker. GAPDH is 37 kDa. **d**, Confocal imaging of fixed HEK293T cells expressing the indicated pEF6-CLEC9A-V5 plasmids, from a single experiment. Cells were labelled with phalloidin (green), DAPI (grey), and α-V5 antibody (magenta in left panels, inverted grey in right panels). **e**, Gating strategy to identify cDC1s and their activation by incubation with macrophage pyroptotic corpses. All images are maximum intensity projections of Z-stack acquisitions, with 10 μm scale bars.

# Reporting Summary

## Statistics

For all statistical analyses, confirm that the following items are present in the figure legend, table legend, main text, or Methods section.

| n/a | Confirmed | |
|---|---|---|
| ☐ | ☒ | The exact sample size (*n*) for each experimental group/condition, given as a discrete number and unit of measurement |
| ☒ | ☐ | A statement on whether measurements were taken from distinct samples or whether the same sample was measured repeatedly |
| ☐ | ☒ | The statistical test(s) used AND whether they are one- or two-sided *Only common tests should be described solely by name; describe more complex techniques in the Methods section.* |
| ☒ | ☐ | A description of all covariates tested |
| ☐ | ☒ | A description of any assumptions or corrections, such as tests of normality and adjustment for multiple comparisons |
| ☐ | ☒ | A full description of the statistical parameters including central tendency (e.g. means) or other basic estimates (e.g. regression coefficient) AND variation (e.g. standard deviation) or associated estimates of uncertainty (e.g. confidence intervals) |
| ☐ | ☒ | For null hypothesis testing, the test statistic (e.g. *F*, *t*, *r*) with confidence intervals, effect sizes, degrees of freedom and *P* value noted *Give P values as exact values whenever suitable.* |
| ☒ | ☐ | For Bayesian analysis, information on the choice of priors and Markov chain Monte Carlo settings |
| ☒ | ☐ | For hierarchical and complex designs, identification of the appropriate level for tests and full reporting of outcomes |
| ☒ | ☐ | Estimates of effect sizes (e.g. Cohen's *d*, Pearson's *r*), indicating how they were calculated |

*Our web collection on statistics for biologists contains articles on many of the points above.*

## Software and code

Policy information about availability of computer code

| Data collection | Immunofluorescence: Zeiss Zen 2012 software v2.3<br>Plate reader Tecan: i-controlTM (v2.0)<br>LLAMA visualisation software: Lefevre, J.G. et al. LLAMA: a robust and scalable machine learning pipeline for analysis of large scale 4D microscopy data: analysis of cell ruffles and filopodia. BMC Bioinformatics 22, 410 (2021).<br>Flow cytometry: BD FACSDiva (v9.0) |
|---|---|
| Data analysis | Prism (v9/10)<br>Arivis Vision4D (v4.1.1)<br>Fiji/ImageJ (v1.52k-1.54f)<br>Microvolution Deconvolution: https://www.microvolution.com/<br>FlowJo (v.10.9) |

For manuscripts utilizing custom algorithms or software that are central to the research but not yet described in published literature, software must be made available to editors and reviewers. We strongly encourage code deposition in a community repository (e.g. GitHub). See the Nature Portfolio guidelines for submitting code & software for further information.

## Data

Policy information about availability of data

All manuscripts must include a data availability statement. This statement should provide the following information, where applicable:
- Accession codes, unique identifiers, or web links for publicly available datasets
- A description of any restrictions on data availability
- For clinical datasets or third party data, please ensure that the statement adheres to our policy

> The data that support the findings of this study are available from the corresponding author

## Research involving human participants, their data, or biological material

Policy information about studies with human participants or human data. See also policy information about sex, gender (identity/presentation), and sexual orientation and race, ethnicity and racism.

| | |
|---|---|
| Reporting on sex and gender | Human monocyte-derived macrophages (HMDM) were produced from buffy coats from blood donations to the Australian Red Cross Blood Service from anonymous, informed and consenting adults. As a consequence, sex characteristics of the donor material were not known. |
| Reporting on race, ethnicity, or other socially relevant groupings | Race, ethnicity, and other characateristics of the blood donors were not provided in accordance with the ethical approval. |
| Population characteristics | See above. |
| Recruitment | Anonymous volunteers |
| Ethics oversight | Ethical approval for all experiments involving human blood products was approved by the Research Ethics and Integrity Committee at the University of Queensland (2024/HE000645 and 2016/HE0001044) |

Note that full information on the approval of the study protocol must also be provided in the manuscript.

# Field-specific reporting

Please select the one below that is the best fit for your research. If you are not sure, read the appropriate sections before making your selection.

☒ Life sciences    ☐ Behavioural & social sciences    ☐ Ecological, evolutionary & environmental sciences

For a reference copy of the document with all sections, see nature.com/documents/nr-reporting-summary-flat.pdf

# Life sciences study design

All studies must disclose on these points even when the disclosure is negative.

| | |
|---|---|
| Sample size | No statistical method was used to determine sample size. Sample size for image quantification experiments was based on technical feasibility and previously published studies (e.g. PMID: 29432122, PMID: 30150290). |
| Data exclusions | No data points were excluded from analyses. |
| Replication | Replications were successful. Where possible, counter assays (e.g. accompanying cell death assays) were used to confirm consistency of responses in cells used for imaging.<br>In vitro experiments; each experiments was performed in n=3 technical replicate, mean of each biological experiments were pooled, and each experiment was performed with at least n=3 biological replicates.<br>In vivo experiments; cohort of 3-4 mice were used for each individual experiments, with n=3 biological replicates |
| Randomization | For imaging experiments, regions of interest were randomly chosen for quantification. Otherwise, randomisation was not applicable as cells from the same source or differentiation were split into different conditions and/or stimulations. |
| Blinding | The number of projections per cell was quantified by manual counting by a blinded operator.<br>For in vivo experiments; mice genotypes were unidentified and randomly allocated to groups. Imaging of peritoneal lavage cells were performed from another researcher on unidentified samples.<br>For cell death analysis (LDH release) and flow cytometry analysis (p-Syk, and DC activation) no blinding was performed |

# Reporting for specific materials, systems and methods

We require information from authors about some types of materials, experimental systems and methods used in many studies. Here, indicate whether each material, system or method listed is relevant to your study. If you are not sure if a list item applies to your research, read the appropriate section before selecting a response.

## Materials & experimental systems

| n/a | Involved in the study |
|---|---|
| ☐ | ☒ Antibodies |
| ☐ | ☒ Eukaryotic cell lines |
| ☒ | ☐ Palaeontology and archaeology |
| ☐ | ☒ Animals and other organisms |
| ☒ | ☐ Clinical data |
| ☒ | ☐ Dual use research of concern |
| ☒ | ☐ Plants |

## Methods

| n/a | Involved in the study |
|---|---|
| ☒ | ☐ ChIP-seq |
| ☐ | ☒ Flow cytometry |
| ☒ | ☐ MRI-based neuroimaging |

## Antibodies

**Antibodies used**

anti-ASC (1:200, N15; Santa-Cruz; 1:800, 67824S, Cell Signaling Technology), anti-MYO10 (1:100, AB224120, Abcam), anti-phospo-SYK (1:100, 2711S, Cell Signaling Technology), anti-Casp3p17 (1:200, 9661, Cell Signaling Technology), and anti-V5 (1:200, AB27671, Abcam). PE anti-mouse p-Syk (1:50, clone l120-722, BD Biosciences #558529), BV711 anti-mouse CD24 (1:800, clone M1/69, BD Biosciences #563450), APC anti-mouse XCR1 (1:200, clone ZET, Biolegend #148206), and BUV395 anti-mouse CD80 (1:200, clone 16-10A1, BD Biosciences #740246), FITC anti-mouse CD11c (1:200, clone N418, Biolegend #117306), PerCP-eFluor 710 anti-mouse SIRP alpha (1:400, clone P84, Invitrogen #46-1721-82), PE/Cyanine7 anti-mouse CD24 (1:400, clone M1/69, Biolegend #101822), PE anti-mouse CD86 (1:400, clone GL-1, Biolegend #105008), BV711 anti-mouse CD80 (1:200, clone 16-10A1, Biolegend #104743), APC anti-mouse I-A/I-E (MHCII) (1:200, clone M5/114.15.2, Biolegend #107614), and Live/Dead Fixable Violet Dead, (1:400, ThermoFisher # L34964), DAPI (0.1μg/mL, Sigma Aldrich #D9542), Phalloidin-iFluor 405 (1:40, Abcam #AB176752), Alexa Fluor 488 Phalloidin (1:40, Invitrogen #A12379), Alexa Fluor 647 Phalloidin (1:40, Invitrogen #A22287), anti-rabbit Alexa Fluor 594 (1:500, Invitrogen #A32740), donkey anti-mouse Alexa Fluor 488 (1:500, Invitrogen #A21202), and goat anti-rabbit Aexa Fluor 647 (1:500, Invitrogen #A32733).

**Validation**

anti-ASC (Santa Cruz): https://www.scbt.com/de/p/asc-antibody-n-15 - this antibody has been discontinued

anti-ASC (Cell Signaling Technology): https://www.cellsignal.com/products/primary-antibodies/asc-tms1-d2w8u-rabbit-mab/67824, antibody has been validate by Cell Signaling Technology for mouse cell lines J774.A and Raw264.7 cells for Western blot and Flow cytometry. Paraffin-embedded mouse brain, mouse thymus or mouse colon for immunohistochemistry and mouse Tg2576 brain, mouse primary bone marrow-derived macrophage for immunofluorescent

anti-MYO10: https://www.abcam.com/en-us/products/primary-antibodies/myo10-antibody-ab224120#application=icc-if
antibody has been validated by Abcam on human cell line HeLa for immunofluorescence

anti-phospho-Syk (optimised for IF and FC, using positive and negative controls for SYK phosphorylation): https://www.cellsignal.com/products/primary-antibodies/phospho-syk-tyr525-526-antibody/2711 and has been validated by Cell Signaling Technology on human Ramos cells for Western Blot.

anti-Casp3p17: https://www.cellsignal.com/products/primary-antibodies/cleaved-caspase-3-asp175-antibody/9661
antibody has been validated by Cell Signaling Technology on human and mouse cell lines HeLA, NiH/3T3, C6 cells, Jurkat cells for Western Blot. Paraffin-embedded human tonsils, mouse embryos were used for validation for immunohistochemistry. HT-29 cells were used for validation of immunofluorescence.

anti-V5: https://www.abcam.com/en-us/products/primary-antibodies/v5-tag-antibody-sv5-pk1-ab27671
antibody has been validated by Abcam on transfected cell lysates (simian virus 5 strain W3) for Western blot.

PE anti-mouse p-Syk (clone l120-722, BD Biosciences #558529), https://www.bdbiosciences.com/en-au/products/reagents/flow-cytometry-reagents/research-reagents/single-color-antibodies-ruo/pe-mouse-anti-syk-py348.558529
antibody has been validated by BD Bioscience on human peripheral blood lymphocytes for flow cytometry, mouse testing is in development by BD Bioscience

BV711 anti-mouse CD24 (clone M1/69, BD Biosciences #563450), https://www.bdbiosciences.com/en-au/products/reagents/flow-cytometry-reagents/research-reagents/single-color-antibodies-ruo/bv711-rat-anti-mouse-cd24.563450
antibody has been validated by BD Bioscience on mouse spleenocytes for flow cytometry

APC anti-mouse XCR1 (clone ZET, Biolegend #148206),  https://www.biolegend.com/fr-ch/products/apc-anti-mouse-rat-xcr1-antibody-10222
antibody has been validated by Biolegend on mouse spleenocytes for flow cytometry

BUV395 anti-mouse CD80 (clone 16-10A1, BD Biosciences #740246), https://www.bdbiosciences.com/en-au/products/reagents/flow-cytometry-reagents/research-reagents/single-color-antibodies-ruo/buv395-hamster-anti-mouse-cd80.740246
antibody has been validated by BD Bioscience on mouse transfected cell lines

FITC anti-mouse CD11c (clone N418, Biolegend #117306), https://www.biolegend.com/fr-ch/products/fitc-anti-mouse-cd11c-antibody-1815
antibody has been validated by Biolegend on mouse spleenocytes for flow cytometry

PerCP-eFluor 710 anti-mouse SIRP alpha (clone P84, Invitrogen #46-1721-82), https://www.thermofisher.com/antibody/product/CD172a-SIRP-alpha-Antibody-clone-P84-Monoclonal/46-1721-82
antibody has been validated by ThermoFisher on mouse bone marrow cells and bovine PBMC for flow cytometry

PE/Cyanine7 anti-mouse CD24 (clone M1/69, Biolegend #101822), https://www.biolegend.com/fr-ch/products/pe-cyanine7-anti-mouse-cd24-antibody-3862
antibody has been validated by Biolegend on mouse spleenocytes for flow cytometry

PE anti-mouse CD86 (clone GL-1, Biolegend #105008), https://www.biolegend.com/fr-ch/products/pe-anti-mouse-cd86-antibody-256
antibody has been validated by Biolegend on mouse spleenocytes for flow cytometry

BV711 anti-mouse CD80 (clone 16-10A1, Biolegend #104743), https://www.biolegend.com/fr-ch/products/brilliant-violet-711-anti-mouse-cd80-antibody-17823
antibody has been validated by Biolegend on mouse spleenocytes for flow cytometry

APC anti-mouse I-A/I-E (MHCII) (clone M5/114.15.2, Biolegend #107614), https://www.biolegend.com/fr-ch/products/apc-anti-mouse-i-a-i-e-antibody-2488
antibody has been validated by Biolegend on mouse spleenocytes for flow cytometry

Live/Dead Fixable Violet Dead, (ThermoFisher # L34964), https://www.thermofisher.com/order/catalog/product/L34964
antibody has been validated by ThermoFisher on Jurkat cells for flow cytometry

Phalloidin-iFluor 405 ( Abcam #AB176752), https://www.abcam.com/en-us/products/reagents/phalloidin-ifluor-405-reagent-ab176752
antibody has been validated by Abcam on bovine BFA cell lines for immunofluorescence

Alexa Fluor 488 Phalloidin (Invitrogen #A12379), https://www.thermofisher.com/order/catalog/product/A12379
antibody has been validated by ThermoFisher on human dermal fibroblasts, bovine pulmonary artery endothelial cells, HeLA cells line for immunofluorescence

Alexa Fluor 647 Phalloidin (Invitrogen #A22287), https://www.thermofisher.com/order/catalog/product/A22287
antibody has been validated by ThermoFisher on multiple cells lines including human A549, BPAE, HCASM, HeLA, U2OS and HUVECs for immunofluorescence

goat anti-rabbit Alexa Fluor 594 (Invitrogen #A32740), https://www.thermofisher.com/antibody/product/Goat-anti-Rabbit-IgG-H-L-Highly-Cross-Adsorbed-Secondary-Antibody-Polyclonal/A32740
antibody has been validated by ThermoFisher using positive MCF10A and negative A-431 cell models as well as different cell lines for immunofluorescence

donkey anti-mouse Alexa Fluor 488 (Invitrogen #A21202), https://www.thermofisher.com/antibody/product/Donkey-anti-Mouse-IgG-H-L-Highly-Cross-Adsorbed-Secondary-Antibody-Polyclonal/A-21202
antibody has been validated by ThermoFisher using different cell lines (MDCK), and primary cells including rat primary cortical neutrons or zebrafsih for immunofluorescence

goat anti-rabbit Aexa Fluor 647 (Invitrogen #A32733) https://www.thermofisher.com/antibody/product/Goat-anti-Rabbit-IgG-H-L-Highly-Cross-Adsorbed-Secondary-Antibody-Polyclonal/A32733.
antibody has been validated by ThermoFisher using different cell lines (A549, THP-1, HeLa, MCF7, HEK-293), and primary cells including primary cortical neutrons or rat brain sections for immunofluorescence

# Eukaryotic cell lines

Policy information about cell lines and Sex and Gender in Research

| | |
|---|---|
| Cell line source(s) | RAW 264.7 (Mouse, male): American Type Culture Collection TIB-71, https://www.atcc.org/products/tib-71<br>HBEC (human): https://www.atcc.org/products/pcs-300-010<br><br>Platinum-E (Plat-E) cell line (human): HMorita, S., Kojima, T. & Kitamura, T. Plat-E: an efficient and stable system for transient packaging of retroviruses. Gene Ther 7, 1063–1066 (2000). https://doi.org/10.1038/sj.gt.3301206, kindly provided by Prof. Matt Sweet, IMB, The University of Queensland, Brisbane, Australial<br><br>HEK293T (human, female): https://www.cellosaurus.org/CVCL_0063 |
| Authentication | Cell lines were not authenticated. |
| Mycoplasma contamination | All cell lines used were Mycoplasma negative Cells were routinely tested for Mycoplasma contamination via Mycoalert kits or PCR. |
| Commonly misidentified lines (See ICLAC register) | No commonly misidentified cell lines were used in this study. |

# Animals and other research organisms

Policy information about studies involving animals; ARRIVE guidelines recommended for reporting animal research, and Sex and Gender in Research

| | |
|---|---|
| Laboratory animals | Wildtype (C57BL/6J, or littermate controls as appropriate), Casp1C284A/C284A (B6J-Casp1C284Aem1Ksc CR; generated in-house), Casp11-/- (B6.Casp4tm; backcrossed to C57BL/6J), Gsdmd-/- (C57BL/6N-Gasdmdem1Vmd; backcrossed to C57BL/6J), Ninj1-/- (C57BL/6N-Ninj1-ENU-KO) and Clec9a-GFP knock-in (Jackson, B6(Cg)-Clec9atm1.1Crs/J) mice were used in this study. Mice were bred in-house and housed in specific pathogen-free conditions at the University of Queensland. 6-20 week-old male and female animals were sex- and aged-matched for each experiment. Mice were kept at 12 hour dark/light cycle at ambient temperature with ad libitum feeding and water. |
| Wild animals | The study did not involve wild animals. |
| Reporting on sex | Findings apply to both sexes. Bone marrow for derivation of primary macrophages was harvested from both male and female mice. LPS challenge in vivo was performed only one female mice due to co-housing of different genotypes. |
| Field-collected samples | The study did not involve samples collected from the field. |
| Ethics oversight | All experiments involving mice were approved by The University of Queensland Molecular Biosciences Animal Ethics Committee (2021/AE000419, 2023/AE000019, 2023/AE000020). |

Note that full information on the approval of the study protocol must also be provided in the manuscript.

# Plants

| | |
|---|---|
| Seed stocks | *Report on the source of all seed stocks or other plant material used. If applicable, state the seed stock centre and catalogue number. If plant specimens were collected from the field, describe the collection location, date and sampling procedures.* |
| Novel plant genotypes | *Describe the methods by which all novel plant genotypes were produced. This includes those generated by transgenic approaches, gene editing, chemical/radiation-based mutagenesis and hybridization. For transgenic lines, describe the transformation method, the number of independent lines analyzed and the generation upon which experiments were performed. For gene-edited lines, describe the editor used, the endogenous sequence targeted for editing, the targeting guide RNA sequence (if applicable) and how the editor was applied.* |
| Authentication | *Describe any authentication procedures for each seed stock used or novel genotype generated. Describe any experiments used to assess the effect of a mutation and, where applicable, how potential secondary effects (e.g. second site T-DNA insertions, mosiacism, off-target gene editing) were examined.* |

# Flow Cytometry

## Plots

Confirm that:

☐ The axis labels state the marker and fluorochrome used (e.g. CD4-FITC).

☒ The axis scales are clearly visible. Include numbers along axes only for bottom left plot of group (a 'group' is an analysis of identical markers).

☒ All plots are contour plots with outliers or pseudocolor plots.

☒ A numerical value for number of cells or percentage (with statistics) is provided.

## Methodology

| | |
|---|---|
| Sample preparation | For analysing p-Syk signalling in cDC1 dendritic cells; Flt3L-BMDCs were washed with PBS and immediately fixed with 1% PFA for 20 minutes at room temperature. Cells were spin down at 500 g for 5 minutes and resuspended in pre-chilled 300 μL True-Phos Perm Buffer (425401, Biolegend) and incubated at -20C for 45 minutes. Cells were washed with PBS + 1% FBS and spun at 2000 g for 5 minutes. Flt3L-BMDCs were stained for 45 minutes at 4°C in PBS with 1% FBS containing the following fluorescently conjugated antibodies and dyes.<br>For analysing cDC1 dendritic cell activation, Flt3L-BMDCs were washed and stained for 20 minutes at 4°C in MACS buffer (PBS with 1% FBS, 2 mM EDTA) containing the following fluorescently conjugated antibodies and dye. |
| Instrument | BD LSRFortessa X-20 (BD Biosciences) and  FACSymphony A5 SE (BD Biosciences) |
| Software | Data acquisition: BD FACSDiva v9.0<br>Data analysis: FlowJo v.10.9 |
| Cell population abundance | Total of 20'000 cells in SSC/FSC were acquired |
| Gating strategy | For analysing p-Syk signalling in cDC1 dendritic cells; Total cells SSC-A/FSC-A -> Single cells FSC-H/FSC-A -> XCR1+ CD24/XCR1 |

Gating strategy

-> p-Syk pos Histogram Count/p-Syk
For analysing cDC1 dendritic cell activation; Total cells SSC-A/FSC-A -> Single cells FSC-H/FSC-A -> Live cells FSC-H/Live-Dead -> Dendritic cells CD11c/CD24 -> cDC1 CD24/SIRPa -> Activation CD24/CD80 or CD24/CD86 or CD24/MHCII

☒ Tick this box to confirm that a figure exemplifying the gating strategy is provided in the Supplementary Information.

