## [Peer Review File · Nature Immunology]

Pyroptotic corpses are crowned with F-actin-rich filopodia that engage CLEC9A signalling in incoming dendritic cells

Corresponding Author: Professor Kate Schroder

Version 0:

Decision Letter:

14th Dec 2023

Dear Professor Schroder,

Thank you for providing a response to reviewers concerns regarding your letter "Pyroptotic F-actin remodelling engages CLEC9A, a bridge between innate and adaptive immunity". We would be interested in considering a revised version that addresses reviewers serious concerns, as you described in your response.

If you choose to revise your manuscript taking into account all reviewer and editor comments, please highlight all changes in the manuscript text file [OPTIONAL: in Microsoft Word format].

* If you have not done so already please begin to revise your manuscript so that it conforms to our Letter format instructions at <http://www.nature.com/ni/authors/index.html>. Refer also to any guidelines provided in this letter.

The Reporting Summary can be found here:

Link Redacted

Note: This URL links to your confidential home page and associated information about manuscripts you may have submitted, or that you are reviewing for us. If you wish to forward this email to co-authors, please delete the link to your

homepage.

If you wish to submit a suitably revised manuscript we would hope to receive it within 6 months. If you cannot send it within this time, please let us know. We will be happy to consider your revision so long as nothing similar has been accepted for publication at Nature Immunology or published elsewhere.

Nature Immunology is committed to improving transparency in authorship. As part of our efforts in this direction, we are now requesting that all authors identified as 'corresponding author' on published papers create and link their Open Researcher and Contributor Identifier (ORCID) with their account on the Manuscript Tracking System (MTS), prior to acceptance. ORCID helps the scientific community achieve unambiguous attribution of all scholarly contributions. You can create and link your ORCID from the home page of the MTS by clicking on 'Modify my Springer Nature account'. For more information please visit please visit www.springernature.com/orcid.

Thank you for the opportunity to review your work.

Sincerely,

Stephanie Houston, PhD
Senior Editor
Nature Immunology

Reviewers' Comments:

Reviewer #1:

Remarks to the Author:

This is an interesting manuscript that vividly documents the formation of filamentous (F)-actin-containing filopodia during the process of pyroptosis. The authors propose that this is important because F-actin has previously been shown to be recognized by the CLEC9A receptor expressed on cross-presenting DCs. The implication is that production of F-actin during pyroptosis promotes adaptive immune responses. This would be highly impactful if shown. However, what is actually shown (rather beautifully, I might add) is that pyroptotic cells produce F-actin-containing structures, and that RAW macrophages activate CLEC9A in response to these cells. In fact, the manuscript is fairly narrow in its scope: there are no experiments examining antigen presentation, or any in vivo adaptive immune response. Nor is it clear whether the F-actin structures shown in the images are important for CLEC9A activation, though I tend to think this is a reasonable presumption. There are no experiments showing that CLEC9A is required for the adaptive immune response to a pyroptotic cell; indeed it is reasonable to think that pyroptotic cells might engage DCs through many different pathways.

In general, the manuscript oversells the novelty and importance of the observations. For example, the abstract proposes that pyroptotic cells "build F-actin structures prior to pyroptotic cell rupture in order to mark the pyroptotic corpse for antigen sampling by dendritic cells" but this is not actually shown in the manuscript. Pyroptotic cells are shown to activate CLEC9-dependent Syk activation (in RAW macrophages), but antigen uptake is not shown, and nor is it shown that the CLEC9 activation seen with a pyroptotic cell requires the formation of F-actin. I would favor limiting the abstract to what is shown and not speculating beyond the data.

Some additional questions and concerns are as follows:

To what extent are these F-actin structures a consequence of the pyroptotic macrophages growing on plastic, to which they strongly adhere? Is there evidence similar structures are formed in soft tissues or suspension culture, and if so, could these structures also activate DCs? If this is not addressable experimentally, an explicit caveat should be discussed.

Necrotic cells have previously been shown to activate CLEC9 to promote antigen uptake and cross-priming of CTLs (Zelenay et al, 2012). It is surprising this paper is not cited and discussed as it seems highly relevant. Indeed, this and other prior work also suggests that F-actin formation during cell death is not a special function of pyroptosis, as it also occurs during necrosis. The authors could discuss whether perhaps the formation of F-actin is a default feature of a dying cell that instead of being induced specifically during pyroptosis is instead specifically prevented during apoptosis.

Minor comments:

A more accurate title would be "Pyroptotic cells assemble F-actin and can activate CLEC9A"

Abstract: Avoid personification of dendritic cells. They don't "deliberately" do anything, and indeed, it is speculative to make assumptions about why cells assemble F-actin after cell death. Indeed, there may be multiple functions, including trapping microbes in the corpse, and nothing in the manuscript addresses "why" the observed phenomena occur.

Abstract: avoid priority claims ("first report", "new alarmin" – F-actin is not a new alarmin); the introduction and final paragraph also contains unnecessary priority claims ("This is the first report" etc.)

Introduction: eicosanoids could be mentioned as another non-cytokine effector released from pyroptotic cells, as shown by several groups. Eicosanoid release during pyroptosis also requires a Ca⁺⁺ influx, so there are interesting parallels to what is reported here.

While high concentrations of glycine does block NINJ1, it may have other effects, so some caution in interpreting the glycine experiment is warranted. This could be addressed either by testing a NINJ1 KO, or by tempering the language concluding that NINJ1 is not required, e.g., "appears to be dispensable" instead of "is dispensable".

It would be helpful to repeat the experiment in Fig. 5A using cDC1 from Clec9a^{-/-} mice in order to demonstrate that the activation seen is Clec9a-dependent.

Reviewer #2:

Remarks to the Author:

Holley, Schroder and colleagues investigate how pyroptotic macrophages communicate from the pyroptotic state to the next layer of the immune response. The study shows that inflammasomes trigger the formation of filopodia, F-actin-rich structures, on the cell membrane before undergoing pyroptotic cell rupture. The authors demonstrate that these filopodia persist on the pyroptotic cells and 'mark' that these cells had undergone pyroptosis. The F-actin structures activate the dendritic cell receptor CLEC9A, which serves a key role in mediating adaptive immunity. Contrary to the common belief that cells disintegrate upon death and are merely dead structures that will eventually silently disappear, the study suggests that inflammasome-activated cells intentionally construct F-actin structures before pyroptotic cell rupture and that this process serves as a signal for antigen sampling by dendritic cells. This research uncovers a new aspect of pyroptotic cell death but also highlights a mechanism by which inflammasomes contribute to the interaction between innate and adaptive immunity. The authors couple high-quality imaging with a genetics-based disentanglement of required cell death machinery to examine the function of F-actin filopodia in linking the innate and adaptive immune response.

The manuscript presents unexpected novel links between pyroptosis and adaptive immune responses that goes beyond the production of pro-inflammatory cytokine activation. Overall, these new aspects of inflammasome signaling are exciting and worth reporting to broader immunological readership.

There are a few concerns that the authors should consider:

1. The first is the link to Clec9a and the follow-on immune response. After reading the title I was expecting a significant insight into how the F-actin protrusions link to adaptive immune responses, but Figure 5 contains only little data to support a link between the protrusions and later immune responses. This figure will benefit from additional experiments that may produce stronger data to support their conclusion.
2. The second area of novelty relates to the formation of the protrusions themselves. The data clearly demonstrate that pore-formation triggers the creation of the filopodia. Given the dependence on calcium, we presume this is not specific to the GSDMD pore, so the question about specificity (GSDME/A/B) arises. Is this process only seen in pyroptotic cells? Please see further comments below.
3. In a similar line of thinking, what is the contribution of inflammasome priming for these processes? Is it essential for the generation of the observed F-actin structures? If so, does priming license other forms of death to become F-actin protruding? These questions are largely unexplored but may further highlight the specificity to pyroptotic pores, or reveal a broader applicability for this interesting feature of cell death.

Minor Comments:

Figure 5, Clec9a and beyond

- The largest drawback to this manuscript is the link to the adaptive immune system via Clec9a. More experiments are needed to support the conclusions of this figure. An experiment where DCs are presented with corpses derived from an infection that triggers pyroptosis onto DCs and subsequently assessing antigen presentation, would be a great context to look at both F-actin filopodia (infect macs +/- EGTA to block filopodia) and Clec9a (infect the Y7F cells).
- Figure 5 could benefit from some additional labelling making it clear that these are DCs. I was confused by the left label that indicates macrophage.
- Have the authors confirmed that the macrophage corpses do not contain p-SYK that may be taken up by the DCs? An additional control showing specificity to pyroptotic cells and the F-actin filopodia should also be included in this experiment. E.g. incubating DCs with corpses from LPS+NIG+EGTA, and/or apoptotic corpses.
- If the pSYK antibody works for western blot, the authors could consider this method as additional evidence to support the difference claimed in Figure 5B.
- In Figure 5B, some of the graphical data is performed with N>100, where each N represents a single cell, and the total number of cells is pooled from ~3 independent experimental repeats. Specifically with respect to Figure 5B, I do not think there is a real difference here. If the authors averaged the cell data (which may be considered technical replicates) and then performed a non-parametric test on the 3 independent experiments, there would be only 3 points per condition and no significant difference.

Protrusion formation

- Inflammasomes are indeed crucial orchestrators of the innate immune response, yet they and their associated inflammatory output are often associated with disease not in the context of an infection. How critical is priming to the formation of the protrusions? Do they still form in the context of AIM2 sensing of self-DNA, or perhaps in cells with autoactivating mutations in inflammasome pathways, etc.

- Similarly, do these F-actin filopodia form occur if you activate apoptosis, necroptosis (thought by some to have also developed as a viral defense mechanism) or ferroptosis in the presence of LPS?
- Is the formation of the actin filaments an active process, or does pyroptosis cause a failure in actin depolymerisation?
- Is there any precedent for calcium regulation of the actin network?
- How stable are these F-actin- rich corpses? How long do they persist (in vitro)? Do they absolutely require clearance by other cells, or do they eventually break down?
- Do F-actin rich filopodia form downstream of GSDME pore formation?
- Do these structures only form in macrophages? Have the authors observed similar structures in other pyroptosis competent immune (neutrophils) or non-immune (keratinocytes) cells?

Other

- Have the authors considered an alternative quantification of cell protrusions for Figure 4B? Since EGTA reduces cell corpse area, a more accurate method of quantifying the impact of EGTA on cell protrusions may be to graph: # projections per corpse area (per cell).
- Considerable emphasis is placed on the effect the F-actin filopodia have on corpse size, yet the significance of this observation is unclear. Whether the size of an F-actin rich corpses influences the overall immune response is not known and whether it is corpse size or the simple presence of F-actin filopodia that contributes to a response is not distinguished.

Minor point regarding the introduction:

“N-terminal pore-forming fragment^{6, 7} that permeabilizes the plasma membrane (PM) and activates NINJ1 to trigger plasma membrane rupture (PMR)⁸ in a multi-step pathway of lytic cell death called pyroptosis⁹.”

This statement suggests the N-terminal fragment directly activates NINJ1 and minor rewording would more accurately reflect the uncertain link between the two proteins.

Reviewer #3:

Remarks to the Author:

In this manuscript entitled ‘Pyroptotic F-actin remodelling engages CLEC9A, a bridge between innate and adaptive immunity’, the authors found that during pyroptosis, after GSDMD pore formation and before NINJ1-mediated PM rupture, cells generate F-actin enriched ‘filopodia’-like structures. Formation of filopodia depends on GSDMD pore formation and Ca²⁺ influx. Those filopodia structures seem to remain after the cells burst and activate dendritic cells by the F-actin receptor CLEC9A, also known as DNGR-1. Overall the manuscript suggests a novel phenomenon and provided high resolution imaging data. Many of the data were purely descriptive and did not even include any quantitative or statistical analysis (i.e. Fig. 1, Fig. 2) – since there were similar structures seen in non-pyroptotic cells this is very important. Moreover the images sometimes lacked controls not undergoing pyroptosis. Moreover, some key questions need to be addressed. The most important are: Is this phenomenon specific to pyroptosis or does it occur any time the plasma membrane loses its integrity or with other pore-forming proteins that do not activate programmed cell death? Does it occur in other cell types such as epithelial cells undergoing pyroptosis? Does the phenomenon have any functional significance? Does it occur in human cells as well as in mouse systems? How does CLEC9A on the outside of a cell bind to F-actin inside another cell? Do the filopodia form near GSDMD pores?

Major points:

1. As the authors also cited, there are publications in the past to investigate the involvement of cytoskeleton during pyroptosis. However, the previous report looking at actin during pyroptosis did not observe such filopodia. The authors failed to discuss the differences or the reasons behind the different phenomenon observed. Meanwhile, many publications have shown the ballooning phenotype of pyroptosis, but in this manuscript, those were not prominent. Those major discrepancies are important and clearly need to be addressed.
2. Critical controls are missing in several experiments. For example, in Figure 2, scanning EM is only performed with cells long after PM rupture. It is important to analyze cells without treatment, before GSDMD pore formation, after pore formation but before cell rupture, and after rupture. Ideally including kinetics and an apoptosis control and controls with bacterial pore-forming proteins will need to be done to conclude that filopodia formation is specific to pyroptosis. In another publication <https://link.springer.com/article/10.1007/s12017-019-08586-y>, scanning EM is carried out with different stages of pyroptosis and the results are not completely consistent with this manuscript.
3. The author proposed the model that GSDMD pore mediate Ca²⁺ influx which activate Myosin X to trigger filopodia formation. This is a novel point but was not analyzed clearly. If filopodia formation is important for the F-actin rich cell corpse maintaining, and following DC activation, then instead of using casp mutant or gsdmd mutant (which blocks everything), they should inhibit actin assembly/Myosin X (inhibitors available and/or KD or KO) to see the filopodia, cell corpse, DC activation. In the manuscript, Extended Fig 4b, actin polymerization is tested but the figure is not clear and there aren't statistics.
4. In Figure 5A,B, the difference is very subtle. It's good to test SYK phosphorylation with immunoblotting and test CLEC9A-SYK complex formation by CoIP. Also for Fig 5C,D, clec9a KO cells should be included as controls.
5. The paper refers to the “basal” membrane of non-polarized hematopoietic cells. I don't think this usage is correct. This should be studied in polarized epithelial cells. The protruding part of migrating cells (the site of the lamellopodia) are usually called the apical surface. Do the F-actin structures form when the cells are not adherent?
6. It is critical for a high impact publication to show that this phenomenon has functional and in vivo significance.
7. I believe that MCC950 is a covalent binder to NLRP3 that is Cys-reactive. As such it is unlikely to be completely “specific”.
8. What happens if GSDMD is cleaved but pore formation is blocked (with DMF, DSF)? Does it affect the formation of these processes? Are they formed in “hyperactivated” myeloid cells that form pores but survive?
9. Is filopodia formation blocked by chelating intracellular calcium?

Minor points

1. Sir-Actin is a cell permeable actin dye, why does actin (such as in Fig 1a/b, magenta) only show up when cell is ruptured?
2. Typo 'Fig 4H' should be 'Fig 4F' in the text.
3. Reference 16 and 18 are the same paper

Version 1:

Decision Letter:

Our ref: NI-LE36691A

23rd Jul 2024

Dear Dr. Schroder,

Thank you for submitting your revised manuscript "Pyroptotic corpses are crowned with F-actin-rich filopodia that engage CLEC9A signalling in incoming dendritic cells" (NI-LE36691A). It has now been seen by the original referees and their comments are below. The reviewers find that the paper has improved in revision, and therefore we'll be happy in principle to publish it in Nature Immunology, pending minor revisions to satisfy the referees' final requests and to comply with our editorial and formatting guidelines.

We will now perform detailed checks on your paper and will send you a checklist detailing our editorial and formatting requirements in about a week. Please do not upload the final materials and make any revisions until you receive this additional information from us.

If you had not uploaded a Word file for the current version of the manuscript, we will need one before beginning the editing process; please email that to immunology@us.nature.com at your earliest convenience.

Thank you again for your interest in Nature Immunology Please do not hesitate to contact me if you have any questions.

Sincerely,

Jamie D K Wilson, D.Phil
Chief Editor
For:

Stephanie Houston, PhD
Senior Editor
Nature Immunology

Reviewer #1 (Remarks to the Author):

I'd like to congratulate the authors on their highly responsive revision. My previous concerns have been fully addressed. My only minor remaining comment is that, despite a common misconception, NLRP3 does not stand for "Nod-like receptor protein 3" (as indicated on p.1 of the manuscript). The correct abbreviation is Nucleotide-binding domain and Leucine-rich Repeat and Pyrin domain containing 3". Please see PMC2630772 for the official nomenclature of NLR proteins.

Reviewer #2 (Remarks to the Author):

The authors have substantially reworked the manuscript in response to the reviewers comments. The critiques of the reviewers were mostly addressed by adding new and supportive data. I am happy to support publication of the manuscript in its current form.

We thank the reviewers for their thorough and insightful reviews of our manuscript, for their valuable suggestions and their positive comments. In this revised manuscript, we have addressed points raised by the reviewers, and we now include 3 new movies and 53 new data panels (29 in main Figures and 24 in Extended Figures) substantiating and extending our original findings. Major changes to our study, as suggested by Reviewers, are highlighted in yellow in the manuscript documents.

These new data add substantial new mechanistic understanding of: (1) the processes controlling GSDMD-induced filopodial assembly in inflammasome-signalling cells (Fig. 2, Extended Figs 2-7); (2) the specificity of this response amongst lytic and apoptotic modes of programmed cell death (Fig. 3, Extended Figs 6-7); and (3) the mechanisms by which incoming dendritic cells recognise these F-actin-rich structures (Fig. 5, Extended Figs 9-10).

Our revised manuscript also now demonstrates the wide variety of circumstances under which inflammasome-signalling cells assemble filopodia. We show that inflammasome-signalling cells generate filopodia in both (Fig. 1-2 and Extended Figures 1-5): (1) human and mouse cells; (2) immune and non-immune cells; (3) cells cultured in suspension or under adherence; (4) cells challenged *in vitro* and *in vivo*; (5) cells undergoing pyroptotic and sub-lytic inflammasome signalling.

We feel that in addressing the Reviewers' points we have significantly enhanced the impact and clarity of our study, and we thank the Reviewers for their constructive input. Please find below a point-by-point reply to each comment.

Reviewer #1

This is an interesting manuscript that vividly documents the formation of filamentous (F)-actin-containing filopodia during the process of pyroptosis. The authors propose that this is important because F-actin has previously been shown to be recognized by the CLEC9A receptor expressed on cross-presenting DCs. The implication is that production of F-actin during pyroptosis promotes adaptive immune responses. This would be highly impactful if shown. However, what is actually shown (rather beautifully, I might add) is that pyroptotic cells produce F-actin-containing structures, and that RAW macrophages activate CLEC9A in response to these cells. We thank the Reviewer for their positive comments and generally agree with the Reviewer's summation of the data in our original submission. Given that cells undergoing pyroptosis have never yet been described to assemble filopodia, much of our original manuscript was dedicated to characterising these structures in detail, and elucidating the inflammasome signalling pathways that generate these structures. While we respectfully disagree with the implication that we did not demonstrate that cDC1 cells endogenously expressing CLEC9A respond to pyroptotic corpses (this was shown in original Fig 5A), we appreciate the Reviewer's point that we did not assess the CLEC9A-dependency of SYK phosphorylation in the original submission. We have since established a colony of *Clec9a* knockout mice, and have used WT versus *Clec9a*-deficient dendritic cells to show that pyroptotic corpses induce CLEC9A-dependent SYK phosphorylation in cDC1 cells (see **new Fig 5b**). We have further mapped the requirements for macrophage inflammasome signalling to trigger signalling by the CLEC9A-SYK axis in incoming cDC1 cells. We now show that GSDMD, extracellular calcium and NINJ1 are all required for macrophages to induce CLEC9A-SYK signalling in cDC1s (see **new Fig 5c**). This suggests that GSDMD pores allow calcium influx to drive filopodial assembly, while NINJ1-driven rupture is required to expose intracellular F-actin from macrophages for recognition by dendritic cell CLEC9A. Our revised manuscript thereby shows that the assembly of F-actin-rich projections during pyroptosis allows pyroptotic corpses to activate CLEC9A on incoming cDC1 cells.

In fact, the manuscript is fairly narrow in its scope: there are no experiments examining antigen presentation, or any *in vivo* adaptive immune response. A large body of literature documents the capacity of CLEC9A to facilitate cDC1 sampling and presentation of dead cell antigens for adaptive immune responses *in vitro* and *in vivo* (e.g. manuscript refs 1-2, 43-48). We thus felt it appropriate to use the NI short letter format to document the key novelty of our study – that pyroptotic cells assemble F-actin-rich structures, thereby ensuring that pyroptotic corpses are a rich source of CLEC9A ligand, and are therefore immunogenic. Our revised manuscript now additionally demonstrates that GSDMD drives the assembly of F-actin-rich filopodia *in vivo* (**new Fig. 1I-J**) and that signalling by the GSDMD-calcium axis in pyroptotic macrophages is sufficient to upregulate cDC1 expression of CD80 and CD86 co-stimulatory molecules and MHCII (**new Fig. 5d-f**).

Nor is it clear whether the F-actin structures shown in the images are important for CLEC9A activation, though I tend to think this is a reasonable presumption. We now address this directly, by blocking filopodial assembly in pyroptotic cells and showing this prevents the pyroptotic corpse from triggering CLEC9A-SYK signalling in incoming cDC1 cells (**new Fig. 5c**). We further show that NINJ1-driven macrophage rupture is not required for filopodial assembly (**new Fig. 2f-g**) but is required for incoming cDC1 cells to access the F-actin within pyroptotic corpses (**new Fig. 5c**).

There are no experiments showing that CLEC9A is required for the adaptive immune response to a pyroptotic cell; indeed it is reasonable to think that pyroptotic cells might engage DCs through many different pathways. We agree that pyroptotic cells likely engage DCs in many different ways. Our revised manuscript shows that signalling by the GSDMD-calcium axis in pyroptotic macrophages is sufficient to upregulate cDC1 expression of CD80 and CD86 co-stimulatory molecules and MHCII (**new Fig. 5d**). Given that CLEC9A is well known to facilitate cDC1 sampling and presentation of dead cell antigens for adaptive immune responses, we felt it appropriate to use the NI short letter format to document the key novelty of our study – that pyroptotic cells assemble F-actin-rich structures, thereby ensuring that pyroptotic corpses are a rich source of CLEC9A ligand, and are therefore immunogenic (unlike apoptosis, which is generally immunologically silent). Experiments investigating the capacity of pyroptotic corpses to influence lymphocyte function are beyond the scope of this short-format report.

In general, the manuscript oversells the novelty and importance of the observations. For example, the abstract proposes that pyroptotic cells “build F-actin structures prior to pyroptotic cell rupture in order to mark the pyroptotic corpse for antigen sampling by dendritic cells” but this is not actually shown in the manuscript. Pyroptotic cells are shown to activate CLEC9-dependent Syk activation (in RAW macrophages), but antigen uptake is not shown, and nor is it shown that the CLEC9 activation seen with a pyroptotic cell requires the formation of F-actin. I would favor limiting the abstract to what is shown and not speculating beyond the data. We have thoroughly revised the manuscript text (including title and abstract) to address this point. We have also added new data (53 new data panels) to further substantiate and extend upon our original findings, including new data showing that pyroptotic corpses trigger CLEC9A-SYK signalling in incoming cDC1 cells (**new Fig. 5b-c**).

Some additional questions and concerns are as follows:

To what extent are these F-actin structures a consequence of the pyroptotic macrophages growing on plastic, to which they strongly adhere? Is there evidence similar structures are formed in soft tissues or suspension culture, and if so, could these structures also activate DCs? If this is not addressable experimentally, an explicit caveat should be discussed. We were also curious about this, and our revised manuscript directly addresses this *in vitro* and *in vivo*. To determine whether inflammasome-driven filopodial assembly required microenvironmental mechanical cues or adherence, we performed live imaging on macrophages cultured in suspension. Indeed, suspended macrophages undergoing nigericin-induced pyroptosis assembled F-actin-rich filopodia (**new Fig. 1h, new**

Extended Fig. 3) similar to adherent macrophages. We also challenged mice with LPS to induce CASP11-GSDMD signalling, and found that *in vivo*-challenged cells similarly exhibit GSDMD-dependent filopodia (**new Fig. 1i-j**). Thus, cell adhesion is not required for macrophages to assemble pyroptotic filopodia, and these are not a consequence of culturing macrophages on plastic.

Necrotic cells have previously been shown to activate CLEC9 to promote antigen uptake and cross-priming of CTLs (Zelenay et al, 2012). It is surprising this paper is not cited and discussed as it seems highly relevant. Indeed, this and other prior work also suggests that F-actin formation during cell death is not a special function of pyroptosis, as it also occurs during necrosis. CLEC9A is indeed documented to sample necrotic cells to mediate antigen uptake and cross-priming of CTLs. Most studies of CLEC9A-dependent dead cell sampling use cells killed by mechanical damage (e.g. freeze thaw) or cells that have undergone secondary necrosis (e.g. Zelenay et al, 2012). We now cite the latter study, which we agree is relevant. In all of these published reports of necrosis-induced CLEC9A activation, necrotic membrane damage allows CLEC9A to access intracellular F-actin from pre-existing F-actin-rich cell structures. To our knowledge, necrotic cells have never been reported to actively assemble F-actin rich structures immediately before cell death, as we show here for pyroptosis and necroptosis. That is, *necrotic cells* (e.g. in Zelenay et al, 2012) **do not assemble filopodia immediately prior to membrane rupture, while pyroptotic and necroptotic cells do**. This is the key novelty of our study's findings within the CLEC9A literature.

The authors could discuss whether perhaps the formation of F-actin is a default feature of a dying cell that instead of being induced specifically during pyroptosis is instead specifically prevented during apoptosis. This is an interesting point. Our revision incorporates several new experiments examining distinct cell death modes and their capacity to induce cell corpses rich in filopodia, and shows that:

- (1) macrophage expression of the pore-forming fragment of GSDMD (GSDMD-NT) is sufficient to induce calcium-dependent filopodial assembly (**new Fig. 2h-i, new Fig. 3b-d**);
 - (2) other gasdermins can similarly generate corpses crowned with filopodia (**new Fig. 3e-g**);
 - (3) active MLKL is sufficient to induce filopodial assembly in necroptotic cells (**new Fig. 3h-j**); and
 - (4) diverse inducers of apoptotic cell death do not trigger the assembly of filopodia (**new Fig. 3k-n**).
- Given that *Gsdmd* deficiency diverts inflammasome-signalling cells from pyroptosis to apoptosis, we had the opportunity to directly compare pyroptotic versus apoptotic death in cells stimulated identically, by comparing the nigericin responses of wild-type versus *Gsdmd*-deficient macrophages. We now show that while nigericin-induced pyroptotic corpses are crowned by filopodia, nigericin-stimulated apoptotic cells do not exhibit filopodia, and their shrinking cell bodies instead leave behind retraction fibres (**Fig. 2a, Fig. 3m**). Such nigericin-induced apoptotic cells do not induce CLEC9A-SYK signalling in incoming cDC1 cells (**new Fig. 5c**).

In all, these studies lead us to conclude that the assembly of F-actin-rich filopodia is not a default feature of a dying cell, but is rather an active process in the biological sense of requiring cellular regulation – filopodial assembly is driven by cell effectors (gasdermins, MLKL) that permeabilise cell membranes during lytic cell death.

To entirely uncouple GSDMD-induced filopodia from cell death, we collaborated with the Kagan lab to examine filopodial assembly in macrophages with sub-lytic inflammasome signalling (for which they have coined the term cell 'hyperactivation'). Stimulation with the hyperactivation stimulus, PGPC, generated macrophages and dendritic cells with a spectacular number of MYO10-capped projections per cell, most of which were highly branched (**new Fig. 1e-f, new Extended Fig. 2a-f**). Given that PGPC-stimulated BMDM and BMDC remain viable (manuscript refs 14, 33), this further indicates that F-actin assembly is not a default feature of a dying cell. Our observation that PGPC induces hundreds more projections/cell than nigericin may indeed suggest the opposite – that cell death may halt the assembly of these F-actin-rich projections.

Minor comments:

A more accurate title would be “Pyroptotic cells assemble F-actin and can activate CLEC9A”. We have revised our title for greater accuracy in reporting the findings of our revised manuscript (“*Pyroptotic corpses are crowned with F-actin-rich filopodia that engage CLEC9A signalling in incoming dendritic cells*”).

Abstract: Avoid personification of dendritic cells. They don’t “deliberately” do anything, and indeed, it is speculative to make assumptions about why cells assemble F-actin after cell death. Indeed, there may be multiple functions, including trapping microbes in the corpse, and nothing in the manuscript addresses “why” the observed phenomena occur. We have amended the text as suggested.

Abstract: avoid priority claims (“first report”, “new alarmin” – F-actin is not a new alarmin); the introduction and final paragraph also contains unnecessary priority claims (“This is the first report” etc.). We have amended the text as suggested. We did not intend to suggest F-actin is new to the alarmin literature; our intended meaning was that F-actin-rich filopodia were newly constructed (this is now reworded for greater clarity).

Introduction: eicosanoids could be mentioned as another non-cytokine effector released from pyroptotic cells, as shown by several groups. Eicosanoid release during pyroptosis also requires a Ca⁺⁺ influx, so there are interesting parallels to what is reported here. This is an excellent point, and we now mention eicosanoids in this context in the introduction as suggested.

While high concentrations of glycine does block NINJ1, it may have other effects, so some caution in interpreting the glycine experiment is warranted. This could be addressed either by testing a NINJ1 KO, or by tempering the language concluding that NINJ1 is not required, e.g., “appears to be dispensable” instead of “is dispensable”. We agree. We have now established a *Ninj1*-deficient mouse colony, and have replaced the former glycine results with experiments from wild-type versus *Ninj1* knockout macrophages. We now show that NINJ1-driven macrophage rupture is not required for filopodial assembly (new Fig. 2f-g) but is required for incoming cDC1 cells to access the F-actin within pyroptotic corpses (new Fig. 5c).

It would be helpful to repeat the experiment in Fig. 5A using cDC1 from *Clec9a*^{-/-} mice in order to demonstrate that the activation seen is *Clec9a*-dependent. We agree. Using WT versus *Clec9a*-deficient dendritic cells, we now show that pyroptotic corpses induce CLEC9A-dependent SYK phosphorylation in cDC1 cells (see new Fig 5b,c). We further show that apoptotic macrophages (nigericin-stimulated *Gsdmd*^{-/-} macrophages) do not engage CLEC9A-SYK signalling in incoming cDC1 cells (see new Fig 5c).

Reviewer #2

Holley, Schroder and colleagues investigate how pyroptotic macrophages communicate from the pyroptotic state to the next layer of the immune response. The study shows that inflammasomes trigger the formation of filopodia, F-actin-rich structures, on the cell membrane before undergoing pyroptotic cell rupture. The authors demonstrate that these filopodia persist on the pyroptotic cells and ‘mark’ that these cells had undergone pyroptosis. The F-actin structures activate the dendritic cell receptor CLEC9A, which serves a key role in mediating adaptive immunity. Contrary to the common belief that cells disintegrate upon death and are merely dead structures that will eventually silently disappear, the study suggests that inflammasome-activated cells intentionally construct F-actin structures before pyroptotic cell rupture and that this process serves as a signal for antigen sampling by dendritic cells. This research uncovers a new aspect of pyroptotic cell death but also highlights a mechanism by which inflammasomes contribute to the interaction between innate and adaptive

immunity. The authors couple high-quality imaging with a genetics-based disentanglement of required cell death machinery to examine the function of F-actin filopodia in linking the innate and adaptive immune response.

The manuscript presents unexpected novel links between pyroptosis and adaptive immune responses that goes beyond the production of pro-inflammatory cytokine activation. Overall, these new aspects of inflammasome signaling are exciting and worth reporting to broader immunological readership. We thank the Reviewer for their positive comments and their support for our study. We also think our findings are exciting, and present new and unexpected avenues for future research!

There are a few concerns that the authors should consider:

1. The first is the link to Clec9a and the follow-on immune response. After reading the title I was expecting a significant insight into how the F-actin protrusions link to adaptive immune responses, but Figure 5 contains only little data to support a link between the protrusions and later immune responses. This figure will benefit from additional experiments that may produce stronger data to support their conclusion. A large body of literature documents the capacity of CLEC9A to facilitate cDC1 sampling and presentation of dead cell antigens for adaptive immune responses *in vitro* and *in vivo* (e.g. manuscript refs 1-2, 43-48). We thus felt it appropriate to use the NI short letter format to document the key novelty of our study – that pyroptotic cells assemble F-actin-rich structures, thereby ensuring that pyroptotic corpses are a rich source of CLEC9A ligand, and are therefore immunogenic. Our revised manuscript now additionally demonstrates that GSDMD drives the assembly of F-actin-rich filopodia *in vivo* (**new Fig. 1i-j**) and that signalling by the GSDMD-calcium axis in pyroptotic macrophages is sufficient to upregulate cDC1 expression of CD80 and CD86 co-stimulatory molecules and MHCII (**new Fig. 5d-f**). Using WT versus *Clec9a*-deficient dendritic cells, we also show that pyroptotic corpses induce CLEC9A-dependent SYK phosphorylation in cDC1 cells (see **new Fig 5b**). We have further mapped the requirements for macrophage inflammasome signalling to trigger signalling by the CLEC9A-SYK axis in incoming cDC1 cells; GSDMD, extracellular calcium and NINJ1 are all required for macrophages to induce CLEC9A-SYK signalling in cDC1s (see **new Fig 5c**). This suggests that GSDMD pores allow calcium influx to drive filopodial assembly, while NINJ1-driven rupture is required to expose intracellular F-actin from macrophages for recognition by dendritic cell CLEC9A. Our revised manuscript thereby shows that the assembly of F-actin-rich projections during pyroptosis allows pyroptotic corpses to activate CLEC9A on incoming cDC1 cells.

Experiments investigating the capacity of pyroptotic corpses to influence antigen presentation, lymphocyte function and resultant adaptive immune responses are beyond the scope of this short-format report. To ensure the reader is not misled in their expectations, we have thoroughly reworked the manuscript title and text to ensure the focus of this short report remains on pyroptotic signalling events and how this serves to engage CLEC9A.

2. The second area of novelty relates to the formation of the protrusions themselves. The data clearly demonstrate that pore-formation triggers the creation of the filopodia. Given the dependence on calcium, we presume this is not specific to the GSDMD pore, so the question about specificity (GSDME/A/B) arises. Is this process only seen in pyroptotic cells? Please see further comments below. This is an excellent question and one that also intrigued us. Our manuscript revision incorporates several new experiments examining distinct cell death modes and their capacity to induce cell corpses rich in filopodia, and shows that:

- (1) macrophage expression of the pore-forming fragment of GSDMD (GSDMD-NT) is not only necessary, but is also sufficient to induce calcium-dependent filopodial assembly (**new Fig. 2h-i, new Fig. 3b-d**);
- (2) other gasdermins can similarly generate corpses crowned with filopodia (**new Fig. 3e-g**);
- (3) active MLKL is sufficient to induce filopodial assembly in necroptotic cells (**new Fig. 3h-j**); and

(4) diverse inducers of apoptotic cell death (including inflammasome-induced apoptosis) do not trigger the assembly of filopodia (**new Fig. 3k-n**), and cells undergoing inflammasome-driven apoptosis do not activate CLEC9A on cDC1 cells (**new Fig. 3c**, *Gsdmd*-deficient macrophages).

We also provide new data expanding upon calcium dynamics during pyroptotic signalling. We now show that:

- (1) nigericin induces Ca^{2+} entry into cells, which on average peaks around 2 mins before the cell becomes permeable (**new Fig. 2c**);
- (2) individual cells often show several peaks of elevated intracellular Ca^{2+} in the 10 mins before the cell becomes permeable; this perhaps reflects discrete events of calcium influx through newly formed individual GSDMD pores (**new Extended Fig. 5a**; **new Video 6**);
- (3) nigericin-induced Ca^{2+} cell entry precedes the assembly of filopodia (**new Video 6**); and
- (4) nigericin-induced Ca^{2+} entry into the cell requires GSDMD (**new Fig. 2c**, **new Videos 6 and 7**).

3. In a similar line of thinking, what is the contribution of inflammasome priming for these processes? Is it essential for the generation of the observed F-actin structures? If so, does priming license other forms of death to become F-actin protruding? These questions are largely unexplored but may further highlight the specificity to pyroptotic pores, or reveal a broader applicability for this interesting feature of cell death. We did not explicitly discuss this in our original manuscript because of space constraints, but this was indirectly assessed in our original manuscript. Original Extended Fig 1c-d (**now Extended Fig. 1f-g**) showed that **unprimed** macrophages stimulated with PA-FlaTox to activate the NLRC4 inflammasome do indeed assemble filopodia – this indicates that inflammasome priming is not required for filopodial assembly. Given that *Gsdmd* deficiency diverts inflammasome-signalling cells from pyroptosis to apoptosis, we had the opportunity to directly compare pyroptotic versus apoptotic death in **LPS-primed** cells stimulated identically, by comparing the nigericin responses of wild-type versus *Gsdmd*-deficient macrophages. Our original submission showed that while nigericin-induced pyroptotic corpses are crowned by filopodia, nigericin-stimulated apoptotic cells do not exhibit filopodia, and their shrinking cell bodies instead leave behind retraction fibres (**Fig. 2a**, **Fig. 3m**). Thus, cell priming does not license filopodial assembly during inflammasome-induced apoptosis.

We now formally test the influence of cell priming on pyroptotic filopodia in **new Fig. 3b**, which shows that macrophage expression of the pore-forming fragment of GSDMD (GSDMD-NT) is sufficient to induce filopodia, and this is not affected by LPS priming (**new Fig. 3b-d**). We conclude from these studies that while priming promotes signalling by some inflammasomes as is well documented, it does not influence GSDMD-induced filopodia.

Minor Comments:

Figure 5, Clec9a and beyond

- The largest drawback to this manuscript is the link to the adaptive immune system via Clec9a. More experiments are needed to support the conclusions of this figure. An experiment where DCs are presented with corpses derived from an infection that triggers pyroptosis onto DCs and subsequently assessing antigen presentation, would be a great context to look at both F-actin filopodia (infect macs +/- EGTA to block filopodia) and Clec9a (infect the Y7F cells). Please refer to our response to Question 1 from this Reviewer, above.
- Figure 5 could benefit from some additional labelling making it clear that these are DCs. I was confused by the left label that indicates macrophage. We apologise for this lack of clarity, which we have addressed in new Fig 5 (which includes diagrams explaining co-culture experiments).

- Have the authors confirmed that the macrophage corpses do not contain p-SYK that may be taken up by the DCs? Indeed, we previously stained pyroptotic macrophages (cultured alone, without dendritic cells), and they did not contain p-SYK (not shown). Since our original submission, we have developed a flow cytometry technique that allows us to quantify phospho-SYK in individual cells, specifically in the cDC1 compartment (see **new Extended Fig. 10a**). We have replaced the original p-SYK data with new data (**Fig. 5b-c**) that quantifies phospho-SYK only within cDC1 cells (macrophages are excluded from phospho-SYK analyses), and show SYK phosphorylation is CLEC9A-dependent. An additional control showing specificity to pyroptotic cells and the F-actin filopodia should also be included in this experiment. E.g. incubating DCs with corpses from LPS+NIG+EGTA, and/or apoptotic corpses. We have adopted this excellent suggestion. We now show that while EGTA does not affect nigericin-induced macrophage lysis, it does block the capacity of pyroptotic corpses to trigger CLEC9A-SYK signalling in incoming cDC1 cells (**new Fig. 5c**). In this experiment, we also expose *Gsdmd*-deficient BMDM to nigericin; we and others have shown that this induces apoptosis as a back-up mechanism of cell death (e.g. **Fig. 3m**). We show that nigericin-stimulated apoptotic macrophages (*Gsdmd*^{-/-}) are unable to induce CLEC9A-SYK signalling in incoming cDC1 cells (**new Fig. 5c**). Thus pyroptotic, but not apoptotic, macrophages engage CLEC9A signalling in cDC1 cells.

- If the pSYK antibody works for western blot, the authors could consider this method as additional evidence to support the difference claimed in Figure 5B. We tried this, but think our new flow cytometry method (**new Extended Fig. 10a**) is more robust as it allows us to quantify phospho-SYK in individual cells, specifically in the cDC1 population (which is not possible by western blot of lysates of co-cultured cells).

- In Figure 5B, some of the graphical data is performed with N>100, where each N represents a single cell, and the total number of cells is pooled from ~3 independent experimental repeats. Specifically with respect to Figure 5B, I do not think there is a real difference here. If the authors averaged the cell data (which may be considered technical replicates) and then performed a non-parametric test on the 3 independent experiments, there would be only 3 points per condition and no significant difference. We removed our original Fig 5b data. Our new method allows us to quantify SYK phosphorylation in individual cells within a sample to give us robust sample means over technical replicates within a biological replicate (see **new Fig. 5b-c, new Extended Fig 10a**). **New Fig 5b-c** now shows the mean of each independent biological replicate (circles). These data were verified for normality before parametric statistics were applied.

Protrusion formation

- Inflammasomes are indeed crucial orchestrators of the innate immune response, yet they and their associated inflammatory output are often associated with disease not in the context of an infection. How critical is priming to the formation of the protrusions? Do they still form in the context of AIM2 sensing of self-DNA, or perhaps in cells with autoactivating mutations in inflammasome pathways, etc. Cell priming is not required for GSDMD-induced filopodia; this is explained in detail in our response to Question 3 from this Reviewer, above. Given that cell priming does not affect GSDMD-induced filopodia, we anticipated that related cell lysis signalling pathways would similarly induce filopodia in unprimed cells, which was indeed the case (GSDMA3 and GSDME: **new Fig. 3e-g**; MLKL: **new Fig.3h-j**). We also show that NLRC4-driven inflammasome signalling in unprimed mouse macrophages induces filopodial assembly (**Extended Fig. 1f**).

- Similarly, do these F-actin filopodia form occur if you activate apoptosis, necroptosis (thought by some to have also developed as a viral defense mechanism) or ferroptosis in the presence of LPS? This excellent question is addressed in our response to Question 2 from this Reviewer, above.

- Is the formation of the actin filaments an active process, or does pyroptosis cause a failure in actin depolymerisation? As described in our reply to Question 2 from this Reviewer (above), we now show that calcium flux through GSDMD pores is not only necessary but is also sufficient to induce the assembly of filopodia, and other effectors of lytic cell death (GSDMA3, GSDME, MLKL) can also induce filopodial assembly. Thus, formation of F-actin filaments is an active process in the biological sense of requiring cellular regulation by cell signalling machinery. Nor can the formation of filopodia be explained solely by inhibition of actin disassembly – if filopodia remained on pyroptotic corpses because of a failure of actin depolymerisation to dismantle them, then the filopodia on pyroptotic corpses would need to be present in resting cells before inflammasome signalling (rather than being assembled downstream of GSDMD signalling; **Fig. 2**). Of course, it is possible that decreased actin disassembly also collaborates with actin assembly to promote filopodia, but this cannot be the primary mechanism.
- Is there any precedent for calcium regulation of the actin network? There is extensive evidence that calcium signalling regulates the actin cytoskeleton. Changes in intracellular calcium have been documented to accompany a variety of morphogenetic and homeostatic events associated with changes in cytoskeletal activity. These include epithelial migration and tissue responses to local injury. Increased calcium can influence the cytoskeleton indirectly, for example by altering Rho GTPases or activating contractility via myosin light chain kinase. Calcium can also directly affect actin regulators. For example, several actin cross-linking proteins, such as alpha-actinin, bear calcium-binding domains that allow them to be directly regulated by changes in intracellular calcium. Similarly, gelsolin has been extensively studied as a calcium-activated actin-severing or capping protein. The diversity of these effects highlights the important and complex roles of calcium signalling in regulating actin dynamics in different cellular contexts. For a review of this literature, please refer to PMID: 37025173.
- How stable are these F-actin- rich corpses? How long do they persist (in vitro)? Do they absolutely require clearance by other cells, or do they eventually break down? F-actin-rich corpses appear to be incredibly stable *in vitro* – pyroptotic F-actin crowns remain on tissue culture plastic even after we wash the dish with Triton! We now address this question directly, by generating pyroptotic corpses with nigericin (approx. 100% lysis by 1 h), and then further incubating these corpses for up to 23 h under standard culture conditions. Pyroptotic corpses and their associated filopodial crowns persist throughout this time course (**new Extended Fig. 8e**).
- Do F-actin rich filopodia form downstream of GSDME pore formation? Yes, the pore-forming N-termini of GSDME and GSDMA3 generate pyroptotic corpses crowned with filopodia (**new Fig. 3e-g**).
- Do these structures only form in macrophages? Have the authors observed similar structures in other pyroptosis competent immune (neutrophils) or non-immune (keratinocytes) cells? We have previously shown that while canonical inflammasome activators do not induce neutrophil pyroptosis, CASP11 activation induces spectacular pyroptosis that is accompanied by the extrusion of neutrophil extracellular traps (NETs; manuscript refs 18 and 32). We thus transfected human and mouse neutrophils with LPS and indeed observed inflammasome-induced filopodia; however, because the cells also extrude NETs, the corpse ultrastructure is less clear than for macrophages, with filopodia tangled up in chromatin webs. For this reason, we have not included this data in the manuscript revision. We have instead addressed this question by assessing inflammasome-stimulated dendritic cells (as a second immune cell) and epithelial cells (as a non-immune cell). Upon inflammasome stimulation, dendritic cells and epithelial cells both assemble MYO10-capped filopodia (**new Fig. 1e-g, new Extended Fig. 2a-g**).

Other

- Have the authors considered an alternative quantification of cell protrusions for Figure 4B? Since EGTA reduces cell corpse area, a more accurate method of quantifying the impact of EGTA on cell protrusions may be to graph: # projections per corpse area (per cell). We think the Reviewer has misunderstood our experimental methods, and we apologise for being insufficiently clear on this point in our original submission. As we now clearly state, in our imaging studies we routinely observe two major phenotypes in inflammasome-signalling wild-type macrophages possessing a halo of F-actin-rich projections: (1) Some ASC speck-containing cells appear to have intact cell bodies and nuclei, suggesting that these are yet to undergo PMR at this time point (as exemplified by panel i in **Extended Fig. 1c**), and (2) other cells had lost cortical actin and did not have DAPI-stained nuclei, consistent with them having undergone PMR and loss of cell contents and nucleus (as exemplified by panel ii in **Extended Fig. 1c**). The number of projections per cell can only be counted for intact (type 1) cells. For type 2 cells, it is impossible to assign filopodia from remnant corpse areas to an individual ruptured cell with fixed cell microscopy, because cell rupture generates multiple smaller corpse remnants. Our original Fig 4b measured the area of corpse remnants **after** cell lysis (i.e., type 2 cells) while filopodia were enumerated in intact cells (i.e., type 1 cells). We have reworked this text for greater clarity, and also replaced our original Fig. 4b with **new Fig. 4e**, which quantifies the area of both of these cell phenotypes. We show that EGTA does not affect the (large) area of ASC-speck-positive **intact** macrophages (used for filopodial enumeration in earlier figures), while cell rupture generates a much smaller corpse remnant. It is the latter that is affected by culturing macrophages in EGTA – in the presence of EGTA to block filopodial assembly, the area of remnant corpses that are attached to the dish is greatly diminished.

We now also enumerate projections/cell settings that will be unaffected by corpse area: (1) *Ninj1* knockout cells that do not rapidly burst to form corpses (**new Fig. 2f-g**), and (2) cells undergoing sub-lytic inflammasome signalling (**new Fig. 1e-f, new Extended Fig. 2a-f**).

- Considerable emphasis is placed on the effect the F-actin filopodia have on corpse size, yet the significance of this observation is unclear. Whether the size of an F-actin rich corpses influences the overall immune response is not known and whether it is corpse size or the simple presence of F-actin filopodia that contributes to a response is not distinguished. We agree that the simple presence of F-actin within filopodia is the key immunogen, and the corpse area may be less important. We have reworded this text accordingly.

Minor point regarding the introduction:

“N-terminal pore-forming fragment that permeabilizes the plasma membrane (PM) and activates Ninjurin-1 (NINJ1) to trigger plasma membrane rupture (PMR) in a multi-step pathway of lytic cell death called pyroptosis.” This statement suggests the N-terminal fragment directly activates NINJ1 and minor rewording would more accurately reflect the uncertain link between the two proteins. We did not intend to suggest GSDMD directly activates NINJ1, and have amended this text for greater clarity. (“*Once activated, CASP1 and CASP11 both cleave Gasdermin-D (GSDMD) to liberate an N-terminal pore-forming fragment (GSDMD-NT)^{5, 6} that permeabilizes the plasma membrane (PM). This can ultimately lead to plasma membrane rupture (PMR)⁷ by Ninjurin-1 (NINJ1), in a multi-step pathway of lytic cell death called pyroptosis⁸*”).

Reviewer #3

In this manuscript entitled ‘Pyroptotic F-actin remodelling engages CLEC9A, a bridge between innate and adaptive immunity’, the authors found that during pyroptosis, after GSDMD pore formation and before NINJ1-mediated PM rupture, cells generate F-actin enriched ‘filopodia’-like structures. Formation of filopodia depends on GSDMD pore formation and Ca²⁺ influx. Those filopodia

structures seem to remain after the cells burst and activate dendritic cells by the F-actin receptor CLEC9A, also known as DNDR-1. Overall the manuscript suggests a novel phenomenon and provided high resolution imaging data. We thank the Reviewer for their positive comments.

Many of the data were purely descriptive and did not even include any quantitative or statistical analysis (i.e. Fig. 1, Fig. 2) – since there were similar structures seen in non-pyroptotic cells this is very important. Given that we are the first to report filopodial assembly by cells undergoing pyroptosis, we felt it important to adequately describe these structures before going on to provide mechanistic data. Our revised manuscript shows substantial new quantitative data that offers new mechanistic insights, analysed by robust statistical techniques, as discussed below.

Moreover the images sometimes lacked controls not undergoing pyroptosis.

Every fixed microscopy image in Figs 1-3 includes one or more matched controls not undergoing pyroptosis (e.g. LPS prime alone control for LPS+Nig samples; PAM-primed plus mock transfection for PAM+ transfected LPS samples; or GSDMD-deficient cells that cannot undergo pyroptosis). In new Fig. 3 experiments in which we transfect cells with cell death effector-encoding mRNAs to induce diverse modes of cell death, mock-transfected cells are imaged and used for quantitation as appropriate controls. Our original manuscript submission did not include scanning electron microscopy of non-pyroptotic cells because of space constraints, but we now include scanning electron microscopy for non-pyroptotic cells (LPS-primed in **Extended Fig. 6**, PAM-primed in **Extended Fig. 7**), as well as time course analyses of cells undergoing pyroptosis (**new Extended Fig. 6-7**). Non-pyroptotic cells (unstimulated or primed in **Fig 1c**, **new Figs 1e, 1f, 1i, 2b-g**; inflammasome-deficient: **Fig. 2a, b**) show markedly and robustly significantly fewer filopodia than pyroptotic cells.

Moreover, some key questions need to be addressed. The most important are: Is this phenomenon specific to pyroptosis or does it occur any time the plasma membrane loses its integrity or with other pore-forming proteins that do not activate programmed cell death? We were also curious about this. Our revision incorporates several new experiments examining distinct cell death modes induced by pore-forming proteins, and their capacity to induce cell corpses rich in filopodia. Our new data show that

- (1) macrophage expression of the pore-forming fragment of GSDMD (GSDMD-NT) is not only necessary, but also sufficient to induce calcium-dependent filopodial assembly (**new Fig. 2h-i, new Fig. 3b-d**);
- (2) the pore-forming domain of other gasdermins (GSDMA3, GSDME) can similarly generate corpses crowned with filopodia (**new Fig. 3e-g**);
- (3) active MLKL is sufficient to induce filopodial assembly in necroptotic cells (**new Fig. 3h-j**).

We conclude from these data that filopodial assembly is not specific to pyroptosis, but can also occur during other forms of lytic cell death, such as necroptosis.

Myeloid cells such as macrophages and dendritic cells respond to membrane pores (including those that would not be directly lethal) by activating the NLRP3 inflammasome and downstream pyroptosis. Thus, the proposed experiment with a non-lytic pore-forming protein would be difficult to interpret, as we would anticipate that it would indirectly induce NLRP3-induced pyroptosis. We thus chose to use a slightly different experimental set-up to answer the excellent question of whether non-lytic pores that disrupt plasma membrane integrity can induce filopodia. To entirely uncouple GSDMD-induced filopodia from GSDMD-induced pyroptosis, we collaborated with the Kagan lab to examine filopodial assembly in macrophages with sub-lytic inflammasome-GSDMD signalling (for which they have coined the term cell ‘hyperactivation’). Stimulation with the hyperactivation stimulus, PGPC, generated macrophages and dendritic cells with a spectacular number of MYO10-capped projections per cell, most of which were highly branched (**new Fig. 1e-f, new Extended Fig. 2a-f**). Given that PGPC-stimulated BMDM and BMDC remain viable (manuscript refs 14, 33), this

indicates that the assembly of F-actin-rich projections does not require the cell to be undergoing programmed cell death. Our observation that PGPC induces hundreds more projections/cell than nigericin may indeed suggest the opposite – that cell death may halt the assembly of these F-actin-rich projections.

Does it occur in other cell types such as epithelial cells undergoing pyroptosis? Indeed, dendritic cells and epithelial cells both assemble MYO10-capped filopodia in response to inflammasome activators (**new Fig. 1e-g, new Extended Fig. 2a-g**).

Does the phenomenon have any functional significance? A large body of literature documents the capacity of CLEC9A to facilitate cDC1 sampling and presentation of dead cell antigens for adaptive immune responses *in vitro* and *in vivo* (e.g. manuscript refs 1-2, 43-48). We thus felt it appropriate to use the NI short letter format to document the key novelty of our study – that pyroptotic cells assemble F-actin-rich structures, thereby ensuring that pyroptotic corpses are a rich source of CLEC9A ligand, and are therefore immunogenic. Our revised manuscript now additionally demonstrates that GSDMD drives the assembly of F-actin-rich filopodia *in vivo* (**new Fig. 1i-j**) and that signalling by the GSDMD-calcium axis in pyroptotic macrophages is sufficient to upregulate cDC1 expression of CD80 and CD86 co-stimulatory molecules and MHCII (**new Fig. 5d-f**). Using WT versus *Clec9a*-deficient dendritic cells, we also show that pyroptotic corpses induce CLEC9A-dependent SYK phosphorylation in cDC1 cells (see **new Fig 5b**). We have further mapped the requirements for macrophage inflammasome signalling to trigger signalling by the CLEC9A-SYK axis in incoming cDC1 cells; GSDMD, extracellular calcium and NINJ1 are all required for macrophages to induce CLEC9A-SYK signalling in cDC1s (see **new Fig 5c**), while NINJ1 is not required to assemble filopodia (**new Fig. 2f-g**). This suggests that GSDMD pores allow calcium influx to drive filopodial assembly, while NINJ1-driven rupture is required to expose intracellular F-actin from macrophages for recognition by dendritic cell CLEC9A. Our revised manuscript thereby shows that the assembly of F-actin-rich projections during pyroptosis allows pyroptotic corpses to engage CLEC9A on incoming cDC1 cells.

Does it occur in human cells as well as in mouse systems? Indeed, our new data shows that diverse inflammasome activators trigger the assembly of filopodia in human epithelial cells (**new Fig. 1g, new Extended Fig. S2g**) and human macrophages (**new Extended Fig. 1h**).

How does CLEC9A on the outside of a cell bind to F-actin inside another cell? This is an excellent question, and was a point of curiosity for us. We now show that NINJ1-driven macrophage rupture is not required for filopodial assembly (**new Fig. 2f-g**) but is required for incoming cDC1 cells to access the (otherwise intracellular) F-actin of pyroptotic corpses (**new Fig. 5c**). This suggests that calcium influx through GSDMD pores is sufficient to drive filopodial assembly, while NINJ1-driven cell rupture is required to expose intracellular F-actin from macrophages for recognition by dendritic cell CLEC9A.

Do the filopodia form near GSDMD pores? When taken up by a cell, calcium rapidly diffuses throughout the cytosol, and this is confirmed by our new Ca^{2+} imaging studies (**new Fig. 2c-d, new Extended Fig. 5a, Videos 6-7**). We now show that GSDMD pores allow calcium to enter the cell, after which calcium readily diffuses throughout the cell (e.g. see Video 6). We anticipate that this drives filopodial assembly both proximal and distal to GSDMD pores, consistent with our data showing filopodia are ubiquitous throughout the cell periphery. We have trialled a range of commercial GSDMD antibodies and cannot find one that allows us to localise by microscopy endogenous GSDMD pores in macrophages (i.e. the antibodies we have tested all appear to stain GSDMD knockout macrophages). We are thus unable to address the proximity of filopodia to GSDMD pores experimentally.

Major points:

1. As the authors also cited, there are publications in the past to investigate the involvement of cytoskeleton during pyroptosis. However, the previous report looking at actin during pyroptosis did not observe such filopodia. The authors failed to discuss the differences or the reasons behind the different phenomenon observed. The Reviewer suggests that pyroptosis-associated filopodia have never been observed in previous imaging studies, including the previous report of actin during pyroptosis (which we presume to be the key 2019 PNAS study: 10.1073/pnas.1818598116). Actually, we often observe these projections (albeit at low resolution) in published studies, although the authors themselves either did not notice or were preoccupied with different research objectives and chose not to pursue this phenotype. In the 2019 PNAS study to which we presume the Reviewer is referring, the authors study cytoskeletal changes during human myeloid cell pyroptosis – here, filopodial projections can be observed in low contrast in **Movie S1** (see movie still images below, with filopodia evident at low resolution indicated by yellow arrowheads). This was not commented on in the publication, as they implemented a different imaging approach that did not resolve these structures as finely as our own approaches. Of note, filopodia are not high-contrast structures, and so are often difficult to observe using DIC or phase-contrast microscopy (which conversely, resolve pyroptotic blisters very well, as these are high-contrast structures).

Davis et al (2019) **PNAS** 10.1073/pnas.1818598116
S1 Movie frames: THP-1 cells stimulated with PMA+nigericin

In a second published example (Ye et al., *Biochem Pharmacol* 2021 10.1016/j.bcp.2021.114791), scanning electron microscopy of primary murine macrophages stimulated with LPS+nigericin demonstrates that macrophages have few filopodia before cell rupture, but abundant filopodia during and after pyroptosis (see below). Our electron microscopy is consistent with these images (see **new Extended Fig. 6** for nigericin time course).

Ye et al (2021) **Biochem Pharmacol** 10.1016/j.bcp.2021.114791
Figure 1C. Murine bone marrow-derived macrophages stimulated with LPS+nigericin

More recently, a 2023 Nature article shows quite high-resolution DIC imaging in which filopodia are clearly evident (see yellow arrows) before PM bleb (pyroptotic blister) formation (see below). Again, the Ye *et al* and Degen *et al* manuscripts pursued a different research question and did not investigate these filopodial phenotypes.

Degen et al (2023) **Nature** doi.org/10.1038/s41586-023-05991-z

Extended Fig 1f. Murine bone marrow-derived macrophages transfected with LPS. Pyroptotic cells lost membrane integrity (acquisition of PI, a membrane-impermeable DNA dye) and were labeled by annexin-V. Plasma membrane (PM) blebs observed during pyroptosis (white arrows). DIC, differential interference contrast. Scale bars, 20 μ m.

Meanwhile, many publications have shown the ballooning phenotype of pyroptosis, but in this manuscript, those were not prominent. Those major discrepancies are important and clearly need to be addressed. We did observe and present pyroptotic ballooning/blisters. Our 4D LLSM data clearly indicates cell swelling and blistering (Fig. 4a-b). Other figures in our manuscript are maximum-intensity projections of Z-stacks and do not show blisters because: (1) these are difficult to observe in a cell from the top-down in a flat image, especially when dealing with irregular-shaped and flat primary macrophages as opposed to round spherical macrophage-like cell lines, (2) pyroptotic blisters are not actin-rich and are therefore not labelled well with actin dyes in our fluorescence microscopy figures, and (3) these blisters are delicate and do not survive treatment with fixatives, obscuring them from observation in our fixed confocal and scanning electron microscopy images. It is for these reasons that we imaged ballooning at high resolution separately, using the LLSM technique that is well suited to capturing these structures (Fig. 4a-b). We have clarified this in our revised text.

2. Critical controls are missing in several experiments. For example, in Figure 2, scanning EM is only performed with cells long after PM rupture. It is important to analyze cells without treatment, before GSDMD pore formation, after pore formation but before cell rupture, and after rupture. We did not include non-pyroptotic cells in Fig 2 electron microscopy because of space constraints, but now include time course data for both nigericin stimulation (Extended Fig. 6) and LPS transfection (Extended Fig. 7). For each of these stimuli, figure panel A shows GSDMD cleavage and cell lysis over a detailed time course of signalling, identifying the suggested time points:

- Without an inflammasome activator (i.e., prime-only samples not undergoing pyroptosis)
- Before GSDMD pore formation (5 mins nigericin, 10 mins LPS transfection)
- After GSDMD pore formation but before cell rupture (10 mins nigericin, 30 mins LPS transfection)
- After cell rupture (45 mins nigericin, 120 mins LPS transfection)

Panels B-E of these figures then go on to show scanning electron microscopy of macrophages at each of these time points, for nigericin stimulation (Extended Fig. 6) and LPS transfection (Extended Fig. 7), as requested.

Ideally including kinetics and an apoptosis control and controls with bacterial pore-forming proteins will need to be done to conclude that filopodia formation is specific to pyroptosis. Pyroptosis kinetics is addressed in the previous response. We now also provide scanning electron microscopy of a macrophage undergoing apoptosis in new **Fig. 3n**. Our revision incorporates several new experiments examining distinct cell death modes and their capacity to induce cell corpses rich in filopodia, to determine whether filopodial assembly is specific to pyroptosis. We now show that:

- (1) macrophage expression of the pore-forming fragment of GSDMD (GSDMD-NT) is sufficient to induce calcium-dependent filopodial assembly (**new Fig. 2h-i, new Fig. 3b-d**);
- (2) other gasdermins can similarly generate corpses crowned with filopodia (**new Fig. 3e-g**);
- (3) active MLKL is sufficient to induce filopodial assembly in necroptotic cells (**new Fig. 3h-j**); and
- (4) diverse inducers of apoptotic cell death do not trigger the assembly of filopodia (**new Fig. 3k-n**).

Given that *Gsdmd* deficiency diverts inflammasome-signalling cells from pyroptosis to apoptosis, we had the opportunity to directly compare pyroptotic versus apoptotic death in cells stimulated identically, by comparing the nigericin responses of wild-type versus *Gsdmd*-deficient macrophages. We now show that while nigericin-induced pyroptotic corpses are crowned by filopodia, nigericin-stimulated apoptotic cells do not exhibit filopodia, and their shrinking cell bodies instead leave behind retraction fibres (**Fig. 2a, Fig. 3m**). Such nigericin-induced apoptotic cells do not induce CLEC9A-SYK signalling in incoming cDC1 cells (**new Fig. 5c**).

In all, these studies lead us to conclude that the assembly of F-actin-rich filopodia is common to multiple modes of programmed cell lysis, but is not a feature of apoptotic cell death.

In another publication <https://link.springer.com/article/10.1007/s12017-019-08586-y>, scanning EM is carried out with different stages of pyroptosis and the results are not completely consistent with this manuscript. The referenced publication examines DHA-induced cell death in the BV2 microglia-like cell line and does not observe the assembly of filopodial structures. BV-2 microglia-like cells are far removed from primary macrophages, and in our hands, BV-2 cells do not signal via the inflammasome pathway because they do not express the inflammasome adaptor ASC (a finding also supported by Fig. 6 of publication [doi:10.1007/s10571-022-01285-6](https://doi.org/10.1007/s10571-022-01285-6)). DHA is not well established as an inflammasome activator, and nowhere in this study (or the author's preceding study first reporting DHA-induced BV2 death) do the authors confirm that DHA-induced cell death is indeed pyroptotic (e.g. by rescuing cells from death by GSDMD knockout). The evidence that these cells were undergoing pyroptosis is therefore extremely weak. When evaluated in the context of our high-resolution imaging approaches to characterise filopodia, coupled with genetic evidence of pyroptosis and published images that capture similar structures in rigorous studies of macrophage pyroptosis in high impact journals (see our response to major point 1 from this Reviewer, above), we feel that that the BV2 study is not a convincing counter to our extensive and high-resolution characterisation of this phenotype.

3. The author proposed the model that GSDMD pore mediate Ca²⁺ influx which activate Myosin X to trigger filopodia formation. This is a novel point but was not analyzed clearly. If filopodia formation is important for the F-actin rich cell corpse maintaining, and following DC activation, then instead of using casp mutant or *gsdmd* mutant (which blocks everything), they should inhibit actin assembly/Myosin X (inhibitors available and/or KD or KO) to see the filopodia, cell corpse, DC activation. We agree with the Reviewer's reasoning. We were enthusiastic about doing exactly this experiment but had some reservations about whether it would be feasible given the importance of the actin network for cell viability and signalling. We have been unable to find an inhibitor that specifically blocks Myosin X function, and so have turned to siRNA approaches (as we feared that CRISPR-knockout cells would not be viable or would be unable to signal via inflammasomes). Our routine siRNA protocol incubates cells for 24 to 48h after siRNA delivery to allow protein knockdown and cell recovery. Given our reservations about the viability of cells with *Myo10* knockdown, we first tried 24 h knockdown, which was successful in dramatically lowering Myo10 protein levels

(**Rebuttal Fig. 1a** below). However, *Myo10*-knockdown cells had radically altered morphology (shrunken cell body and condensed nuclei) and looked apoptotic by confocal imaging (**Rebuttal Fig. 1b**). *Myo10*-knockdown cells also showed compromised responses to NLRP3 or CASP11 inflammasome-activating stimuli for signalling outputs that are independent of filopodia: cell death and IL-1 β production (**Rebuttal Fig. 1c**). We thus modified our approach, hoping that shorter-term siRNA treatment would allow us knockdown *Myo10* without compromising cell health and signalling. However, 8 h treatment with *Myo10* siRNA again compromised inflammasome signalling outputs (**Rebuttal Fig. 1d**) despite only modestly reducing *Myo10* protein levels (**Rebuttal Fig. 1a**). We conclude from these data that interfering with Myosin-X functions fundamentally alters macrophage functions, and renders these cells unable to signal via inflammasomes.

To address this important question in a slightly different way, we now show data from several new experiments:

- (1) We show that macrophage expression of the pore-forming fragment of GSDMD (GSDMD-NT) is sufficient to induce calcium-dependent filopodial assembly (**new Fig. 2h-i, new Fig. 3b-d**);
- (2) We expand upon calcium dynamics during pyroptotic signalling, showing that:
 - nigericin induces Ca²⁺ entry into cells, which on average peaks around 2 mins before the cell becomes permeable (**new Fig. 2c**);
 - Ca²⁺ quickly diffuses within the cell after entry (**new Video 6**);
 - nigericin-induced Ca²⁺ cell entry precedes the assembly of filopodia (**new Video 6**); and

- nigericin-induced Ca^{2+} entry into the cell requires GSDMD (**new Fig. 2c, new Videos 6 and 7**).
- (3) By specifically blocking calcium flux through GSDMD pores with EGTA, we show that:
- EGTA blocks filopodial assembly in wild-type macrophages stimulated with nigericin (**Fig. 2e**), and macrophages ectopically expressing the pore-forming domain of GSDMD to induce pyroptosis (**new Fig. 2i**), without affecting pyroptotic cell death (**Extended Fig. 5b, new Fig. 2h, new Fig. 5c**).
 - Pyroptotic corpses with filopodia (-EGTA) stimulate CLEC9A-SYK signalling in incoming cDC1 cells, while pyroptotic corpses without filopodia (+EGTA) do not (**Fig. 5c**).
 - Pyroptotic corpses with filopodia (-EGTA) trigger the upregulation of dendritic cell activation markers (CD80, CD86, MHCII), while pyroptotic corpses without filopodia (+EGTA) do not (**Fig. 5d-f**).

In the manuscript, Extended Fig 4b, actin polymerization is tested but the figure is not clear and there aren't statistics. We have removed this figure to focus on areas of key interest to all three Reviewers – the signalling events that allow cells undergoing pyroptotic death (and other death modes) to assemble filopodia, and the interactions of these F-actin-rich structures with CLEC9A.

4. In Figure 5A,B, the difference is very subtle. It's good to test SYK phosphorylation with immunoblotting and test CLEC9A-SYK complex formation by CoIP. We tried immunoblot, but think our new flow cytometry method is more robust as it allows us to quantify phospho-SYK specifically in the cDC1 population (which is not possible by western blot of lysates of co-cultured cells). Fig. 5 now shows CLEC9A interaction with phospho-SYK by proximity ligation assay (**Fig. 5a**), and uses wild-type versus *Clec9a*-deficient dendritic cells to show that pyroptotic corpses trigger SYK phosphorylation via CLEC9A (**new Fig. 5b-c**). Also for Fig 5C,D, *clec9a* KO cells should be included as controls. We have adopted this excellent suggestion, and now show SYK phosphorylation in cDC1 from wild-type versus *Clec9a*-deficient mice to show that pyroptotic corpses trigger SYK phosphorylation via CLEC9A (**new Fig. 5b-c**).

5. The paper refers to the “basal” membrane of non-polarized hematopoietic cells. I don't think this usage is correct. The protruding part of migrating cells (the site of the lamellopodia) are usually called the apical surface. This should be studied in polarized epithelial cells. We apologise, and have corrected this text so that we no longer refer to a “basal membrane” – if we were to do so, we agree we would need to study the responses of polarised epithelia. Do the F-actin structures form when the cells are not adherent? We were also curious about this, and our revised manuscript directly addresses this *in vitro* and *in vivo*. To determine whether inflammasome-driven filopodial assembly required microenvironmental mechanical cues or adherence, we performed live imaging on macrophages cultured in suspension. Indeed, suspended macrophages undergoing nigericin-induced pyroptosis assembled F-actin-rich filopodia (**new Fig. 1h, new Extended Fig. 3**) similar to adherent macrophages. We also challenged mice with LPS to induce CASP11-GSDMD signalling, and found that *in vivo*-challenged cells similarly exhibit GSDMD-dependent filopodia (**new Fig. 1i-j**). Thus, cell adhesion is not required for macrophages to assemble pyroptotic filopodia, and these are not a consequence of culturing macrophages on plastic.

6. It is critical for a high impact publication to show that this phenomenon has functional and *in vivo* significance. A large body of literature documents the capacity of CLEC9A to facilitate cDC1 sampling and presentation of dead cell antigens for adaptive immune responses *in vitro* and *in vivo* (e.g. manuscript refs 1-2, 43-48). We thus felt it appropriate to use the NI short letter format to document the key novelty of our study – that pyroptotic cells assemble F-actin-rich structures, thereby ensuring that pyroptotic corpses are a rich source of CLEC9A ligand, and are therefore immunogenic. Our revised manuscript now additionally demonstrates that GSDMD drives the assembly of F-actin-rich filopodia *in vivo* (**new Fig. 1i-j**) and that signalling by the GSDMD-calcium axis in pyroptotic

macrophages is sufficient to upregulate cDC1 expression of CD80 and CD86 co-stimulatory molecules and MHCII (**new Fig. 5d-f**). Using WT versus *Clec9a*-deficient dendritic cells, we also show new data that pyroptotic corpses induce CLEC9A-dependent SYK phosphorylation in cDC1 cells (see **new Fig 5b**). We have further mapped the requirements for macrophage inflammasome signalling to trigger signalling by the CLEC9A-SYK axis in incoming cDC1 cells; GSDMD, extracellular calcium and NINJ1 are all required for macrophages to induce CLEC9A-SYK signalling in cDC1s (see **new Fig 5c**). This suggests that GSDMD pores allow calcium influx to drive filopodial assembly, while NINJ1-driven rupture is required to expose intracellular F-actin from macrophages for recognition by dendritic cell CLEC9A. Our revised manuscript thereby shows that the assembly of F-actin-rich projections during pyroptosis allows pyroptotic corpses to activate CLEC9A on incoming cDC1 cells.

7. I believe that MCC950 is a covalent binder to NLRP3 that is Cys-reactive. As such it is unlikely to be completely “specific”. This is incorrect – MCC950 is not a Cys-reactive or covalent binder. Perhaps the Reviewer is confusing this NLRP3 inhibitor with one of the available GSDMD inhibitors (all of which covalently bind to a reactive Cys in GSDMD, amongst other protein targets). We discovered that MCC950 is an NLRP3 inhibitor that does not block any other inflammasomes (Coll et al., Nature Medicine 2016), and further identified it as a direct **non-covalent** NLRP3 binder (Coll et al, Nat Chem Biol 2019). This tool compound is now used in >7000 articles, and its robust specificity for NLRP3 underpins the development of several MCC950 analogs (including ours – Selnoflast – acquired by Roche) that are now in human phase I and II trials as NLRP3 blockers. Regardless of this, experiments with MCC950 were also performed with inflammasome knockout cells (e.g. *Casp1.C284A* macrophages), so the capacity of MCC950 to block NLRP3-induced filopodia was not an off-target effect of MCC950.

8 What happens if GSDMD is cleaved but pore formation is blocked (with DMF, DSF)? Does it affect the formation of these processes? This is an interesting question. Our model, supported by substantial new data (see above) predicts that GSDMD pore formation is required for calcium influx and resultant filopodial assembly. We attempted the experiment suggested by the Reviewer, and found that DSF is a weak inhibitor of GSDMD-dependent signalling outputs in our hands (**Rebuttal Fig. 2a** below). DSF also has a myriad of off-target responses with IC₅₀s under 10 μM (aldehyde dehydrogenase, dopamine beta-hydroxylase, caspase-1, caspase-3, GSDME; see our review doi.org/10.1016/j.tips.2022.04.003). We therefore proceeded with DMF, which was more potent in these assays (**Rebuttal Fig. 2a**). We hoped to find a concentration of DMF that did not block inflammasome signalling, as (amongst other off-target effects) DMF can block NLRP3 inflammasome signalling (see doi.org/10.1016/j.tips.2022.04.003). We thus tested DMF in both NLRP3 and CASP11 activation assays, and found that in both cases, DMF blocked GSDMD **cleavage** upstream of GSDMD pore formation (**Rebuttal Fig. 2b** below). This is consistent with a recent study from Kate Fitzgerald’s lab (DOI:10.1126/science.abb9818), which shows that DMF succinates full-length GSDMD, thereby preventing its cleavage by CASP1 and CASP11. We thus cannot use DMF to specifically block GSDMD pores while leaving GSDMD cleavage unaffected.

We thus took the opposite approach to that suggested by the Reviewer to answer the same question of whether GSDMD pore formation is necessary, or indeed, sufficient for filopodial assembly. Instead of blocking GSDMD pore formation downstream of cleavage, we ectopically expressed the pore-forming domain of GSDMD and other gasdermins to determine whether this was sufficient for filopodial assembly. Indeed, macrophage expression of the pore-forming fragment of GSDMD (GSDMD-NT) is sufficient to induce calcium-dependent filopodial assembly (**new Fig. 2h-i**, **new Fig. 3b-d**), and the pore-forming domain of other gasdermins can similarly generate corpses crowned with filopodia (**new Fig. 3e-g**). Thus, formation of the gasdermin pore is not only necessary, but is sufficient to induce filopodial assembly.

Rebuttal Figure 2. DMF affects inflammasome signalling upstream of GSDMD pore formation. (A) Wild-type LPS-primed BMDM treated with DMF or DSF 15 mins before exposure to nigericin for 45 mins. The GSDMD-dependent signalling outputs, IL-1 β release and cell lysis, were measured over a dose-response range. (B) Primed (PAM overnight, LPS for 4 h) BMDM were treated with 25 μ M DMF for 15 mins before exposure to nigericin (45 mins) or transfected LPS (2 h). GSDMD and pro-IL-1 β cleavage were assessed by western blot.

Are they formed in “hyperactivated” myeloid cells that form pores but survive? We were also curious about this, and collaborated with the Kagan lab to answer this question. Stimulation with the hyperactivation stimulus, PGPC, generated macrophages and dendritic cells with a spectacular number of MYO10-capped projections per cell, most of which were highly branched (new Fig. 1e-f, new Extended Fig. 2a-f). So yes, hyperactivated myeloid cells indeed assemble filopodia, and cell death is not required for filopodial assembly.

9. Is filopodia formation blocked by chelating intracellular calcium? Yes. We used the cell permeable form of EGTA (EGTA-AM) to chelate intracellular calcium, and found that it does indeed block filopodial formation when we ectopically express the pore-forming domain of GSDMD (new Fig. 2i).

Minor points

1. Sir-Actin is a cell permeable actin dye, why does actin (such as in Fig 1a/b, magenta) only show up when cell is ruptured? We add Sir-Actin to macrophages immediately before we add the inflammasome activating stimulus (e.g. nigericin); under these conditions, we find that it is only weakly cell permeable. As stated in our manuscript and our referenced article (manuscript ref 26), we deliberately applied SiR-actin in this manner so that it does not label living macrophages, so that we see it enter cells and label F-actin only when cells become permeable and the intracellular F-actin is exposed to the (largely extracellular) dye.
2. Typo ‘Fig 4H’ should be ‘Fig 4F’ in the text. We apologise for this mistake, which we have fixed.
3. Reference 16 and 18 are the same paper. We apologise for this mistake, which we have fixed.